# POST-HOC PROBABILISTIC VISION-LANGUAGE MODELS

**Anton Baumann**[1,2,*]   **Rui Li**[3]   **Marcus Klasson**[8]   **Santeri Mentu**[3,4]
**Shyamgopal Karthik**[1,2,5,6]   **Zeynep Akata**[1,2,6,7]   **Arno Solin**[3,4]   **Martin Trapp**[9,10]

[1]Technical University of Munich  [2]Helmholtz Munich  [3]ELLIS Institute Finland & Aalto University
[4]Finnish Center for Artificial Intelligence  [5]University of Tübingen  [6]Munich Center for Machine Learning
[7]Munich Data Science Institute  [8]Ericsson Research  [9]KTH Royal Institute of Technology  [10]Digital Futures

## ABSTRACT

Vision-language models (VLMs), such as CLIP and SigLIP, have found remarkable
success in classification, retrieval, and generative tasks. For this, VLMs determin-
istically map images and text descriptions to a joint latent space in which their
similarity is assessed using the cosine similarity. However, a deterministic mapping
of inputs fails to capture uncertainties over concepts arising from domain shifts
when used in downstream tasks. In this work, we propose post-hoc uncertainty
estimation in VLMs that does not require additional training. Our method leverages
a Bayesian posterior approximation over the last layers in VLMs and analytically
quantifies uncertainties over cosine similarities. We demonstrate its effectiveness
for uncertainty quantification and support set selection in active learning. Com-
pared to baselines, we obtain improved and well-calibrated predictive uncertainties,
interpretable uncertainty estimates, and sample-efficient active learning. Our results
show promise for safety-critical applications of large-scale models.

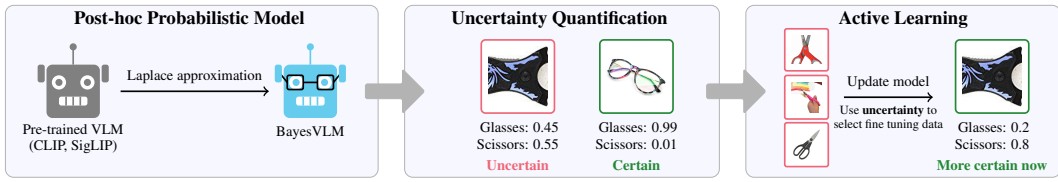

Figure 1: We introduce an efficient and effective post-hoc method to provide uncertainty estimates
for vision-language models (*e.g.*, CLIP, SigLIP) using a Laplace approximation. We demonstrate that
uncertainty estimates derived from this approximation improve the calibration of these models on
several zero-shot classification benchmarks (Sec. 4.1) and are effective in active learning (Sec. 4.2).

## 1 INTRODUCTION

Pre-trained large-scale vision-language models (VLMs) (Bordes et al., 2024; Zhang et al., 2024),
such as CLIP (Radford et al., 2021) and SigLIP (Zhai et al., 2023), have achieved remarkable success
in tasks like zero-shot classification, retrieval, and generation, driven by training on billion-scale data
sets (Gadre et al., 2023; Schuhmann et al., 2022). However, when employing large-scale machine
learning models reliably in real-world settings and on downstream applications, we expect them not
only to provide accurate predictions but also to enable us to quantify their predictive uncertainties.
Obtaining efficient and effective uncertainty estimates is particularly relevant for safety-critical
applications, as well as when making decisions based on those estimates, such as in active learning.

Previous work on uncertainty quantification for VLMs has primarily focused on calibration (Guo
et al., 2017; Tu et al., 2023), test-time adaptation (Ayhan & Berens, 2018; Farina et al., 2024; Yoon
et al., 2024; Lafon et al., 2025), fine-tuning (Fort et al., 2021; Tu et al., 2023; Ju et al., 2025),
or training probabilistic VLMs from scratch (Chun, 2024; Chun et al., 2025). However, each of
those approaches has limitations regarding their applicability in real-world settings. For example,
calibration methods cannot capture epistemic uncertainties, adapter and retraining-based methods

---

*Work partially done during an internship at Aalto University.

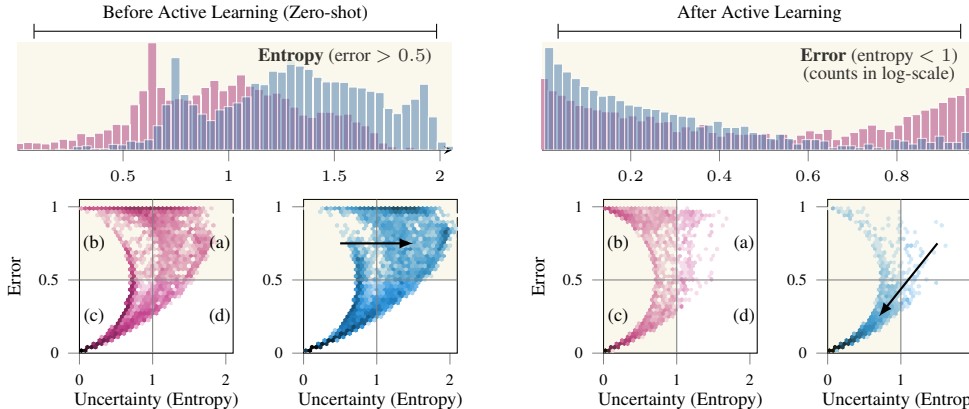

Figure 2: Predictive error vs. uncertainty (entropy) on the EuroSAT data set (Helber et al., 2019) for the OpenCLIP ViT-H-14 model. **The zero-shot** comparison (left side) of the original model (■) and its Bayesian counterpart (■) indicates that our Bayesian model exhibits better calibration and substantially reduces overconfident predictions. **Active Learning** results (right side) show that those improvements lead to a substantially reduced misclassification rate after adaptation; quadrant (b).

come with substantial computational demands and require retraining in streaming/active learning settings, and test-time adaptation methods significantly increase inference costs.

To manifest efficient and effective uncertainty quantification for the reliable application of VLMs, we identify the following desiderata: The method should be applicable to any VLM architecture (*model-agnostic*), and uncertainties should be obtained in a *post-hoc* manner without retraining the model from scratch. During inference, it should have low to no computational overhead (*efficient*) and capture relevant sources of uncertainties (*effective*). Finally, the method should extract uncertainties from the original VLM without adding new layers or adapters that require training (*training-free*).

The Bayesian framework provides a principled way to model epistemic and aleatoric uncertainties, and has shown promise as a 'toolbox' for uncertainty quantification in deep learning (Papamarkou et al., 2024). Consider Fig. 2, which shows results on the EuroSAT data set (Helber et al., 2019), a land use and land cover classification task based on Sentinel-2 satellite images, for the popular OpenCLIP model (■). We observe that the Bayesian counterpart (■) results in less overconfident predictions before active learning (compare quadrant b and a) and substantially reduces the error in the predictions after active learning, compared to the fine-tuned OpenCLIP model (■). Much of the misclassification of the OpenCLIP model after active learning can be attributed to its overconfident behaviour before and after active learning, indicating the benefits of using a Bayesian formulation.

This work proposes BayesVLM, an efficient and effective post-hoc uncertainty quantification method for pre-trained VLMs that adheres to the outlined desiderata. We leverage a Laplace approximation (MacKay, 1992) to the Bayesian posterior, thereby eliminating the need for additional training, architectural changes, or modifications to the training objective. For this, we introduce independent probabilistic models for each modality, adhering to the i.i.d. assumption and enabling efficient posterior inference. Further, we derive an analytical expression for the distribution over cosine similarities for efficient uncertainty propagation. We evaluate our approach on zero-shot classification benchmarks and for uncertainty-aware active fine-tuning (Gal et al., 2017; Hübotter et al., 2024), finding improvements in performance over baselines in both scenarios. Lastly, we assess the efficiency and robustness of our approach (BayesVLM) and find that BayesVLM provides efficient, effective, and robust uncertainty estimates, even when the VLM is pre-trained on proprietary data.
**Contributions** The overall contributions are illustrated in Fig. 1 and can be summarised as follows: *(i)* we propose BayesVLM, an efficient and effective post-hoc method for uncertainty quantification in pre-trained VLMs, without architecture changes or further training (Sec. 3); *(ii)* we present the first direct Bayesian formulation of vision-language models and derive an analytical expression of the distribution over cosine similarities for efficient uncertainty propagation (Sec. 3.2); *(iii)* we demonstrate the efficacy of BayesVLM in both zero-shot and active learning settings, showing improvements over baselines in both settings. And we assess its efficiency and robustness, finding that BayesVLM provides robust estimates while introducing little to no computational overhead (Sec. 4).

## 2 RELATED WORK

**Vision-language models** Models like CLIP (Radford et al., 2021) and SigLIP (Zhai et al., 2023), trained on massive data sets such as LAION (Schuhmann et al., 2022), have become widespread in various applications, including zero-shot classification, generative modeling (Rombach et al., 2022; Podell et al., 2024), and retrieval (Saito et al., 2023; Karthik et al., 2024). This work presents an effective post-hoc approach to uncertainty estimation for these pre-trained VLMs.

**Uncertainty in vision-language models** Quantifying uncertainties in VLMs has observed increasing interest, with approaches involving learning probabilistic embeddings, for example, by learning additional probabilistic adapters (Chun et al., 2025; Lafon et al., 2025) or through pre-training/fine-tuning with a probabilistic loss (Chun, 2024; Ju et al., 2025). In addition, recent approaches also explored training-free uncertainty quantification, *e.g.*, through test-time augmentation (Ayhan & Berens, 2018) or zero-shot out-of-distribution detection (Fu et al., 2025). Another key approach is to solely focus on calibration through methods such as temperature scaling (Guo et al., 2017). Further related works are discussed in App. B.1. In contrast, we present a training-free post-hoc approach that does not require architectural changes, but estimates the Bayesian posterior of a pre-trained model and efficiently propagates uncertainty arising from the Bayesian posterior to the VLM output.

**Active learning** The goal of active learning (Ren et al., 2021; Settles, 2009) is to improve model performance by 'actively' selecting additional informative data through an acquisition function (Holub et al., 2008; Sener & Savarese, 2018). A particularly relevant area is Bayesian active learning (MacKay, 1992; Gal et al., 2017), where acquisition functions leverage model uncertainties. Notable examples include the BALD score (Houlsby et al., 2011) and EPIG (Bickford Smith et al., 2023), both of which are functions of information gain. While such methods are gaining traction in large language models (Hübotter et al., 2025), they remain relatively underexplored for VLMs, where ad-hoc strategies are more prevalent. In our work, we bridge this gap.

## 3 METHODS

**Notation** We denote vectors by bold lower-case letters (*e.g.*, $\boldsymbol{x}, \boldsymbol{a}$) and use bold upper-case letters for matrices (*e.g.*, $\boldsymbol{X}, \boldsymbol{P}$). Further, sets are denoted in upper-case calligraphic letters (*e.g.*, $\mathcal{D}, \mathcal{I}$) and model parameters or hyperparameters are denoted using Greek letters (*e.g.*, $\alpha, \boldsymbol{\theta}$). In particular, let $\boldsymbol{x}_i^{\text{IMG}} \in \mathbb{R}^{p_{\text{IMG}}}$ and $\boldsymbol{x}_j^{\text{TXT}} \in \mathbb{R}^{p_{\text{TXT}}}$ denote the $i^{\text{th}}$ image and $j^{\text{th}}$ text description, respectively. Further, let $\phi \colon \mathbb{R}^{p_{\text{IMG}}} \to \mathbb{R}^{d_{\text{IMG}}}$ and $\psi \colon \mathbb{R}^{p_{\text{TXT}}} \to \mathbb{R}^{d_{\text{TXT}}}$ denote the image and text encoders of the VLM, where $p_{\text{IMG}}$ and $p_{\text{TXT}}$ are the respective input dimensionalities and $d_{\text{IMG}}, d_{\text{TXT}}$ is the dimensionality of the respective feature space. Then, by denoting the linear image and text projections as $\boldsymbol{P} \in \mathbb{R}^{d \times d_{\text{IMG}}}$ and $\boldsymbol{Q} \in \mathbb{R}^{d \times d_{\text{TXT}}}$ respectively, the feature embeddings in the joint space can be written as $\boldsymbol{g} = \boldsymbol{P}\phi(\boldsymbol{x}^{\text{IMG}})$ and $\boldsymbol{h} = \boldsymbol{Q}\psi(\boldsymbol{x}^{\text{TXT}})$. We write $\boldsymbol{G}$ and $\boldsymbol{H}$ to denote the matrices of stacked image and text embeddings, respectively, whose rows correspond to the individual $\boldsymbol{g}_i$ and $\boldsymbol{h}_j$. Lastly, we use the hat symbol to denote unit-length normalised vectors, *e.g.*, $\hat{\boldsymbol{g}} = \boldsymbol{g}/\|\boldsymbol{g}\|$. The notation is listed in full in App. A.

### 3.1 BACKGROUND

**Language-image pre-training** We consider VLMs trained by minimising the InfoNCE loss (Oord et al., 2018) (*e.g.*, CLIP (Radford et al., 2021)) and present additional experiments for the SigLIP loss (Zhai et al., 2023). Specifically, the InfoNCE loss is defined as the sum of two cross-entropy terms, one for each relational direction—image to text ($\mathcal{L}_{\text{CE}}(\boldsymbol{X}^{\text{IMG}}, \boldsymbol{X}^{\text{TXT}})$) and text to image ($\mathcal{L}_{\text{CE}}(\boldsymbol{X}^{\text{TXT}}, \boldsymbol{X}^{\text{IMG}})$). The total loss is defined as follows $\mathcal{L}_{\text{InfoNCE}}(\boldsymbol{X}^{\text{IMG}}, \boldsymbol{X}^{\text{TXT}}) =$

$$-\underbrace{\frac{1}{2n}\sum_{i=1}^{n}\log\frac{\exp(t\hat{\boldsymbol{g}}_i^\top\hat{\boldsymbol{h}}_i)}{\sum_{j=1}^{n}\exp(t\hat{\boldsymbol{g}}_i^\top\hat{\boldsymbol{h}}_j)}}_{\text{IMG}\to\text{TXT},\ \mathcal{L}_{\text{CE}}(\boldsymbol{X}^{\text{IMG}},\boldsymbol{X}^{\text{TXT}})} - \underbrace{\frac{1}{2n}\sum_{i=1}^{n}\log\frac{\exp(t\hat{\boldsymbol{h}}_i^\top\hat{\boldsymbol{g}}_i)}{\sum_{j=1}^{n}\exp(t\hat{\boldsymbol{h}}_i^\top\hat{\boldsymbol{g}}_j)}}_{\text{IMG}\leftarrow\text{TXT},\ \mathcal{L}_{\text{CE}}(\boldsymbol{X}^{\text{TXT}},\boldsymbol{X}^{\text{IMG}})}, \quad (1)$$

where $t$ is a learnable temperature parameter, $n$ denotes the number of image-text pairs, and $\hat{\boldsymbol{g}}$ and $\hat{\boldsymbol{h}}$ are the unit-length normalised embeddings. This contrastive loss function encourages embeddings for matching image-text pairs to be similar while simultaneously pushing unrelated image-text pairs away from each other (Oord et al., 2018). In practice, evaluating this loss is infeasible on billions of

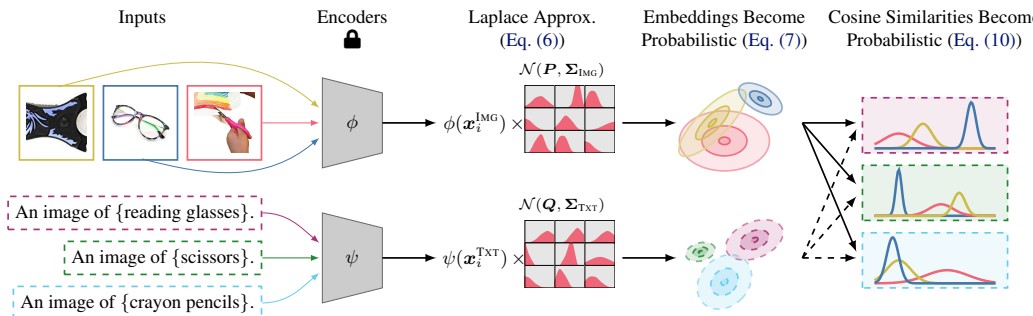

Figure 3: **Illustration of uncertainty propagation in BayesVLMs:** We estimate uncertainties over the last layers of both encoders using a Laplace approximation, which induces probabilistic feature embeddings. We then approximate the distribution over cosine similarities by estimating the expected value and variance. The cosine similarity distribution is then propagated to the VLM output.

data points. The common practice adopted is to evaluate it on a sufficiently large batch. Recently, the SigLIP loss (Zhai et al., 2023) was proposed as an alternative to the InfoNCE loss, a binary classification loss over cosine similarities. Further details on SigLIP are given in App. B.2.

**Laplace approximation**    Given a data set $\mathcal{D} = \{(\boldsymbol{x}_i, \boldsymbol{y}_i)\}_{i=1}^n$ and denote the neural network parameters as $\boldsymbol{\theta}$. In Bayesian deep learning, we aim to estimate the posterior distribution, *i.e.*,

$$p(\boldsymbol{\theta} \mid \mathcal{D}) = \frac{p(\boldsymbol{\theta}) \prod_{i=1}^n p(\boldsymbol{y}_i \mid \boldsymbol{x}_i, \boldsymbol{\theta})}{\int_{\boldsymbol{\theta}} p(\boldsymbol{\theta}) \prod_{i=1}^n p(\boldsymbol{y}_i \mid \boldsymbol{x}_i, \boldsymbol{\theta}) \, \mathrm{d}\boldsymbol{\theta}} = \frac{\text{prior} \times \text{likelihood}}{\text{marginal likelihood}}. \tag{2}$$

Since the marginal likelihood involves an intractable high-dimensional integral, we approximate the posterior. We adopt the Laplace approximation (LA) (MacKay, 1992), a post-hoc method that has been increasingly used in the Bayesian deep learning community (Daxberger et al., 2021; Li et al., 2025; Meronen et al., 2024; Ritter et al., 2018; Roy et al., 2022; Scannell et al., 2024).

Specifically, LA fits a Gaussian distribution to the posterior, centred at the MAP estimate of a *pretrained* model, and is therefore 'post-hoc'. The *prior* is implicitly defined by the L2 regularisation (weight decay) commonly used during training (Radford et al., 2021; Zhai et al., 2023), and corresponds to a diagonal Gaussian prior on the parameters, *i.e.*, $p(\boldsymbol{\theta}) = \mathcal{N}(\boldsymbol{0}, \lambda^{-1}\boldsymbol{I})$. The *likelihood* is defined by the training loss. The final approximate posterior is given as $p(\boldsymbol{\theta} \mid \mathcal{D}) \approx \mathcal{N}(\boldsymbol{\theta}_{\mathrm{MAP}}, \boldsymbol{\Sigma})$ where $\boldsymbol{\theta}_{\mathrm{MAP}}$ is the MAP estimate and $\boldsymbol{\Sigma} = (-\nabla_{\boldsymbol{\theta}}^2 \log p(\mathcal{D} \mid \boldsymbol{\theta})|_{\boldsymbol{\theta}=\boldsymbol{\theta}_{\mathrm{MAP}}} + \lambda\boldsymbol{I})^{-1}$ is the Hessian of the negative log joint evaluated at $\boldsymbol{\theta}_{\mathrm{MAP}}$. A detailed derivation is given in App. B.3.

## 3.2    BAYESVLM: POST-HOC PROBABILISTIC VLMS

To estimate predictive uncertainties in a post-hoc fashion for VLMs, we independently estimate the posterior of the image projection $\boldsymbol{P}$ and text projection $\boldsymbol{Q}$. For CLIP, we reformulate the contrastive loss to obtain tractable likelihoods for $\boldsymbol{P}$ and $\boldsymbol{Q}$, enabling separate posterior inference. We then approximate the Hessian of the log-likelihood and show how the resulting posteriors induce a distribution over cosine similarities. Finally, we derive a Gaussian approximation of this distribution for efficient downstream inference. Our BayesVLM pipeline is illustrated in Fig. 3.

**Estimate posterior: Likelihood approximation**    The first step in formulating our Bayesian model, BayesVLM, is to define its likelihood function. When doing so, we encounter the following key challenges: popular loss functions for VLMs, such as the InfoNCE loss (Eq. (1)), entangle modalities and data points. While this is a desirable behaviour when learning multi-modal models, it breaks the usual i.i.d. assumption made in Bayesian models. Specifically, we have that

$$(\boldsymbol{x}_i^{\mathrm{IMG}}, \boldsymbol{x}_i^{\mathrm{TXT}}) \overset{\text{non-i.i.d.}}{\sim} p(\boldsymbol{x}_i^{\mathrm{IMG}}, \boldsymbol{x}_i^{\mathrm{TXT}} \mid \boldsymbol{X}_{\backslash i}^{\mathrm{IMG}}, \boldsymbol{X}_{\backslash i}^{\mathrm{TXT}}, \boldsymbol{\theta}), \tag{3}$$

which hinders straightforward application of the Bayesian framework, as data is only conditionally independent. For that purpose, we are instead assuming two independent probabilistic models, one for each modality, with likelihood functions corresponding to the conditional probability for each

modality rather than their joint, *i.e.*,

$$\boldsymbol{x}_i^{\text{IMG}} \overset{\text{i.i.d.}}{\sim} p(\boldsymbol{x}_i^{\text{IMG}} \mid \boldsymbol{X}^{\text{TXT}}, \boldsymbol{\theta}), \qquad \boldsymbol{x}_i^{\text{TXT}} \overset{\text{i.i.d.}}{\sim} p(\boldsymbol{x}_i^{\text{TXT}} \mid \boldsymbol{X}^{\text{IMG}}, \boldsymbol{\theta}). \qquad \text{(i.i.d. assumption)}$$

Consequently, in case of the InfoNCE loss, each likelihood function is given by its respective modality-specific sub-loss term, *i.e.*, in case of the probabilistic model for the image modality, we have $\mathcal{L}_{\text{CE}}(\boldsymbol{X}^{\text{IMG}}, \boldsymbol{X}^{\text{TXT}})$, and corresponds to a categorical distribution. A similar approximation is also applied to SigLIP. Crucially, defining independent probabilistic models for each modality additionally necessitates independence between the encoders. For example, when treating the projection layers $\boldsymbol{P}$ and $\boldsymbol{Q}$ probabilistically, we obtain that:

$$\boldsymbol{P} \perp\!\!\!\perp \boldsymbol{Q}. \qquad \text{(Consequence of i.i.d. assumption)}$$

Following the i.i.d. assumption, the probabilistic model for the image modality is

$$\boldsymbol{x}_i^{\text{IMG}} \quad \xrightarrow[\text{image projection layer } \boldsymbol{P}]{\text{image encoder } \phi(\cdot) \text{ and}} \quad \hat{\boldsymbol{g}}_i = \frac{\boldsymbol{P}\phi(\boldsymbol{x}_i^{\text{IMG}})}{\|\boldsymbol{P}\phi(\boldsymbol{x}_i^{\text{IMG}})\|} \quad \xrightarrow[\text{compute logits}]{\textbf{given} \text{ text embeddings } \hat{\boldsymbol{H}}} \quad \hat{\boldsymbol{H}}\hat{\boldsymbol{g}}_i,$$

and the likelihood becomes a categorical distribution (see App. C.2.1 for formulation)

$$\log p(\boldsymbol{X}^{\text{IMG}} \mid \boldsymbol{X}^{\text{TXT}}, \boldsymbol{\theta}) = \log \textstyle\prod_{i=1}^{n} p(\boldsymbol{x}_i^{\text{IMG}} \mid \boldsymbol{X}^{\text{TXT}}, \boldsymbol{\theta}) = \log \textstyle\prod_{i=1}^{n} [\text{softmax}(\hat{\boldsymbol{H}}\hat{\boldsymbol{g}}_i)]_i. \qquad (4)$$

The probabilistic model for text input can be obtained similarly. We can now apply the LA to this probabilistic model to estimate the approximate posterior.

*Why is this still a reasonable approximation?* For VLMs, it is important to capture interactions between modalities, and assuming independence seems problematic at first. However, as we are using a local post-hoc posterior estimation through the LA, we are effectively introducing an independence conditionally on the MAP estimate of the (joint) contrastive loss. Thus, crucially, even though we assume independence between modalities, we can still capture interactions between modalities. Note that this assumption is also important for computational reasons, as it helps us derive a computationally efficient approach. Our empirical assessment of the Hessian block structure, as discussed in App. F.13, shows that cross-modal curvature terms are moderate in magnitude, indicating low to moderate cross-modal dependencies. A detailed discussion is given in Apps. C.1 and C.2.1.

**Estimate posterior: Hessian approximation** Computing the full Hessian of the negative log-likelihood for the posterior covariance in the Laplace approximation is infeasible, as its size scales quadratically with the number of model parameters, making both its estimation and subsequent predictions computationally prohibitive. We, therefore, adopt the Generalised Gauss–Newton (GGN) approximation (Schraudolph, 2002), which requires the Jacobian of the outputs with respect to the parameters. For linear projection layers, this Jacobian can be derived in closed form. For the image and text encoders, however, the parameter count is prohibitively large, so we treat them as deterministic and approximate the posterior only over the projection matrices $\boldsymbol{P}$ and $\boldsymbol{Q}$.

To further reduce computational and memory costs, we use the Kronecker-factored (KFAC) Generalised Gauss–Newton (GGN) approximation (Ritter et al., 2018; Martens & Grosse, 2015), which expresses the Hessian as a Kronecker product of two smaller matrices. This preserves a richer posterior structure than diagonal approximations. Following (Ritter et al., 2018), the KFAC GGN approximation for the Hessian of $\boldsymbol{P}$ is

$$\underbrace{\left( ^{1}\!/\!\sqrt{n} \textstyle\sum_{i=1}^{n} \phi(\boldsymbol{x}_i^{\text{IMG}})\phi(\boldsymbol{x}_i^{\text{IMG}})^{\top} \right)}_{\boldsymbol{A}_{\text{IMG}}} \otimes \underbrace{\left( ^{1}\!/\!\sqrt{n} \textstyle\sum_{i=1}^{n} \boldsymbol{J}_{\text{IMG}}(\boldsymbol{x}_i^{\text{IMG}})^{\top} \boldsymbol{\Lambda}_{\text{IMG}} \, \boldsymbol{J}_{\text{IMG}}(\boldsymbol{x}_i^{\text{IMG}}) \right)}_{\boldsymbol{B}_{\text{IMG}}}, \qquad (5)$$

where $\boldsymbol{J}_{\text{IMG}}(\boldsymbol{x}_i^{\text{IMG}}) = \partial\hat{\boldsymbol{H}}\frac{\boldsymbol{g}_i}{\|\boldsymbol{g}_i\|}/\partial\boldsymbol{g}_i$ and $\boldsymbol{\Lambda}_{\text{IMG}} = \text{diag}(\boldsymbol{\pi}) - \boldsymbol{\pi}\boldsymbol{\pi}^{\top}$, with $\pi_c = \exp(f_c)/\sum_{c'} \exp(f_{c'})$, $\hat{\boldsymbol{g}}_i^{\top}\hat{\boldsymbol{h}}_c =: f_c$. As estimating the Kronecker factors $\boldsymbol{A}$ and $\boldsymbol{B}$ over the training data set is infeasible, following prior work (Ritter et al., 2018), we leverage a subset of the data and include a pseudo-data count $\tau$ to compensate for the reduced sample size. The posterior covariance over $\boldsymbol{P}$ is approximated as

$$\boldsymbol{\Sigma}_{\text{IMG}} = (\tau(\boldsymbol{A}_{\text{IMG}} \otimes \boldsymbol{B}_{\text{IMG}}) + \lambda\boldsymbol{I})^{-1} \approx \underbrace{\left( \sqrt{\tau}\boldsymbol{A}_{\text{IMG}} + \sqrt{\lambda}\boldsymbol{I} \right)^{-1}}_{\tilde{\boldsymbol{A}}_{\text{IMG}}^{-1}} \otimes \underbrace{\left( \sqrt{\tau}\boldsymbol{B}_{\text{IMG}} + \sqrt{\lambda}\boldsymbol{I} \right)^{-1}}_{\tilde{\boldsymbol{B}}_{\text{IMG}}^{-1}}. \qquad (6)$$

Note that the Kronecker factors $\boldsymbol{A}$ and $\boldsymbol{B}$ can be understood as model statistics under the training data. After having the Gaussian posterior over $\boldsymbol{P}$ and $\boldsymbol{Q}$, as Gaussians are closed under linear transformations, the distribution over $\boldsymbol{g}$ (and $\boldsymbol{h}$) can be obtained analytically:

$$p(\boldsymbol{g} \mid \mathcal{D}) = \mathcal{N}\left(\boldsymbol{P}_{\mathrm{MAP}}\phi(\boldsymbol{x}^{\mathrm{IMG}}), \left(\phi(\boldsymbol{x}^{\mathrm{IMG}})^{\top}\widetilde{\boldsymbol{A}}_{\mathrm{IMG}}^{-1}\phi(\boldsymbol{x}^{\mathrm{IMG}})\right)\widetilde{\boldsymbol{B}}_{\mathrm{IMG}}^{-1}\right). \tag{7}$$

Analogous results hold for the text projection $\boldsymbol{Q}$ and text embedding $\boldsymbol{h}$, which we omit here for brevity. See App. C.2.2 for detailed derivations, and Algorithm 1 outlines the steps described above.

**Make predictions: Cosine similarities approximation**  Given a posterior distribution over the model parameters, evaluating the VLM on an image-text pair yields *random* image and text embeddings rather than deterministic ones, inducing a distribution over their cosine similarity. While the cosine similarity remains well-defined, it becomes a random variable and is generally not Gaussian. The default prediction method, Monte Carlo estimation, requires costly sampling. To improve efficiency, we propose *ProbCosine*, a Gaussian approximation of the cosine similarity distribution based on the first two moments of the image and text embeddings.

Let the Gaussian distributions for the probabilistic image and text embeddings have means $\boldsymbol{\mu_g} = (\mu_{\boldsymbol{g},1}, \ldots, \mu_{\boldsymbol{g},d})$ and $\boldsymbol{\mu_h} = (\mu_{\boldsymbol{h},1}, \ldots, \mu_{\boldsymbol{h},d})$, and diagonal covariances $\boldsymbol{\Sigma_g} = \mathrm{diag}(\sigma_{\boldsymbol{g},1}^2, \ldots, \sigma_{\boldsymbol{g},d}^2)$ and $\boldsymbol{\Sigma_h} = \mathrm{diag}(\sigma_{\boldsymbol{h},1}^2, \ldots, \sigma_{\boldsymbol{h},d}^2)$. Given the cosine similarity $\mathrm{S}_{\cos}(\boldsymbol{x}, \boldsymbol{y}) = \boldsymbol{x}^{\top}\boldsymbol{y}/\|\boldsymbol{x}\|\|\boldsymbol{y}\|$ between two vectors, the expected cosine similarity under the distribution of $\boldsymbol{g}$ and $\boldsymbol{h}$ is approximately:

$$\mathbb{E}[\mathrm{S}_{\cos}(\boldsymbol{g}, \boldsymbol{h})] \approx \frac{\sum_i^d \mu_{\boldsymbol{g},i}\mu_{\boldsymbol{h},i}}{\sqrt{\sum_i \mu_{\boldsymbol{g},i}^2 + \sigma_{\boldsymbol{g},i}^2}\sqrt{\sum_i \mu_{\boldsymbol{h},i}^2 + \sigma_{\boldsymbol{h},i}^2}}, \tag{8}$$

where we use the fact that $\mathbb{E}[x^2] = \mu_x^2 + \sigma_x^2$ and $\mathbb{E}[\|\boldsymbol{x}\|] \leq \sqrt{\sum_i \mu_{\boldsymbol{x},i}^2 + \sigma_{\boldsymbol{x},i}^2}$ by applying the triangle inequality. We can obtain the second moment (variance) $\mathbb{V}\mathrm{ar}[\mathrm{S}_{\cos}(\boldsymbol{g}, \boldsymbol{h})]$ similarly, which is given as:

$$\mathbb{V}\mathrm{ar}[\mathrm{S}_{\cos}(\boldsymbol{g}, \boldsymbol{h})] = \frac{\sum_i \sigma_{\boldsymbol{g},i}^2(\sigma_{\boldsymbol{h},i}^2 + \mu_{\boldsymbol{h},i}^2) + \sigma_{\boldsymbol{h},i}^2\mu_{\boldsymbol{g},i}^2}{\sum_i \mu_{\boldsymbol{g},i}^2 + \sigma_{\boldsymbol{g},i}^2 \sum_i \mu_{\boldsymbol{h},i}^2 + \sigma_{\boldsymbol{h},i}^2}. \tag{9}$$

Henceforth, the local Gaussian approximation to the distribution over cosine similarities is:

$$p(\mathrm{S}_{\cos}(\boldsymbol{g}, \boldsymbol{h})) \approx \mathcal{N}\left(\mathbb{E}[\mathrm{S}_{\cos}(\boldsymbol{g}, \boldsymbol{h})], \mathbb{V}\mathrm{ar}[\mathrm{S}_{\cos}(\boldsymbol{g}, \boldsymbol{h})]\right). \tag{10}$$

Finally, the predictive distribution $p(y \mid \boldsymbol{x})$, *e.g.*, in a zero-shot classification setting, is calculated with the probit approximation (Ghosal et al., 2022; Gibbs, 1998). Hence, our approach allows for the direct propagation of model uncertainties to the class conditional. As shown in Fig. 10 (App. F), compared to ground truth, our approximation qualitatively results in a low approximation error. A detailed derivation can be found in App. C.3, and Algorithm 2 outlines the steps described above.

## 3.3  APPLICATION: PROBABILISTIC ACTIVE FEW-SHOT LEARNING

Active learning (Ren et al., 2021; Settles, 2009) naturally evaluates uncertainty quality by selecting informative samples via predictive uncertainties. We assess BayesVLM with Bayesian acquisition functions and adaptive target-region selection. Given unseen test data $\mathcal{X}_{\mathrm{test}} = \{\boldsymbol{x}_i^{\star}\}_{i=1}^{n_{\mathrm{test}}}$ with unknown labels, the goal is to choose a labeled subset $\{(\boldsymbol{x}_j, y_j)\}_{j=1}^{m}$ with $\boldsymbol{x}_j, y_j \sim p(\boldsymbol{x}, y)$ that best reduces label uncertainty on $\mathcal{X}_{\mathrm{test}}$. We first bias selection toward the query-set predictive distribution, then rank support candidates by influence or informativeness.

**Target region selection**  Following Margatina et al. (2021); Hübotter et al. (2025), we first apply $k$-NN in feature space to pre-select support candidates near the test data, focusing on training points likely useful for the downstream task and reducing acquisition-function cost. Because features are stochastic, we compute either the expected cosine similarity (Eq. (8)) or the 2-Wasserstein distance between image-feature distributions. Details of the calculations are given in App. D.1.

**Acquisition functions**  We consider the BALD (Gal et al., 2017) and EPIG (Bickford Smith et al., 2023) scores as acquisition functions and assess their viability on downstream tasks. Both acquisition functions can utilise model uncertainties estimated by the LA, but they differ conceptually in terms of which uncertainties are targeted. See App. D.2 for details.

**Online Laplace approximation**  We maintain a Laplace posterior over the image-projection matrix $\boldsymbol{P}$ and update it online by *(i)* a gradient step on $\boldsymbol{P}$ and *(ii)* updating the Kronecker factors. The prior precision can optionally be re-estimated after each step (Lin et al., 2023); see App. D.3.

Table 1: **Does BayesVLM provide useful uncertainty estimates in zero-shot settings? Yes.** With the OpenCLIP ViT-B-32 model, our BayesVLM performs on par with CLIP and temp. scaling on ACC (%) and NLPD, while being better calibrated according to the ECE.

| Metrics | Methods | FLOWERS-102 | FOOD-101 | CIFAR-10 | CIFAR-100 | IMAGENET-R | UCF101 | SUN397 |
|---|---|---|---|---|---|---|---|---|
| ACC ↑ | CLIP (Radford et al., 2021) | **68.99**±0.5899 | 80.21±0.2507 | **93.61**±0.2446 | **73.76**±0.4399 | 74.52±0.5032 | 59.82±0.7971 | 67.18±0.3333 |
| | CLIP (temp. scaling) | **68.99**±0.5899 | 80.21±0.2507 | **93.61**±0.2446 | **73.76**±0.4399 | 74.52±0.5032 | 59.82±0.7971 | 67.18±0.3333 |
| | TTA (Farina et al., 2024) | 68.87±0.5905 | **81.68**±0.2435 | 88.54±0.3185 | 65.64±0.4749 | **78.29**±0.4760 | **63.07**±0.7847 | **68.58**±0.3295 |
| | BayesVLM | 68.87±0.4630 | 80.43±0.3968 | **93.62**±0.2444 | 73.63±0.4406 | 74.45±0.4361 | 61.43±0.4868 | 66.96±0.4703 |
| NLPD ↓ | CLIP (Radford et al., 2021) | 1.90±0.0486 | 0.70±0.0094 | 0.21±0.0079 | 0.97±0.0173 | 1.07±0.0237 | 1.59±0.0366 | 1.16±0.0131 |
| | CLIP (temp. scaling) | **1.67**±0.0373 | 0.69±0.0073 | 0.21±0.0061 | **0.94**±0.0138 | 1.04±0.0191 | 1.46±0.0282 | **1.11**±0.0100 |
| | TTA (Farina et al., 2024) | 1.86±0.0475 | **0.67**±0.0094 | 0.35±0.0092 | 1.26±0.0178 | **0.90**±0.0210 | 1.50±0.0363 | 1.14±0.0131 |
| | BayesVLM | 1.73±0.0320 | **0.68**±0.0126 | **0.20**±0.0067 | 0.95±0.0152 | 1.03±0.0177 | **1.44**±0.0183 | 1.12±0.0155 |
| ECE ↓ | CLIP (Radford et al., 2021) | 6.59 | 3.91 | 1.45 | 6.31 | 5.20 | 11.52 | 8.71 |
| | CLIP (temp. scaling) | 5.51 | 4.74 | 1.88 | 3.07 | 4.80 | 3.61 | 2.67 |
| | TTA (Farina et al., 2024) | 9.63 | 4.18 | 2.02 | 5.27 | 2.88 | 11.75 | 9.92 |
| | BayesVLM | **4.22** | **1.69** | **0.72** | **1.92** | **1.78** | **3.57** | **2.06** |

Table 2: **Can ProbCosine improve the zero-shot performance of pre-trained probabilistic models? Yes.** Applying ProbCosine (Ours) to PCME++ (Chun, 2024) consistently improves zero-shot performance over its standard prediction (Mean) across classification benchmarks and metrics.

| Metrics | Methods | FLOWERS-102 | FOOD-101 | CIFAR-10 | CIFAR-100 | IMAGENET-R | UCF101 | SUN397 |
|---|---|---|---|---|---|---|---|---|
| ACC ↑ | Mean | **40.59**±0.0063 | 65.47±0.0030 | **75.16**±0.0043 | 42.52±0.0049 | 42.87±0.0057 | 45.97±0.0035 | **28.50**±0.0073 |
| | Ours | 40.43±0.0063 | **65.54**±0.0030 | 75.12±0.0043 | **42.60**±0.0049 | **42.83**±0.0057 | **46.00**±0.0035 | **28.50**±0.0073 |
| NLPD ↓ | Mean | 3.22±0.0471 | 1.30±0.0125 | 0.77±0.0132 | 2.28±0.0216 | 2.77±0.0346 | 2.18±0.0169 | 3.83±0.0550 |
| | Ours | **3.04**±0.0407 | **1.25**±0.0109 | **0.75**±0.0117 | **2.21**±0.0193 | **2.59**±0.0301 | **2.09**±0.0146 | **3.50**±0.0472 |
| ECE ↓ | Mean | 8.81 | 6.78 | 4.79 | 10.78 | 17.38 | 12.62 | 26.03 |
| | Ours | **2.79** | **1.54** | **2.02** | **4.89** | **10.82** | **5.61** | **19.41** |

## 4  EXPERIMENTS

We outline our setup and address three questions: *(i)* Uncertainty quantification: Does BayesVLM provide reliable uncertainty estimates? *(ii)* Active learning: Can we select informative data for fine-tuning using BayesVLM uncertainty estimates? *(iii)* Efficiency and robustness: Does BayesVLM introduce overhead during inference, does it work in closed-source data settings, and how sensitive is its performance to key hyperparameters? Further setup details and additional results appear in Apps. E and F.

**Data sets** We evaluate zero-shot classification on FLOWERS-102 (Nilsback & Zisserman, 2008), FOOD-101 (Bossard et al., 2014), CIFAR-10/100 (Krizhevsky & Hinton, 2009), IMAGENET-R (Hendrycks et al., 2021), UCF101 (Soomro et al., 2012), and SUN397 (Xiao et al., 2010). For active learning, we form a cross-domain setup with test data from a single domain and a training pool spanning all domains, using OfficeHome (Venkateswara et al., 2017) (Art, Clipart, Product) and an ImageNet variant with ImageNet-R and ImageNet-Sketch (Wang et al., 2019).

**Network architectures** In the zero-shot experiments, we used the OpenCLIP (Ilharco et al., 2021) ViT-B-32 and ViT-L-14, and the SigLIP-B-16 model (Zhai et al., 2023). In the active learning experiments, we use either CLIP-Huge and SigLIP-Base and fine-tune their projection layers.

**Zero-shot baselines** We compare with vanilla CLIP/SigLIP, CLIP/SigLIP with temperature scaling (Guo et al., 2017; Nixon et al., 2019), and test-time augmentation (TTA) (Farina et al., 2024). Temperature scaling uses the parameter minimising negative log predictive density (NLPD) (Quinonero-Candela et al., 2005) on the ImageNet validation set (Deng et al., 2009). We also show ProbCosine can pair with probabilistic VLMs trained from scratch, *e.g.*, ProLIP (Chun et al., 2025) and PCME++ (Chun, 2024). Our focus is on training-free uncertainty estimation, not methods requiring extra adaptation (Upadhyay et al., 2023; Zhou et al., 2025).

**Acquisition functions** For active learning, we incorporate the uncertainties from BayesVLM into the acquisition functions BALD and EPIG and compare against random and entropy-based selection. Both BALD and EPIG use target region selection with nearest neighbour (NN), which selects a test sample based on the uncertainty score, and then selects its 1-NN of the labelled training samples. We also combine the random and entropy baselines with this targeted selection strategy.

**Hyperparameter settings** We estimated the Hessian with 327k image-text pairs (10 CLIP mini-batches) from LAION-400M (Schuhmann et al., 2022), and used the same estimate across all experiments. The pseudo-data count $\tau$ was selected via grid search to minimise NLPD on the ImageNet

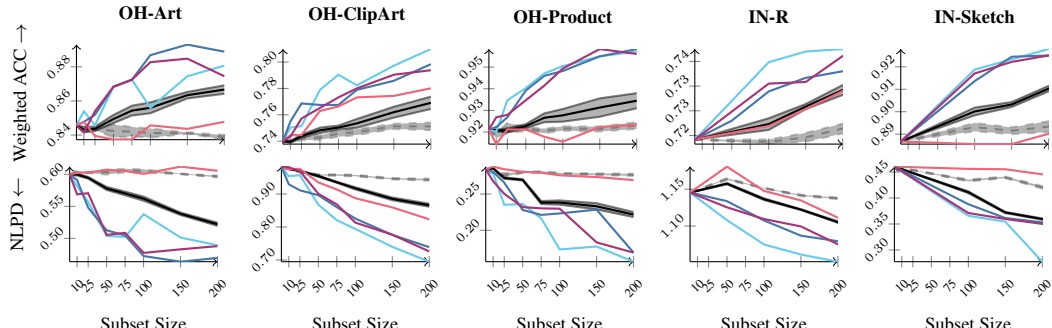

Figure 4: **Can we select informative data for fine-tuning using BayesVLM uncertainty estimates? Yes.** On the OfficeHome data set (OH) and ImageNet variants (IN), when using uncertainty-based scores (EPIG (——) and BALD (——)) to select the fine-tuning data, we achieve better performance compared with Entropy (targeted) (——), Entropy (——), Random selection (targeted) (——), and Random selection (- - -). Thus, highlighting the benefits of using uncertainties from BayesVLM.

validation set, and the prior precision $\lambda$ was set by maximising the marginal likelihood (App. C.2). The same hyperparameters were used for both zero-shot and active learning experiments.

**Evaluation metrics**  For the zero-shot experiments, we report the mean and standard error of accuracy (ACC), NLPD (Quinonero-Candela et al., 2005), and the expected calibration error (ECE) (Guo et al., 2017) computed over the test set. We use a paired $t$-test with $p = 0.05$ to bold results with a significant statistical difference. Active-learning results use class-weighted accuracy and NLPD.

## 4.1 Uncertainty Quantification: Does BayesVLM provide reliable estimates?

We first test the uncertainty estimates of BayesVLM in the zero-shot setting. In Table 1, we report the zero-shot performance of the CLIP-base model using our post-hoc BayesVLM approach, alongside baseline methods, with a focus on predictive quality and uncertainty calibration. Results for CLIP-Large are provided in Table 9 (App. F.7). We observe that BayesVLM achieves similar ACC but lower NLPD than the deterministic CLIP across all data sets, showing that BayesVLM is less overconfident when predicting the incorrect class. BayesVLM performs similarly to temp. scaling on ACC and NLPD, but outperforms all baselines on the ECE. Although TTA achieves higher ACC on some benchmarks, BayesVLM is significantly better calibrated, which results in more useful uncertainty estimates. We conclude that BayesVLM improves model calibration and uncertainty estimation without compromising performance, indicating the effectiveness of our post-hoc strategy.

To test ProbCosine (Sec. 3.2), we applied it to probabilistic embeddings from the pre-trained VLMs PCME++ (Chun, 2024) and ProLIP (Chun et al., 2025) (see Table 11). Zero-shot results for PCME++ (Table 2) show that PCME++ combined with ProbCosine keeps accuracy while consistently improving calibration, indicating ProbCosine can improve any VLM with Gaussian embeddings. Similarly, ProbCosine improves calibration in most cases when combined with ProLIP (Table 11).

## 4.2 Active Learning: Can we select informative data using BayesVLM?

To further assess the utility of BayesVLM's uncertainty estimates, we evaluate it in the active learning setting. We consider a cross-domain setting where the unlabelled target data is from a single domain while the labelled training samples are from multiple domains. The goal is to select the most informative samples from the diverse pool for adapting to the target domain, given a maximum budget (subset size) of support set samples. We experiment with the OfficeHome (Venkateswara et al., 2017) (OH) dataset, where the domains are {Art, Clipart, Product}, and an ImageNet-variant (IN) with domains {R, Sketch}. We incorporated the BayesVLM uncertainties into BALD and EPIG and compared against random and entropy-based selection, either from *(i)* the full training pool (Random, Entropy) or with *(ii)* selection from the test set followed by a 1-NN selection (targeted).

As shown in Fig. 4, both EPIG and BALD, with BayesVLM uncertainties for data selection, outperform Random and Entropy across various subset sizes and target domains. On OH-Product and IN-Sketch, EPIG and BALD obtain similar weighted ACC as Entropy-targeted. However, EPIG

consistently achieves lower NLPD than Entropy-targeted, which shows that the finetuned model is less overconfident on incorrect predictions when trained with samples selected using BayesVLM. Similar conclusions can be observed for CLIP-Huge and SigLIP-Base (Figs. 13 and 14 in App. F).

In Fig. 2, we show the change in the predictive error $(1 - p(y = y^* \mid x))$ and the predictive uncertainty (entropy) for BayesVLM before (zero-shot) and after active learning on EuroSAT (Helber et al., 2019) using EPIG. We use 200 support points and compare against CLIP with entropy selection. BayesVLM reduces overconfident predictions in the zero-shot setting (samples move (b) → (a)), and more effectively adapts to the new data set based on the support set (samples move to (c)).

### 4.3 EFFICIENCY AND ROBUSTNESS: HOW EFFICIENT AND ROBUST IS BAYESVLM?

Following the protocol for zero-shot experiments, we assessed the performance of BayesVLM when estimating the Hessian in settings where the training data is not available, an increasingly common setting for modern machine learning models. In particular, we estimated the Hessian of BayesVLM using the CC12M as a proxy dataset for CLIP models and used the LAION-400M dataset as a proxy for Google's SigLIP model. We find that BayesVLM provides robust uncertainty estimates for CLIP even when estimated on the proxy dataset, *cf.*, Table 3, and it remains stable under mild distribution shifts in the proxy dataset (see App. F.11). Moreover, BayesVLM provides competitive results for SigLIP, a VLM model trained on proprietary data (Table 10 in App. F.7), is robust w.r.t. the pseudo-data count $\tau$ (App. F.8), provides interpretable uncertainties under corruptions (Fig. 5), and and maintains calibration under substantial distribution shift (App. F.12).

Table 3: **Does BayesVLM work in closed-source data settings? Yes.** With OpenCLIP ViT-B-32 trained on LAION-400M and BayesVLM estimated on the proxy dataset CC12M, we find that results are robust and show only slight degradation; statistically significant differences are **bold** ($p = 0.05$).

| Metrics | Dataset | FLOWERS-102 | FOOD-101 | CIFAR-10 | CIFAR-100 | IMAGENET-R | UCF101 | SUN397 |
|---|---|---|---|---|---|---|---|---|
| ACC ↑ | LAION-400M | **68.87**±0.4630 | 80.43±0.3968 | 93.62±0.2444 | 73.63±0.4406 | 74.45±0.4361 | 61.43±0.4868 | 66.96±0.4703 |
| | CC12M | 68.12±0.4660 | 80.35±0.3974 | 93.57±0.2453 | 73.78±0.4398 | 74.32±0.4369 | 61.46±0.4867 | 66.81±0.4709 |
| NLPD ↓ | LAION-400M | **1.73**±0.0320 | 0.68±0.0126 | 0.20±0.0067 | 0.95±0.0152 | **1.03**±0.0177 | 1.44±0.0183 | **1.12**±0.0155 |
| | CC12M | 1.77±0.0330 | 0.68±0.0129 | 0.20±0.0067 | 0.95±0.0152 | 1.03±0.0180 | 1.44±0.0185 | 1.13±0.0162 |
| ECE ↓ | LAION-400M | 4.22 | 1.69 | 0.72 | 1.92 | 1.78 | **3.77** | **2.06** |
| | CC12M | **3.84** | **0.99** | **0.70** | **1.43** | **1.39** | 3.83 | 3.89 |

**Computational overhead** Compared to the deterministic CLIP, BayesVLM adds under 5% runtime for CLIP-base and less than 1% for huge models (Table 7 in App. F.5). Inference cost rises only $0.11\%$ GFLOPs for CLIP-base, whereas TTA needs an $80\times$ increase, see Table 8 in App. F.5.

**Probabilistic cosine similarities** We qualitatively assessed the distribution obtained by Prob-Cosine on a randomly selected test example from the OfficeHome clipart domain, evaluating the mean and variance of the cosine similarity under increasing corruption in both image and text domains. Text corruption was introduced by randomly replacing characters with 'x', and image corruption by randomly adding grey squares. Fig. 5 shows the mean and variance of cosine similarities as corruption increases. We observe that the expected cosine similarity generally decreases and variance increases with more corruption, indicating that our approximation effectively captures model uncertainties under distribution shift. Note that we observe a slight increase in the cosine similarity after one character has been replaced,

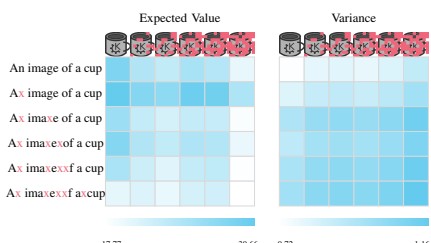

Figure 5: Illustration of ProbCosine under increasing corruption. The mean similarity decreases and variance increases with higher levels of corruption, demonstrating effective uncertainty estimation under distribution shift.

indicating that performing predictions solely on the expected cosine similarity can be problematic. In this case, the variance over cosine similarities can capture the change in the input, highlighting the importance of capturing and propagating the model uncertainties.

**Number of data points for Hessian estimation** We evaluated how the number of samples affects Hessian estimation by computing the trace over 10 random subsets of LAION-400M. As shown in

Fig. 11 (App. F.4), the traces for both image and text projections quickly converge with low variance, suggesting that 10 mini-batches are sufficient for a stable estimate.

**Number of negative samples**  We vary the batch size $K \in \{32768, 8192, 2048\}$ and estimate the posterior from 1–5 random batches, reporting mean±std over five trials. Since the posterior depends on negative samples only via the Hessian $\boldsymbol{B}$ (*cf.* Eq. (5)), we show the relative trace $\mathrm{tr}(\boldsymbol{B}_{i \times K}) / \mathrm{tr}(\boldsymbol{B}_{5 \times K})$, which is expected to be one. As observed in Figure 6 and Fig. 16 (App. F.9), a base batch size of 32768 stays near 1 across all batches with minimal variance, indicating stable estimates of the Hessian.

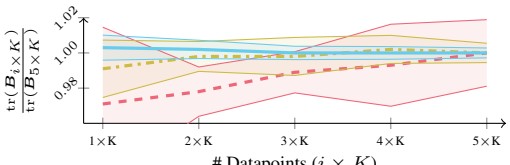

Figure 6: Relative trace of the image Hessian $B$-factor for varying base batch sizes $K$ (2048 (- - -), 8192 (-·-·-), 32768 (——)) and 1–5 random batches. Error bars show $\pm 1$ std over five trials.

## 5 DISCUSSION & CONCLUSION

In this work, we introduced a novel approach for post-hoc uncertainty estimation and propagation for large-scale vision language models (VLMs) such as CLIP (Radford et al., 2021) and SigLIP (Zhai et al., 2023). For this, we first formulated probabilistic models admissible to a Bayesian treatment and then utilised a post-hoc posterior approximation over the last layer of each encoder. Moreover, we derived an analytic approximation of the distribution over cosine similarities for efficient uncertainty propagation. Thus, our approach allows efficient and effective uncertainty quantification without any architectural changes or additional training. We demonstrated the effectiveness of BayesVLM in zero-shot and active learning settings, showing improvements over baselines, and additionally assessed its robustness and efficiency, showing that BayesVLM is a valuable tool for reliable application of VLMs.

Beyond the settings considered in this work, BayesVLM is also applicable to several related problems. An interesting application is the detection of OOD or failure modes. Having access to uncertainties over the image and text embeddings directly enables such scenarios. For example, credible intervals obtained from our Bayesian treatment provide a principled signal for detecting distribution shift or unreliable predictions. Another promising direction is uncertainty-aware retrieval. Since retrieval methods rely on similarity scores in a shared embedding space, uncertainties over the embedding projections can be incorporated into the retrieval process. This allows retrieval systems to more reliably detect cases where inputs fall outside the model's training distribution.

**Limitations**  The limitations of our approach are *(i)* we need access to training data to estimate the Hessian, *(ii)* we require that embeddings are Gaussian distributed, *(iii)* our method only utilises Bayesian projection layers, and *(iv)* we assume independence between image and text projection parameters in the local curvature approximation. Because training data for many VLMs are closed-source, we also assessed potential performance degradation when estimating the Hessian on proxy datasets and found that BayesVLM yields robust estimates. However, further research is needed in closed-source settings.

REPRODUCIBILITY STATEMENT

To ensure the reproducibility of our work, we have provided detailed information on our method and experimental setups. We will discuss the respective details below.

**BayesVLM method & algorithms** In addition to the details presented in the main text (Sec. 3), we provided detailed derivations in App. C of *(i)* the likelihood function approximation in App. C.1, *(ii)* the Laplace approximation used in our method in App. C.2, and *(iii)* the distribution over cosine similarities in App. C.3. Moreover, we provided algorithmic descriptions of our method in Algorithm 1 and Algorithm 2, outlining the precomputation of BayesVLM and the forward inference. Lastly, we presented detailed descriptions of the active learning algorithm used in our work in App. D and provided specific details on *(i)* the targeted selection algorithm in App. D.1, *(ii)* the acquisition functions used in this work in App. D.2, and *(iii)* the online Laplace updates in App. D.3.

**Experiments** In addition to the details provided in the main text in Sec. 4, we provided extensive additional information in App. E. Specifically, we *(i)* detail information on the pre-trained models used in this work in App. E.1, *(ii)* present detailed information on the Hessian estimation and respective hyper-parameters in App. E.2, and in App. E.3, and *(iii)* present details on the hyperparameters and setup of the active learning experiments in App. E.4. We also presented additional experiments and experimental results that extend beyond those presented in the main text in App. F.

**Implementation** The code for the experiments is available at: `https://aaltoml.github.io/BayesVLM/`. Models and precomputed Hessian estimates can be accessed at: `https://huggingface.co/collections/aalto-ml/bayesvlm`.

ACKNOWLEDGEMENTS

AS, RL, and SM acknowledge funding from the Research Council of Finland (grant number 339730 and 362408). MT acknowledges funding from the Research Council of Finland (grant number 347279) and support from the Wallenberg AI, Autonomous Systems, and Software Program (WASP), funded by the Knut and Alice Wallenberg Foundation. MK and SM acknowledge funding from the Finnish Center for Artificial Intelligence (FCAI). AB, SK, and ZA acknowledge partial funding by the ERC (853489 - DEXIM) and the Alfried Krupp von Bohlen und Halbach Foundation. SK thanks the International Max Planck Research School for Intelligent Systems (IMPRS-IS). We acknowledge CSC – IT Center for Science, Finland, for awarding this project access to the LUMI supercomputer, owned by the EuroHPC Joint Undertaking, hosted by CSC (Finland) and the LUMI consortium through CSC. We acknowledge the computational resources provided by the Aalto Science-IT project. Finally, we thank Riccardo Mereu, Jonas Hübotter, and Omar Eldeeb for providing feedback on the manuscript.

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

## APPENDIX

The appendix is organized as follows: App. A summarizes the notation used in the paper. App. B reviews background on vision–language models and the Laplace approximation. App. C presents the derivation of the posterior estimation and the efficient computation of distributions over cosine similarity. App. D outlines the active learning setup. App. E details the experimental setup, while additional results are provided in App. F.

**Use of Large Language Models**  In this paper, LLMs were used only for minor grammatical edits, word polishing, or rephrasing. They did not contribute to research ideation, experiments, or core writing. All suggestions from LLMs were manually verified and edited by the authors prior to final inclusion.

## A  NOTATION

We will briefly summarise the notation used throughout the paper. See Table 4 for the modality-specific notation used and Table 5 for an overview of the notation of general operands and operators.

Table 4: Summary of modality-specific notation.

| Description | Image | Text |
|---|:---:|:---:|
| Input | $x^{\text{IMG}}$ | $x^{\text{TXT}}$ |
| Encoder | $\phi(\cdot)$ | $\psi(\cdot)$ |
| Projection matrix | $P$ | $Q$ |
| Embedding | $g$ | $h$ |
| Normalised embedding | $\hat{g}$ | $\hat{h}$ |
| Stacked embeddings | $G$ | $H$ |
| Kronecker factors | $A_{\text{IMG}}, B_{\text{IMG}}$ | $A_{\text{TXT}}, B_{\text{TXT}}$ |
| Covariance matrix | $\Sigma_{\text{IMG}}$ | $\Sigma_{\text{TXT}}$ |
| Jacobian matrix | $J_{\text{IMG}}$ | $J_{\text{TXT}}$ |

Table 5: Summary of general notation.

| Description | Notation |
|---|:---:|
| Number of data points | $n$ |
| Number of test data points | $n_{\text{test}}$ |
| Number of support set points | $m$ |
| Kronecker product | $\otimes$ |
| Prior precision | $\lambda$ |
| Pseudo-data count | $\tau$ |

## B  BACKGROUND

This section provides additional background information and an extended discussion of related work.

### B.1  EXTENDED RELATED WORK

**Uncertainty in vision-language models**  Many efforts have aimed to learn probabilistic embeddings by making architectural changes to the VLMs and pre-training with a probabilistic loss (Chun, 2024; Chun et al., 2025; 2021; Ji et al., 2023; Li et al., 2022; Neculai et al., 2022). To reduce training costs, several works have proposed enabling uncertainty estimation in pre-trained VLMs via additional training of adapters (Morales-Álvarez et al., 2024; Upadhyay et al., 2023; Lafon

et al., 2025), learning distributions of prompts (Cho et al., 2024; Lu et al., 2022; Yang et al., 2024), model ensembles (Miao et al., 2024), or test-time adaptation (Zhou et al., 2025). These works use a proxy data set different from the pre-training set to learn the predictive uncertainties. Test-time augmentation is a training-free method used for obtaining input-dependent predictive uncertainties by augmenting the test input (Ayhan & Berens, 2018; Farina et al., 2024; Shanmugam et al., 2021), which trades off simplicity against higher inference costs. Other recent training-free approaches focus on zero-shot out-of-distribution detection in CLIP (Fu et al., 2025) or estimating the distribution on the hypersphere as a von-Mises Fisher distribution (Ju et al., 2025). Moreover, calibration of VLMs has been studied for mitigating overconfident predictions (Tu et al., 2023; 2024; Yoon et al., 2024) where temperature scaling is a common post-hoc method for calibrating pre-trained models using a held-out validation set (Galil et al., 2023; Guo et al., 2017). Here, we apply the Laplace approximation to estimate uncertainties directly from the pre-trained VLM without the need for additional training, architectural changes, or training from scratch. Our approach estimates a Bayesian posterior distribution with the pre-training data or a proxy data set before test time and has a similar inference speed to the pre-trained VLM.

**Active learning** In active learning (Ren et al., 2021; Settles, 2009), the model determines through an acquisition function which additional data points are needed to make reliable predictions on a given downstream task. The acquisition function quantifies the informativeness of samples using entropy (Holub et al., 2008; Safaei & Patel, 2025; Wang & Shang, 2014) or diversity-based scores (Ash et al., 2020; Agarwal et al., 2020), coresets (Sener & Savarese, 2018), and parametric models (Sinha et al., 2019; Xie et al., 2023). Here, we focus on acquisition functions utilising model uncertainties from Bayesian active learning (Bickford Smith et al., 2023; Gal et al., 2017; Houlsby et al., 2011). A popular method is the BALD score (Gal et al., 2017; Houlsby et al., 2011), which measures the reduction in epistemic uncertainties of the model. More recently, EPIG was proposed to measure the information gain in the space of predictions rather than parameters (Bickford Smith et al., 2023), building on MacKay's foundational work on information-theoretic experimental design (MacKay, 1992). While such acquisition functions have gained traction in large language models (Hübotter et al., 2025), they remain underexplored in VLMs, where ad-hoc strategies like prompt tuning (Bang et al., 2024) are more prevalent. This work bridges this gap by adapting Bayesian active learning methods to VLMs.

## B.2 LANGUAGE-IMAGE PRE-TRAINING

We consider VLMs trained by minimising the InfoNCE loss (Oord et al., 2018) (*e.g.*, CLIP (Radford et al., 2021)) or the SigLIP loss (Zhai et al., 2023). Specifically, the InfoNCE loss is defined as the sum of two cross-entropy terms, one for each relational direction—image to text ($\mathcal{L}_{\mathrm{CE}}(\boldsymbol{X}^{\mathrm{IMG}}, \boldsymbol{X}^{\mathrm{TXT}})$) and text to image ($\mathcal{L}_{\mathrm{CE}}(\boldsymbol{X}^{\mathrm{TXT}}, \boldsymbol{X}^{\mathrm{IMG}})$). The total loss is defined as follows $\mathcal{L}_{\mathrm{InfoNCE}}(\boldsymbol{X}^{\mathrm{IMG}}, \boldsymbol{X}^{\mathrm{TXT}}) =$

$$\underbrace{-\frac{1}{2n}\sum_{i=1}^{n}\log\frac{\exp(t\hat{\boldsymbol{g}}_i^\top \hat{\boldsymbol{h}}_i)}{\sum_{j=1}^{n}\exp(t\hat{\boldsymbol{g}}_i^\top \hat{\boldsymbol{h}}_j)}}_{\mathrm{IMG}\rightarrow\mathrm{TXT},\ \mathcal{L}_{\mathrm{CE}}(\boldsymbol{X}^{\mathrm{IMG}},\boldsymbol{X}^{\mathrm{TXT}})} \underbrace{-\frac{1}{2n}\sum_{i=1}^{n}\log\frac{\exp(t\hat{\boldsymbol{h}}_i^\top \hat{\boldsymbol{g}}_i)}{\sum_{j=1}^{n}\exp(t\hat{\boldsymbol{h}}_i^\top \hat{\boldsymbol{g}}_j)}}_{\mathrm{IMG}\leftarrow\mathrm{TXT},\ \mathcal{L}_{\mathrm{CE}}(\boldsymbol{X}^{\mathrm{TXT}},\boldsymbol{X}^{\mathrm{IMG}})}, \quad (11)$$

where $t$ is a learnable temperature parameter, $n$ denotes the number of image-text pairs, and $\hat{\boldsymbol{g}}$ and $\hat{\boldsymbol{h}}$ are the unit-length normalised embeddings. This contrastive loss function encourages embeddings for matching image-text pairs to be similar while simultaneously pushing unrelated image-text pairs away from each other (Oord et al., 2018).

Recently, the SigLIP loss (Zhai et al., 2023) has been proposed as an alternative to the InfoNCE loss, aimed at improving numerical stability and training speed. In contrast to InfoNCE, the SigLIP loss uses a binary classification loss over the cosine similarities, *i.e.*, $\mathcal{L}_{\mathrm{SigLIP}}(\boldsymbol{X}^{\mathrm{IMG}}, \boldsymbol{X}^{\mathrm{TXT}}) =$

$$-\frac{1}{n}\sum_{i=1}^{n}\sum_{j=1}^{n}\log\frac{1}{1+\exp(z_{ij}(-t\hat{\boldsymbol{g}}_i^\top \hat{\boldsymbol{h}}_j + b))}, \quad (12)$$

where $z_{ii} = 1$, $z_{ij} = -1$ if $i \neq j$ and $b$ is a learnable bias term. For classification settings, the SigLIP loss does not provide normalised class conditional probabilities $p(y \mid \boldsymbol{x})$ but provides binary classification probabilities. Henceforth, when fine-tuning a SigLIP pre-trained VLM for classification tasks, one typically uses the cross-entropy loss instead.

### B.3 Laplace approximation

Given a data set $\mathcal{D} = \{(\boldsymbol{x}_i, \boldsymbol{y}_i)\}_{i=1}^n$ and denote the model parameters as $\boldsymbol{\theta}$, in Bayesian deep learning, we aim to estimate the posterior distribution

$$p(\boldsymbol{\theta} \mid \mathcal{D}) = \frac{p(\boldsymbol{\theta}) \prod_{i=1}^n p(\boldsymbol{y}_i \mid \boldsymbol{x}_i, \boldsymbol{\theta})}{\int_{\boldsymbol{\theta}} p(\boldsymbol{\theta}) \prod_{i=1}^n p(\boldsymbol{y}_i \mid \boldsymbol{x}_i, \boldsymbol{\theta}) \mathrm{d}\boldsymbol{\theta}} \tag{13}$$

$$= \frac{\boxed{\text{prior}} \times \boxed{\text{likelihood}}}{\boxed{\text{marginal likelihood}}}.$$

Unfortunately, computing the denominator (marginal likelihood) is generally intractable (not feasible) as it requires integration over a high-dimensional space w.r.t. a potentially non-linear function. A classical approach to circumvent this challenge is to approximate the posterior using a Laplace approximation MacKay (1992), which has recently gained traction in the Bayesian deep learning community Ritter et al. (2018); Daxberger et al. (2021); Li et al. (2025); Meronen et al. (2024); Roy et al. (2022); Scannell et al. (2024).

The Laplace approximation hinges on the idea that the posterior distribution is proportional to the joint, *i.e.*,

$$p(\boldsymbol{\theta} \mid \mathcal{D}) \propto p(\boldsymbol{\theta}, \mathcal{D}) = p(\boldsymbol{\theta}) \prod_{i=1}^n p(\boldsymbol{y}_i \mid \boldsymbol{x}_i, \boldsymbol{\theta}) \tag{14}$$

up to an unknown normalisation constant (the marginal likelihood). Moreover, using a second-order Taylor expansion of the log joint around the maximum-a-posteriori (MAP) estimate $\boldsymbol{\theta}_{\mathrm{MAP}}$ (mode of the function) one obtains the unnormalised log density function of a Gaussian centred at $\boldsymbol{\theta}_{\mathrm{MAP}}$, *i.e.*, $\log p(\boldsymbol{\theta}, \mathcal{D}) \approx$

$$\log p(\boldsymbol{\theta}_{\mathrm{MAP}}, \mathcal{D}) - \frac{1}{2}(\boldsymbol{\theta} - \boldsymbol{\theta}_{\mathrm{MAP}})^\top \boldsymbol{\Sigma}^{-1}(\boldsymbol{\theta} - \boldsymbol{\theta}_{\mathrm{MAP}}), \tag{15}$$

where

$$\boldsymbol{\Sigma} = \left(-\nabla_{\boldsymbol{\theta}}^2 \log p(\boldsymbol{\theta}, \mathcal{D})|_{\boldsymbol{\theta}=\boldsymbol{\theta}_{\mathrm{MAP}}}\right)^{-1} = \left(-\nabla_{\boldsymbol{\theta}}^2 \log p(\mathcal{D} \mid \boldsymbol{\theta})|_{\boldsymbol{\theta}=\boldsymbol{\theta}_{\mathrm{MAP}}} - \nabla_{\boldsymbol{\theta}}^2 \log p(\boldsymbol{\theta})|_{\boldsymbol{\theta}=\boldsymbol{\theta}_{\mathrm{MAP}}}\right)^{-1} \tag{16}$$

is the Hessian matrix of the log joint (prior × likelihood) at $\boldsymbol{\theta}_{\mathrm{MAP}}$. By matching the marginal likelihood in Eq. (2) with the normalisation constant of a Gaussian, we obtain the Laplace approximation:

$$p(\boldsymbol{\theta} \mid \mathcal{D}) \approx \mathcal{N}(\boldsymbol{\theta}_{\mathrm{MAP}}, \boldsymbol{\Sigma}^{-1}), \tag{17}$$

with covariance given by the inverse of the Hessian matrix.

As LA fits a Gaussian distribution to the posterior, centred at the MAP estimate of a *pre-trained* model, it is 'post-hoc'. The *prior* is implicitly defined by the L2 regularisation (weight decay) commonly used during training Radford et al. (2021); Zhai et al. (2023), and corresponds to a diagonal Gaussian prior on the parameters, *i.e.*, $p(\boldsymbol{\theta}) = \mathcal{N}(\boldsymbol{0}, \lambda^{-1}\boldsymbol{I})$. The *likelihood* is defined by the training loss.

## C  Derivations

This section provides detailed derivations of the equations presented in the main text. App. C.1 discusses the setting where the i.i.d. assumption is not made and the challenges associated with it. App. C.2 discusses the i.i.d. assumption, the resulting probabilistic model, and the derivations for estimating the posterior. App. C.3 covers the derivations for efficient prediction, *i.e.*, the distribution over cosine similarities.

### C.1  What Happens without the i.i.d. assumption

In this section, we derive the Laplace approximation when we don't make the i.i.d. assumption. We will show this results in multiple computationally expensive or infeasible terms in the posterior covariance, and the posterior obtained by our i.i.d. assumption keeps the computationally feasible term.

**Algorithm 1** Turn VLM into BayesVLM

1: **Input:** VLM encoders $\{\text{IMG}, \text{TXT}\}$, training data $\mathcal{D}$
2: **for each** encoder $\text{ENC} \in \{\text{IMG}, \text{TXT}\}$ **do**
3:     Compute $\boldsymbol{A}_{\text{ENC}}$ factor with Eq. (48)
4:     Compute $\boldsymbol{B}_{\text{ENC}}$ factor with Eq. (49)
5: **end for**
6: Find $\lambda$ by maximising the marginal likelihood (Eq. (81))
7: *(Optional)* Find optimal $\tau$ or set $\tau = 1$
8: **for each** encoder $\text{ENC} \in \{\text{IMG}, \text{TXT}\}$ **do**
9:     Update $\widetilde{\boldsymbol{A}}_{\text{ENC}} \leftarrow \sqrt{\tau}\boldsymbol{A}_{\text{ENC}} + \sqrt{\lambda}\boldsymbol{I}$
10:     Update $\widetilde{\boldsymbol{B}}_{\text{ENC}} \leftarrow \sqrt{\tau}\boldsymbol{B}_{\text{ENC}} + \sqrt{\lambda}\boldsymbol{I}$
11: **end for**
12: **Return:** $\{(\widetilde{\boldsymbol{A}}_{\text{IMG}}, \widetilde{\boldsymbol{B}}_{\text{IMG}}), (\widetilde{\boldsymbol{A}}_{\text{TXT}}, \widetilde{\boldsymbol{B}}_{\text{TXT}})\}$

**Algorithm 2** Compute Predictions

1: **Input:** BayesVLM, $(x_{\text{IMG}}, x_{\text{TXT}})$
    Compute embeddings using Eq. (7), *i.e.*,
2:     $\mu_{\boldsymbol{g}} \leftarrow \boldsymbol{P}_{\text{MAP}}\, \phi(\boldsymbol{x}_{\text{IMG}})$
3:     $\Sigma_{\boldsymbol{g}} \leftarrow \left(\phi(\boldsymbol{x}^{\text{IMG}})^{\top}\widetilde{\boldsymbol{A}}_{\text{IMG}}^{-1}\phi(\boldsymbol{x}^{\text{IMG}})\right)\widetilde{\boldsymbol{B}}_{\text{IMG}}^{-1}$
4:     $\mu_{\boldsymbol{h}} \leftarrow \boldsymbol{Q}_{\text{MAP}}\, \psi(\boldsymbol{x}_{\text{TXT}})$
5:     $\Sigma_{\boldsymbol{h}} \leftarrow \left(\psi(\boldsymbol{x}^{\text{TXT}})^{\top}\widetilde{\boldsymbol{A}}_{\text{TXT}}^{-1}\psi(\boldsymbol{x}^{\text{TXT}})\right)\widetilde{\boldsymbol{B}}_{\text{TXT}}^{-1}$
    Apply ProbCosine, *i.e.*,
6:     Compute $\mathbb{E}[\mathrm{S}_{\cos}(\boldsymbol{g}, \boldsymbol{h})]$ with Eq. (8)
7:     Compute $\mathbb{V}\mathrm{ar}[\mathrm{S}_{\cos}(\boldsymbol{g}, \boldsymbol{h})]$ with Eq. (9)
    Apply probit approximation (Gibbs, 1998), *i.e.*,
8:     **Return:** $\mathrm{softmax}\left(\frac{t\,\mathbb{E}[\mathrm{S}_{\cos}(\boldsymbol{g},\boldsymbol{h})]}{\sqrt{1+\pi/8*t^2\,\mathbb{V}\mathrm{ar}[\mathrm{S}_{\cos}(\boldsymbol{g},\boldsymbol{h})]}}\right)$

We start by reformulating the InfoNCE loss. Given a dataset with $n$ image-text pairs $(\boldsymbol{x}_i^{\text{IMG}}, \boldsymbol{x}_i^{\text{TXT}})$, the InfoNCE loss is defined as $\mathcal{L}_{\text{InfoNCE}}(\boldsymbol{X}^{\text{IMG}}, \boldsymbol{X}^{\text{TXT}}) =$

$$-\underbrace{\frac{1}{2n}\sum_{i=1}^{n}\log\frac{\exp(t\hat{\boldsymbol{g}}_i^{\top}\hat{\boldsymbol{h}}_i)}{\sum_{j=1}^{n}\exp(t\hat{\boldsymbol{g}}_i^{\top}\hat{\boldsymbol{h}}_j)}}_{\mathcal{L}_{\text{CE}}^{\text{IMG}}(\boldsymbol{X}^{\text{IMG}},\boldsymbol{X}^{\text{TXT}})} - \underbrace{\frac{1}{2n}\sum_{i=1}^{n}\log\frac{\exp(t\hat{\boldsymbol{h}}_i^{\top}\hat{\boldsymbol{g}}_i)}{\sum_{j=1}^{n}\exp(t\hat{\boldsymbol{h}}_i^{\top}\hat{\boldsymbol{g}}_j)}}_{\mathcal{L}_{\text{CE}}^{\text{TXT}}(\boldsymbol{X}^{\text{TXT}},\boldsymbol{X}^{\text{IMG}})}, \tag{18}$$

where $t$ is a learnable temperature parameter, $\hat{\boldsymbol{g}}$ and $\hat{\boldsymbol{h}}$ are the unit-length normalised image and text embeddings. Evaluating this loss in practice is infeasible on billions of data points. Therefore, the common practice adopted in VLMs, such as CLIP, is to evaluate it on a sufficiently large batch. Specifically, denote a batch of image-text pairs as $\mathcal{B} = \{\boldsymbol{X}_{\mathcal{B}}^{\text{IMG}}, \boldsymbol{X}_{\mathcal{B}}^{\text{IMG}}\}$. Then the InfoNCE loss over the whole data set is approximated by:

$$\mathcal{L}_{\text{InfoNCE}}(\boldsymbol{X}^{\text{IMG}}, \boldsymbol{X}^{\text{TXT}}) \approx \sum_{\mathcal{B}}\mathcal{L}_{\text{InfoNCE}}(\boldsymbol{X}_{\mathcal{B}}^{\text{IMG}}, \boldsymbol{X}_{\mathcal{B}}^{\text{TXT}}). \tag{19}$$

For each batch, we can view the InfoNCE loss as two separate classification losses, one over image inputs and the other over text inputs. To avoid clutter, we drop the temperature parameter from now on. Looking at the loss for the image inputs $\mathcal{L}_{\text{CE}}^{\text{IMG}}(\boldsymbol{X}_{\mathcal{B}}^{\text{IMG}}, \boldsymbol{X}_{\mathcal{B}}^{\text{TXT}})$, we can reformulate it as follows:

$$\mathcal{L}_{\text{CE}}^{\text{IMG}}(\boldsymbol{X}_{\mathcal{B}}^{\text{IMG}}, \boldsymbol{X}_{\mathcal{B}}^{\text{TXT}}) = -\frac{1}{2|\mathcal{B}|}\sum_{i=1}^{|\mathcal{B}|}\log\frac{\exp(\hat{\boldsymbol{g}}_i^{\top}\hat{\boldsymbol{h}}_i)}{\sum_{j=1}^{|\mathcal{B}|}\exp(\hat{\boldsymbol{g}}_i^{\top}\hat{\boldsymbol{h}}_j)} \tag{20}$$

$$= -\frac{1}{2|\mathcal{B}|}\sum_{i=1}^{|\mathcal{B}|}\log\left[\mathrm{softmax}\left(\left[\hat{\boldsymbol{g}}_i^{\top}\hat{\boldsymbol{h}}_1, \hat{\boldsymbol{g}}_i^{\top}\hat{\boldsymbol{h}}_2, \ldots, \hat{\boldsymbol{g}}_i^{\top}\hat{\boldsymbol{h}}_{|\mathcal{B}|}\right]\right)\right]_i \tag{21}$$

$$= -\frac{1}{2|\mathcal{B}|}\sum_{i=1}^{|\mathcal{B}|}\log\left[\mathrm{softmax}\left(\hat{\boldsymbol{H}}\hat{\boldsymbol{g}}_i\right)\right]_i, \tag{22}$$

where $[\mathrm{softmax}(\boldsymbol{z})]_i \triangleq \frac{\exp(z_i)}{\sum_j \exp(z_j)}$ is the $i$-th output of softmax function. We can see that the loss is equivalent to the cross-entropy loss on the following model, where label $\boldsymbol{y}_i^{\text{IMG}}$ is a one-hot encoded vector with $i$-th element equal to one,

$$\boldsymbol{x}_i^{\text{IMG}} \xrightarrow[\text{image projection layer } \boldsymbol{P}]{\text{Image encoder } \phi(\cdot) \text{ and}} \hat{\boldsymbol{g}}_i = \frac{\boldsymbol{P}\phi(\boldsymbol{x}_i^{\text{IMG}})}{\|\boldsymbol{P}\phi(\boldsymbol{x}_i^{\text{IMG}})\|} \xrightarrow[\text{to compute logit}]{\text{use text embeddings } \hat{\boldsymbol{H}}} \hat{\boldsymbol{H}}\hat{\boldsymbol{g}}_i.$$

Similarly, the text loss $\mathcal{L}_{\text{CE}}^{\text{TXT}}(\boldsymbol{X}^{\text{TXT}}, \boldsymbol{X}^{\text{IMG}})$ can be viewed as cross-entropy loss on the following model where label $\boldsymbol{y}_i^{\text{TXT}}$ is a one-hot encoded vector with $i$-th element equal to one,

$$\boldsymbol{x}_i^{\text{TXT}} \xrightarrow[\text{text projection layer } \boldsymbol{Q}]{\text{Text encoder } \psi(\cdot) \text{ and}} \hat{\boldsymbol{h}}_i = \frac{\boldsymbol{Q}\psi(\boldsymbol{x}_i^{\text{TXT}})}{\|\boldsymbol{Q}\psi(\boldsymbol{x}_i^{\text{TXT}})\|} \xrightarrow[\text{to compute logit}]{\text{use image embeddings } \hat{\boldsymbol{G}}} \hat{\boldsymbol{G}}\hat{\boldsymbol{h}}_i.$$

Under this view, VLMs trained with the InfoNCE loss can be viewed as using the following equivalent model and loss:

$$f(\boldsymbol{x}_i^{\text{IMG}}, \boldsymbol{x}_i^{\text{TXT}} \mid \boldsymbol{X}_{\backslash i}^{\text{IMG}}, \boldsymbol{X}_{\backslash i}^{\text{TXT}}, \boldsymbol{\theta}) = \left[\hat{\boldsymbol{H}}\hat{\boldsymbol{g}}_i, \hat{\boldsymbol{G}}\hat{\boldsymbol{h}}_i\right], \tag{23}$$

$$\ell_i^{\text{IMG,TXT}} = -\underbrace{\log[\text{softmax}\left(\hat{\boldsymbol{H}}\hat{\boldsymbol{g}}_i\right)]_i}_{\ell_i^{\text{IMG}}} - \underbrace{\log[\text{softmax}\left(\hat{\boldsymbol{G}}\hat{\boldsymbol{h}}_i\right)]_i}_{\ell_i^{\text{TXT}}} \tag{24}$$

$$\mathcal{L}_{\text{CE}}^{\text{IMG}}(\boldsymbol{X}_{|\mathcal{B}|}^{\text{IMG}}, \boldsymbol{X}_{|\mathcal{B}|}^{\text{TXT}}) = \frac{1}{2|\mathcal{B}|} \sum_{i=1}^{|\mathcal{B}|} \ell_i^{\text{IMG,TXT}}. \tag{25}$$

Because data is only conditionally independent in this model, *i.e.*,

$$(\boldsymbol{x}_i^{\text{IMG}}, \boldsymbol{x}_i^{\text{TXT}}) \sim p(\boldsymbol{x}_i^{\text{IMG}}, \boldsymbol{x}_i^{\text{TXT}} \mid \boldsymbol{X}_{\backslash i}^{\text{IMG}}, \boldsymbol{X}_{\backslash i}^{\text{TXT}}, \boldsymbol{\theta}), \tag{26}$$

the usual i.i.d. assumption made in Bayesian models is violated. Note that performing Bayesian inference over non-i.i.d. data in general settings is an active research field (Ralaivola et al., 2009). Nevertheless, we can still consider applying the Laplace approximation in this case. Crucially, note that Laplace approximation is derived through a second-order Taylor approximation of the negative log joint $-\log p(\mathcal{D} \mid \boldsymbol{\theta})p(\boldsymbol{\theta})$, which only requires the negative log joint to be a twice-differentiable function. Therefore, we can still consider the Laplace for local posterior approximation at the MAP estimation. The interpretation of the underlying probabilistic model, however, may be more challenging in those cases.

We will now derive the negative log likelihood Hessian for the image projection layer $\boldsymbol{P}$. Define shorthand $f_{\boldsymbol{P},\boldsymbol{Q}}(\boldsymbol{x}_i) = f(\boldsymbol{x}_i^{\text{IMG}}, \boldsymbol{x}_i^{\text{TXT}} \mid \boldsymbol{X}_{\backslash i}^{\text{IMG}}, \boldsymbol{X}_{\backslash i}^{\text{TXT}}, \boldsymbol{\theta})$, the GGN approximation for the Hessian over image projection layer $\boldsymbol{P}$ is given as

$$\frac{\partial^2 \ell_i^{\text{IMG,TXT}}}{\partial^2 \boldsymbol{P}} \approx \frac{\partial f_{\boldsymbol{P},\boldsymbol{Q}}(\boldsymbol{x}_i)}{\partial \boldsymbol{P}}^{\top} \frac{\partial^2 \ell_i}{\partial^2 f_{\boldsymbol{P},\boldsymbol{Q}}(\boldsymbol{x}_i)} \frac{\partial f_{\boldsymbol{P},\boldsymbol{Q}}(\boldsymbol{x}_i)}{\partial \boldsymbol{P}}, \tag{27}$$

where

$$\frac{\partial f_{\boldsymbol{P},\boldsymbol{Q}}(\boldsymbol{x}_i)}{\partial \boldsymbol{P}}^{\top} = \left[\left(\frac{\partial \hat{\boldsymbol{H}}\hat{\boldsymbol{g}}_i}{\partial \boldsymbol{P}}\right)^{\top} \quad \left(\frac{\partial \hat{\boldsymbol{G}}\hat{\boldsymbol{h}}_i}{\partial \boldsymbol{P}}\right)^{\top}\right], \tag{28}$$

$$\frac{\partial^2 \ell_i^{\text{IMG,TXT}}}{\partial^2 f_{\boldsymbol{P},\boldsymbol{Q}}(\boldsymbol{x}_i)} = \begin{bmatrix} \dfrac{\partial^2 \ell_i^{\text{IMG,TXT}}}{\partial^2 \hat{\boldsymbol{H}}\hat{\boldsymbol{g}}_i} & \dfrac{\partial^2 \ell_i^{\text{IMG,TXT}}}{\partial \hat{\boldsymbol{H}}\hat{\boldsymbol{g}}_i \partial \hat{\boldsymbol{G}}\hat{\boldsymbol{h}}_i} \\[4mm] \dfrac{\partial^2 \ell_i^{\text{IMG,TXT}}}{\partial \hat{\boldsymbol{G}}\hat{\boldsymbol{h}}_i \partial \hat{\boldsymbol{H}}\hat{\boldsymbol{g}}_i} & \dfrac{\partial^2 \ell_i^{\text{IMG,TXT}}}{\partial^2 \hat{\boldsymbol{G}}\hat{\boldsymbol{h}}_i} \end{bmatrix}. \tag{29}$$

When writing out the matrix multiplication, we have:

$$\frac{\partial^2 \ell_i^{\text{IMG,TXT}}}{\partial^2 \boldsymbol{P}} \approx \frac{\partial f_{\boldsymbol{P},\boldsymbol{Q}}(\boldsymbol{x}_i)^\top}{\partial \boldsymbol{P}} \frac{\partial^2 \ell_i}{\partial^2 f_{\boldsymbol{P},\boldsymbol{Q}}(\boldsymbol{x}_i)} \frac{\partial f_{\boldsymbol{P},\boldsymbol{Q}}(\boldsymbol{x}_i)}{\partial \boldsymbol{P}} \tag{30}$$

$$= \underbrace{\left(\frac{\partial \hat{\boldsymbol{H}}\hat{\boldsymbol{g}}_i}{\partial \boldsymbol{P}}\right)^\top}_{\mathbb{R}^{d \times |\mathcal{B}|}} \underbrace{\frac{\partial^2 \ell_i^{\text{IMG}}}{\partial^2 \hat{\boldsymbol{H}}\hat{\boldsymbol{g}}_i}}_{\mathbb{R}^{|\mathcal{B}| \times |\mathcal{B}|}} \frac{\partial \hat{\boldsymbol{H}}\hat{\boldsymbol{g}}_i}{\partial \boldsymbol{P}} \tag{31}$$

$$+ \left(\frac{\partial \hat{\boldsymbol{H}}\hat{\boldsymbol{g}}_i}{\partial \boldsymbol{P}}\right)^\top \frac{\partial^2 \ell_i^{\text{TXT}}}{\partial^2 \hat{\boldsymbol{H}}\hat{\boldsymbol{g}}_i} \frac{\partial \hat{\boldsymbol{H}}\hat{\boldsymbol{g}}_i}{\partial \boldsymbol{P}} + \left(\frac{\partial \hat{\boldsymbol{G}}\hat{\boldsymbol{h}}_i}{\partial \boldsymbol{P}}\right)^\top \frac{\partial^2 \ell_i^{\text{IMG,TXT}}}{\partial \hat{\boldsymbol{G}}\hat{\boldsymbol{h}}_i \partial \hat{\boldsymbol{H}}\hat{\boldsymbol{g}}_i} \frac{\partial \hat{\boldsymbol{H}}\hat{\boldsymbol{g}}_i}{\partial \boldsymbol{P}} \tag{32}$$

$$+ \left(\frac{\partial \hat{\boldsymbol{H}}\hat{\boldsymbol{g}}_i}{\partial \boldsymbol{P}}\right)^\top \frac{\partial^2 \ell_i^{\text{IMG,TXT}}}{\partial \hat{\boldsymbol{H}}\hat{\boldsymbol{g}}_i \partial \hat{\boldsymbol{G}}\hat{\boldsymbol{h}}_i} \frac{\partial \hat{\boldsymbol{G}}\hat{\boldsymbol{h}}_i}{\partial \boldsymbol{P}} + \left(\frac{\partial \hat{\boldsymbol{G}}\hat{\boldsymbol{h}}_i}{\partial \boldsymbol{P}}\right)^\top \frac{\partial^2 \ell_i^{\text{IMG,TXT}}}{\partial^2 \hat{\boldsymbol{G}}\hat{\boldsymbol{h}}_i} \frac{\partial \hat{\boldsymbol{G}}\hat{\boldsymbol{h}}_i}{\partial \boldsymbol{P}} \tag{33}$$

Here only the first term $\left(\frac{\partial \hat{\boldsymbol{H}}\hat{\boldsymbol{g}}_i}{\partial \boldsymbol{P}}\right)^\top \frac{\partial^2 \ell_i^{\text{IMG}}}{\partial^2 \hat{\boldsymbol{H}}\hat{\boldsymbol{g}}_i} \frac{\partial \hat{\boldsymbol{H}}\hat{\boldsymbol{g}}_i}{\partial \boldsymbol{P}}$ can be computed efficiently while terms in red are intractable or computationally expensive. The approximated posterior for $\boldsymbol{P}$ obtained in our BayesVLM corresponds to dropping the computationally expensive or infeasible terms in the exact model.

## C.2 Estimating the posterior for BayesVLM with Laplace approximation

We now introduce the procedure for estimating the posterior of BayesVLM using the Laplace approximation in this section. We start by introducing the i.i.d. assumption we made and the resulting probabilistic model for BayesVLM in App. C.2.1. Then, we give the derivation for the posterior approximation for BayesVLM in App. C.2.2.

### C.2.1 I.I.D. assumption and the resulting probabilistic model

To efficiently estimate the approximated posterior using the Laplace approximation and obtain a clear probabilistic model underlying it, we assume two independent probabilistic models, one for each modality. Specifically, for each modality, we assume data are i.i.d. given the observations from the other modality:

$$\boldsymbol{x}_i^{\text{IMG}} \overset{\text{i.i.d.}}{\sim} p(\boldsymbol{x}_i^{\text{IMG}} \mid \boldsymbol{X}^{\text{TXT}}, \boldsymbol{\theta}), \qquad \boldsymbol{x}_i^{\text{TXT}} \overset{\text{i.i.d.}}{\sim} p(\boldsymbol{x}_i^{\text{TXT}} \mid \boldsymbol{X}^{\text{IMG}}, \boldsymbol{\theta}). \qquad \text{(i.i.d. assumption)}$$

Following this assumption, the image encoder $\phi(\cdot)$ and text encoder $\psi(\cdot)$ will become independent, and image projection layer $\boldsymbol{P}$ and text projection layer $\boldsymbol{Q}$ will become independent as well:

$$\phi(\cdot) \perp\!\!\!\perp \psi(\cdot), \quad \boldsymbol{P} \perp\!\!\!\perp \boldsymbol{Q}. \qquad \text{(Consequence from i.i.d. assumption)}$$

Under these assumptions, we can untangle the interaction between two modalities and approximate their respective likelihoods as categorical distributions.

When the modalities become independent, for image input $\boldsymbol{x}_i^{\text{IMG}}$, we can only look at the image loss defined as

$$\mathcal{L}_{\text{CE}}^{\text{IMG}}(\boldsymbol{X}_{\mathcal{B}}^{\text{IMG}}, \boldsymbol{X}_{\mathcal{B}}^{\text{TXT}}) = -\frac{1}{2|\mathcal{B}|} \sum_{i=1}^{|\mathcal{B}|} \log \frac{\exp(\hat{\boldsymbol{g}}_i^\top \hat{\boldsymbol{h}}_i)}{\sum_{j=1}^{|\mathcal{B}|} \exp(\hat{\boldsymbol{g}}_i^\top \hat{\boldsymbol{h}}_j)} \tag{34}$$

$$= -\frac{1}{2|\mathcal{B}|} \sum_{i=1}^{|\mathcal{B}|} \log \left[\text{softmax}\left(\left[\hat{\boldsymbol{g}}_i^\top \hat{\boldsymbol{h}}_1, \hat{\boldsymbol{g}}_i^\top \hat{\boldsymbol{h}}_2, \ldots, \hat{\boldsymbol{g}}_i^\top \hat{\boldsymbol{h}}_{|\mathcal{B}|}\right]\right)\right]_i \tag{35}$$

$$= -\frac{1}{2|\mathcal{B}|} \sum_{i=1}^{|\mathcal{B}|} \log \left[\text{softmax}\left(\hat{\boldsymbol{H}}\hat{\boldsymbol{g}}_i\right)\right]_i, \tag{36}$$

where $[\text{softmax}(\boldsymbol{z})]_i \triangleq \frac{\exp(z_i)}{\sum_j \exp(z_j)}$ is the $i$-th output of softmax function. This corresponds to the cross-entropy loss on the following model, where label $\boldsymbol{y}_i^{\text{IMG}}$ is a one-hot encoded vector with $i$-th element equal to one,

$$\boldsymbol{x}_i^{\text{IMG}} \xrightarrow[\text{image projection layer } \boldsymbol{P}]{\text{Image encoder } \phi(\cdot) \text{ and}} \hat{\boldsymbol{g}}_i = \frac{\boldsymbol{P}\phi(\boldsymbol{x}_i^{\text{IMG}})}{\|\boldsymbol{P}\phi(\boldsymbol{x}_i^{\text{IMG}})\|} \xrightarrow[\text{compute logit}]{\textbf{given} \text{ text embeddings } \hat{\boldsymbol{H}}} \hat{\boldsymbol{H}}\hat{\boldsymbol{g}}_i.$$

Therefore, for image input, the corresponding model is

$$f(\boldsymbol{x}_i^{\text{IMG}} \mid \boldsymbol{X}^{\text{TXT}}, \boldsymbol{\theta}) = \hat{\boldsymbol{H}}\hat{\boldsymbol{g}}_i, \tag{37}$$

with the corresponding log likelihood

$$\log p(\boldsymbol{X}^{\text{IMG}} \mid \boldsymbol{X}^{\text{TXT}}, \boldsymbol{\theta}) = \log \prod_{i=1}^{n} p(\boldsymbol{x}_i^{\text{IMG}} \mid \boldsymbol{X}^{\text{TXT}}, \boldsymbol{\theta}) \tag{38}$$

$$= \log \prod_{i=1}^{n} \left[ \text{softmax}\left( \hat{\boldsymbol{H}}\hat{\boldsymbol{g}}_i \right) \right]_i. \tag{39}$$

Similarly, for text input, the corresponding model is

$$f(\boldsymbol{x}_i^{\text{TXT}} \mid \boldsymbol{X}^{\text{IMG}}, \boldsymbol{\theta}) = \hat{\boldsymbol{G}}\hat{\boldsymbol{h}}_i, \tag{40}$$

with the corresponding log likelihood

$$\log p(\boldsymbol{X}^{\text{TXT}} \mid \boldsymbol{X}^{\text{IMG}}, \boldsymbol{\theta}) = \log \prod_{i=1}^{n} p(\boldsymbol{x}_i^{\text{TXT}} \mid \boldsymbol{X}^{\text{IMG}}, \boldsymbol{\theta}) \tag{41}$$

$$= \log \prod_{i=1}^{n} \left[ \text{softmax}\left( \hat{\boldsymbol{G}}\hat{\boldsymbol{h}}_i \right) \right]_i. \tag{42}$$

*Why is this still a reasonable approximation?* For VLMs, it is important to capture interactions between modalities, and assuming independence seems problematic at first. However, as we are using a local post-hoc posterior estimation through the Laplace approximation, we are effectively introducing an independence conditionally on the MAP estimate of the (joint) contrastive loss. Thus, crucially, even though we assume independence between modalities, we can still capture interactions between modalities. Note that this assumption is also important for computational reasons, as it helps us derive a computationally efficient approach.

### C.2.2 POSTERIOR APPROXIMATION WITH LA

Now that we have a well-defined probabilistic model and likelihood, we apply the Laplace approximation to it.

**Why only treat $\boldsymbol{P}$ and $\boldsymbol{Q}$ probabilistically** In the Laplace approximation, for the posterior covariance, we need to compute the Hessian of the log likelihood. This is computationally infeasible for large models and large datasets, and a common approximation is Generalised Gauss–Newton (GGN) approximation (Schraudolph, 2002). Use shorthand $f_{\boldsymbol{\theta}}(\boldsymbol{x})$ for the model and denote the log likelihood as $\ell(y, f_{\boldsymbol{\theta}}(\boldsymbol{x}))$, the GGN approximates to the Hessian is given by

$$\nabla_{\boldsymbol{\theta}}^2 \ell(y, f_{\boldsymbol{\theta}}(\boldsymbol{x})) \approx GGN(\boldsymbol{\theta}) \triangleq \frac{\partial f_{\boldsymbol{\theta}}(\boldsymbol{x})}{\partial \boldsymbol{\theta}}^{\top} \frac{\partial^2 \ell(y, f_{\boldsymbol{\theta}}(\boldsymbol{x}))}{\partial f_{\boldsymbol{\theta}}(\boldsymbol{x})^2} \frac{\partial f_{\boldsymbol{\theta}}(\boldsymbol{x})}{\partial \boldsymbol{\theta}} \tag{43}$$

Note that in GGN approximation, we need to compute the Jacobian of the model output w.r.t. to the model parameters $\frac{\partial f_{\boldsymbol{\theta}}(\boldsymbol{x})}{\partial \boldsymbol{\theta}}$. This is computationally infeasible for image and text encoders due to the large number of output dimensions. For image projection and text projection, this challenge can be bypassed as the Jacobian can be obtained analytically. Therefore, we treat the vision and image encoder as fixed and apply the Laplace approximation only for the image projection and text projection $\boldsymbol{P}$ and $\boldsymbol{Q}$.

**KFAC GGN approximation to Hessian** To estimate the Hessian of the log likelihood for $\boldsymbol{P}$ and $\boldsymbol{Q}$, we use Kronecker-factored approximate curvature (KFAC), which expresses the Hessian as a Kronecker product of two smaller matrices. This significantly reduces computational and memory

costs while preserving a richer posterior structure than diagonal approximations. Following (Ritter et al., 2018), the KFAC GGN approximation for $-\nabla_{\boldsymbol{P}}^2 \log p(\boldsymbol{X}^{\text{IMG}} \mid \boldsymbol{X}^{\text{TXT}}, \boldsymbol{P})$ is

$$\underbrace{\left( \frac{1}{\sqrt{n}} \sum_{i=1}^{n} \phi(\boldsymbol{x}_i^{\text{IMG}}) \phi(\boldsymbol{x}_i^{\text{TXT}})^\top \right)}_{\boldsymbol{A}_{\text{IMG}}} \otimes \underbrace{\left( \frac{1}{\sqrt{n}} \sum_{i=1}^{n} \boldsymbol{J}_{\text{IMG}}(\boldsymbol{x}_i^{\text{IMG}})^\top \boldsymbol{\Lambda}_{\text{IMG}} \boldsymbol{J}_{\text{IMG}}(\boldsymbol{x}_i^{\text{IMG}}) \right)}_{\boldsymbol{B}_{\text{IMG}}}, \qquad (44)$$

and the KFAC GGN approximation for $-\nabla_{\boldsymbol{Q}}^2 \log p(\mathcal{D} \mid \boldsymbol{Q})$ is

$$\underbrace{\left( \frac{1}{\sqrt{n}} \sum_{i=1}^{n} \psi(\boldsymbol{x}_i^{\text{TXT}}) \psi(\boldsymbol{x}_i^{\text{TXT}})^\top \right)}_{\boldsymbol{A}_{\text{TXT}}} \otimes \underbrace{\left( \frac{1}{\sqrt{n}} \sum_{i=1}^{n} \boldsymbol{J}_{\text{TXT}}(\boldsymbol{x}_i^{\text{TXT}})^\top \boldsymbol{\Lambda}_{\text{TXT}} \boldsymbol{J}_{\text{TXT}}(\boldsymbol{x}_i^{\text{TXT}}) \right)}_{\boldsymbol{B}_{\text{TXT}}}, \qquad (45)$$

where $\boldsymbol{J}_{\text{IMG}}(\boldsymbol{x}_i^{\text{IMG}}) = \frac{\partial \hat{\boldsymbol{H}} \frac{\boldsymbol{g}_i}{\|\boldsymbol{g}_i\|}}{\partial \boldsymbol{g}_i}$ and $\boldsymbol{\Lambda}_{\text{IMG}} = \text{diag}(\boldsymbol{\pi}) - \boldsymbol{\pi}\boldsymbol{\pi}^\top$, with $\pi_c = \frac{\exp(f_c)}{\sum_{c'} \exp(f_{c'})}$, $\hat{\boldsymbol{g}}_i^\top \hat{\boldsymbol{h}}_c =: f_c$.

As estimating the Kronecker factors over billions of data is computationally infeasible, following (Ritter et al., 2018), we leverage a subset of the data and include a pseudo-data count $\tau$ to compensate for the reduced sample size. Putting everything together, the posterior covariance over $\boldsymbol{P}$ and $\boldsymbol{Q}$ are approximated as

$$\boldsymbol{\Sigma}_{\text{IMG}} = (\tau(\boldsymbol{A}_{\text{IMG}} \otimes \boldsymbol{B}_{\text{IMG}}) + \lambda \boldsymbol{I})^{-1} \approx \underbrace{\left( \sqrt{\tau} \boldsymbol{A}_{\text{IMG}} + \sqrt{\lambda} \boldsymbol{I} \right)^{-1}}_{\tilde{\boldsymbol{A}}_{\text{IMG}}^{-1}} \otimes \underbrace{\left( \sqrt{\tau} \boldsymbol{B}_{\text{IMG}} + \sqrt{\lambda} \boldsymbol{I} \right)^{-1}}_{\tilde{\boldsymbol{B}}_{\text{IMG}}^{-1}}, \quad (46)$$

$$\boldsymbol{\Sigma}_{\text{TXT}} = (\tau(\boldsymbol{A}_{\text{TXT}} \otimes \boldsymbol{B}_{\text{TXT}}) + \lambda \boldsymbol{I})^{-1} \approx \underbrace{\left( \sqrt{\tau} \boldsymbol{A}_{\text{TXT}} + \sqrt{\lambda} \boldsymbol{I} \right)^{-1}}_{\tilde{\boldsymbol{A}}_{\text{TXT}}^{-1}} \otimes \underbrace{\left( \sqrt{\tau} \boldsymbol{B}_{\text{TXT}} + \sqrt{\lambda} \boldsymbol{I} \right)^{-1}}_{\tilde{\boldsymbol{B}}_{\text{TXT}}^{-1}}, \quad (47)$$

where the respective factors are given as:

$$\boldsymbol{A}_{\text{IMG}} = \frac{1}{\sqrt{n}} \sum_{i=1}^{n} \phi(\boldsymbol{x}_i^{\text{IMG}}) \phi(\boldsymbol{x}_i^{\text{IMG}})^\top$$

$$\boldsymbol{A}_{\text{TXT}} = \frac{1}{\sqrt{n}} \sum_{i=1}^{n} \psi(\boldsymbol{x}_i^{\text{TXT}}) \psi(\boldsymbol{x}_i^{\text{TXT}})^\top, \qquad (48)$$

$$\boldsymbol{B}_{\text{IMG}} = \frac{1}{\sqrt{n}} \sum_{i=1}^{n} \boldsymbol{J}_{\text{IMG}}(\boldsymbol{x}_i^{\text{IMG}})^\top \boldsymbol{\Lambda}_{\text{IMG}} \boldsymbol{J}_{\text{IMG}}(\boldsymbol{x}_i^{\text{IMG}})$$

$$\boldsymbol{B}_{\text{TXT}} = \frac{1}{\sqrt{n}} \sum_{i=1}^{n} \boldsymbol{J}_{\text{TXT}}(\boldsymbol{x}_i^{\text{TXT}})^\top \boldsymbol{\Lambda}_{\text{TXT}} \boldsymbol{J}_{\text{TXT}}(\boldsymbol{x}_i^{\text{TXT}}), \qquad (49)$$

**Jacobian computation**  Here we derive the Jacobians $\boldsymbol{J}_{\text{IMG}}(\boldsymbol{x}_i^{\text{IMG}})$ and $\boldsymbol{J}_{\text{TXT}}(\boldsymbol{x}_i^{\text{TXT}})$ used in the KFAC GGN approximation.

Recall $\hat{\boldsymbol{g}}_i$ and $\hat{\boldsymbol{h}}_j$ denote the normalized image and text embedding, respectively. Let $\hat{\boldsymbol{H}}$ denote the matrix of normalized text embeddings with $\hat{\boldsymbol{h}}_j$ as its columns and $\hat{\boldsymbol{G}}$ the matrix of normalized image embeddings with $\hat{\boldsymbol{g}}_i$ as its columns. Then, for the InfoNCE likelihood, which depends on the dot product between the normalised embedding in the batch, we compute the Jacobian for the image

encoder as follows:

$$J_{\text{IMG}}^{\text{InfoNCE}}(x_i^{\text{IMG}}) = \frac{\partial \hat{H}\hat{g}_i}{\partial g_i} \tag{50}$$

$$= \hat{H}\frac{\partial}{\partial g_i}\frac{g_i}{\|g_i\|} \tag{51}$$

$$= \hat{H}\frac{\|g_i\| - g_i\frac{\partial\|g_i\|}{\partial g_i}}{\|g_i\|^2} \tag{52}$$

$$= \hat{H}\frac{\|g_i\| - \frac{g_i g_i^\top}{\|g_i\|}}{\|g_i\|^2} \tag{53}$$

$$= \hat{H}\left(\frac{1}{\|g_i\|} - \frac{g_i g_i^\top}{\|g_i\|^3}\right). \tag{54}$$

Analogously, we obtain the Jacobian for the text encoder given as:

$$J_{\text{TXT}}^{\text{InfoNCE}}(x_i^{\text{TXT}}) = \hat{G}\left(\frac{1}{\|h_i\|} - \frac{h_i h_i^\top}{\|h_i\|^3}\right). \tag{55}$$

For SigLIP, we obtain the following Jacobians:

$$J_{\text{IMG}}^{\text{SigLIP}}(x_i^{\text{IMG}}) = \frac{\partial \hat{g}_i}{\partial g_i} = \left(\frac{1}{\|g_i\|} - \frac{g_i g_i^\top}{\|g_i\|^3}\right), \tag{56}$$

and

$$J_{\text{TXT}}^{\text{SigLIP}}(x_i^{\text{TXT}}) = \frac{\partial \hat{h}_i}{\partial h_i} = \left(\frac{1}{\|h_i\|} - \frac{h_i h_i^\top}{\|h_i\|^3}\right). \tag{57}$$

**Hessian of likelihood w.r.t. model output computation** Here we derive the loss Hessian w.r.t. model output $\Lambda_{\text{IMG}}$ and $\Lambda_{\text{TXT}}$. For InfoNCE loss used in CLIP, the zero-shot classifier induced computes unnormalised logits for each class $c$, represented by $\hat{g}_i^\top \hat{h}_c =: f_c$. By applying the softmax function, we calculate the probabilities for each class $c$ as $\pi_c = \frac{\exp(f_c)}{\sum_{c'} \exp(f_{c'})}$. The likelihood Hessian of the cross-entropy loss for this classifier is represented by

$$\Lambda_{\text{IMG}}^{\text{InfoNCE}} = \text{diag}(\pi) - \pi\pi^\top. \tag{58}$$

Similarly, the likelihood Hessian for the text encoder follows analogous principles in the text-to-image direction. For a more detailed derivation of the likelihood Hessian, we refer to (Rasmussen & Williams, 2006, Ch. 3.5). Rearranging terms in the analytical expression for $J_{\text{IMG}}^\top \Lambda_{\text{IMG}}^{\text{InfoNCE}} J_{\text{IMG}}$ facilitates space-efficient computation of the GGN approximation.

The SigLIP loss is defined as follows

$$\mathcal{L}_{\text{SigLIP}}(X^{\text{IMG}}, X^{\text{TXT}}) \tag{59}$$

$$= -\frac{1}{n}\sum_{i=1}^n\sum_{j=1}^n \log\frac{1}{1 + \exp(-z_{ij}(t\hat{g}_i^\top\hat{h}_j + b))} \tag{60}$$

$$= \frac{1}{n}\sum_{i=1}^n\sum_{j=1}^n \underbrace{-\log\sigma(a_{ij})}_{:=\ell(\hat{g}_i,\hat{h}_j)}, \tag{61}$$

where $\sigma(a) = \frac{1}{1+e^{-a}}$ denotes the sigmoid function, and $a_{ij} := z_{ij}(t\hat{g}_i^\top\hat{h}_j + b)$, with labels $z_{ij} \in \{-1, 1\}$, a learnable temperature scaling parameter $t$, and a learnable bias $b$.

In order to derive the loss Hessian $\boldsymbol{\Lambda}^{\text{SigLIP}}$, we first derive the component-wise loss gradient of $\ell$:

$$\frac{\partial}{\partial \hat{\boldsymbol{g}}_k} \ell(\hat{\boldsymbol{g}}_i, \hat{\boldsymbol{h}}_j) \stackrel{\text{i}\neq\text{k}}{=} 0 \tag{62}$$

$$\frac{\partial}{\partial \hat{\boldsymbol{g}}_k} \ell(\hat{\boldsymbol{g}}_i, \hat{\boldsymbol{h}}_j) \stackrel{\text{i}=\text{k}}{=} \frac{\partial}{\partial \hat{\boldsymbol{g}}_k} - \log \sigma(a_{ij}) \tag{63}$$

$$= -\frac{1}{\sigma(a_{ij})} \frac{\partial \sigma(a_{ij})}{\partial a_{ij}} \frac{\partial a_{ij}}{\partial \hat{\boldsymbol{g}}_i} \tag{64}$$

$$= (\sigma(a_{ij}) - 1) z_{ij} t \hat{\boldsymbol{h}}_j, \tag{65}$$

which we utilise to derive the component-wise loss Hessian

$$\frac{\partial^2}{\partial \hat{\boldsymbol{g}}_k \partial \hat{\boldsymbol{g}}_k^\top} \ell(\hat{\boldsymbol{g}}_i, \hat{\boldsymbol{h}}_j) \stackrel{\text{i}\neq\text{k}}{=} 0 \tag{66}$$

$$\frac{\partial^2}{\partial \hat{\boldsymbol{g}}_k \partial \hat{\boldsymbol{g}}_k^\top} \ell(\hat{\boldsymbol{g}}_i, \hat{\boldsymbol{h}}_j) \stackrel{\text{i}=\text{k}}{=} \frac{\partial}{\partial \hat{\boldsymbol{g}}_k^\top} \left( \sigma(a_{ij}) z_{ij} t \hat{\boldsymbol{h}}_j - z_{ij} t \hat{\boldsymbol{h}}_j \right) \tag{67}$$

$$= z_{ij} t \hat{\boldsymbol{h}}_k \frac{\partial \sigma(a_{ij})}{\partial a_{ij}} \frac{\partial a_{ij}}{\partial \hat{\boldsymbol{g}}_k^\top} \tag{68}$$

$$= t^2 \sigma(a_{ij}) \left(1 - \sigma(a_{ij})\right) \hat{\boldsymbol{h}}_j \hat{\boldsymbol{h}}_j^\top. \tag{69}$$

Finally, the likelihood Hessian for the SigLIP loss $\mathcal{L}_{\text{SigLIP}}$ can be expressed as

$$\boldsymbol{\Lambda}_{\text{IMG}}^{\text{SigLIP}} = \frac{\partial^2}{\partial \hat{\boldsymbol{g}}_i \partial \hat{\boldsymbol{g}}_i^\top} \mathcal{L}(\hat{\boldsymbol{g}}_{1:n}, \hat{\boldsymbol{h}}_{1:n}) \tag{70}$$

$$= \frac{1}{n} \sum_{j=1}^{n} \sum_{i=1}^{n} \frac{\partial^2}{\partial \hat{\boldsymbol{g}}_i \partial \hat{\boldsymbol{g}}_i^\top} \ell(\hat{\boldsymbol{g}}_i, \hat{\boldsymbol{h}}_j) \tag{71}$$

$$= \frac{t^2}{n} \sum_{j=1}^{n} \sigma(a_{ij}) \left(1 - \sigma(a_{ij})\right) \hat{\boldsymbol{h}}_j \hat{\boldsymbol{h}}_j^\top \tag{72}$$

for the image encoder and as

$$\boldsymbol{\Lambda}_{\text{TXT}}^{\text{SigLIP}} = \frac{t^2}{n} \sum_{i=1}^{n} \sigma(a_{ij}) \left(1 - \sigma(a_{ij})\right) \hat{\boldsymbol{g}}_i \hat{\boldsymbol{g}}_i^\top \tag{73}$$

for the text encoder.

**Efficient Hessian Computation** At first glance, computing the Hessian appears prohibitively expensive: the loss Hessian $\boldsymbol{\Lambda}$ has shape $|\mathcal{B}| \times |\mathcal{B}|$ with $|\mathcal{B}| \approx 32\text{k}$, while the embedding dimension is much smaller ($d \approx 512$). Forming $\boldsymbol{\Lambda}$ explicitly is therefore impractical. By exploiting its low-rank structure and contracting with the Jacobians, however, the computation can be carried out efficiently without ever materializing $\boldsymbol{\Lambda}$, making the GGN approximation feasible even for large batches. For example, for the image encoder with the InfoNCE loss in CLIP, the GGN block simplifies to

$$\boldsymbol{J}_{\text{IMG}}(\boldsymbol{x}_i^{\text{IMG}})^\top \boldsymbol{\Lambda}_{\text{IMG}} \boldsymbol{J}_{\text{IMG}}^{\text{InfoNCE}} \tag{74}$$

$$= \underbrace{\left( \frac{1}{\|\boldsymbol{h}_i\|} - \frac{\boldsymbol{h}_i \boldsymbol{h}_i^\top}{\|\boldsymbol{h}_i\|^3} \right)}_{\boldsymbol{M} \in \mathbb{R}^{d \times d}} \underbrace{\hat{\boldsymbol{G}}^\top}_{\in \mathbb{R}^{d \times |\mathcal{B}|}} \underbrace{\left(\text{diag}(\boldsymbol{\pi}) - \boldsymbol{\pi}\boldsymbol{\pi}^\top\right)}_{\in \mathbb{R}^{|\mathcal{B}| \times |\mathcal{B}|}} \underbrace{\hat{\boldsymbol{G}}}_{\in \mathbb{R}^{|\mathcal{B}| \times d}} \underbrace{\left( \frac{1}{\|\boldsymbol{h}_i\|} - \frac{\boldsymbol{h}_i \boldsymbol{h}_i^\top}{\|\boldsymbol{h}_i\|^3} \right)}_{\boldsymbol{M} \in \mathbb{R}^{d \times d}} \tag{75}$$

$$= \boldsymbol{M} \left( \hat{\boldsymbol{G}}^\top \text{diag}(\boldsymbol{\pi}) \hat{\boldsymbol{G}} - \hat{\boldsymbol{G}}^\top \boldsymbol{\pi}\boldsymbol{\pi}^\top \hat{\boldsymbol{G}} \right) \boldsymbol{M} \tag{76}$$

$$= \boldsymbol{M} \left( \underbrace{\hat{\boldsymbol{G}}^\top}_{\in \mathbb{R}^{d \times |\mathcal{B}|}} \underbrace{(\boldsymbol{\pi} \odot \hat{\boldsymbol{G}})}_{\in \mathbb{R}^{|\mathcal{B}| \times d}} - \underbrace{(\hat{\boldsymbol{G}}^\top \boldsymbol{\pi})}_{\in \mathbb{R}^{d \times |\mathcal{B}|}} \underbrace{(\hat{\boldsymbol{G}}^\top \boldsymbol{\pi})^\top}_{\in \mathbb{R}^{|\mathcal{B}| \times d}} \right) \boldsymbol{M} \tag{77}$$

where $\odot$ denotes row-wise scaling of $\hat{\boldsymbol{G}}$ by the vector $\boldsymbol{\pi}$.

**Marginal likelihood**  To learn the prior precision parameter $\lambda$, we follow prior work (*e.g.*, (Immer et al., 2021)) and optimise the log marginal likelihood within each probabilistic model. For the image projection layer $\boldsymbol{P}$, denote the prior and posterior as below:

$$\text{prior} : \mathcal{N}(\boldsymbol{0}, \lambda_{\text{IMG}}\boldsymbol{I}) \tag{78}$$

$$\text{posterior} : \mathcal{N}(\boldsymbol{P}_{\text{MAP}}, \boldsymbol{\Sigma}_{\text{IMG}}) \tag{79}$$

The marginal likelihood is

$$\log p(\boldsymbol{X}^{\text{IMG}} \mid \boldsymbol{X}^{\text{TXT}}) \approx \sum_{i=1}^{n} \log p(\boldsymbol{x}_i^{\text{IMG}} \mid \boldsymbol{X}^{\text{TXT}}, \boldsymbol{P}_{\text{MAP}}) \tag{80}$$

$$- \frac{1}{2}\left(\boldsymbol{P}_{\text{MAP}}^{\top}\lambda\boldsymbol{I}\boldsymbol{P}_{\text{MAP}} - \log\det(\boldsymbol{\Sigma}_{\text{IMG}}) + \log\det(\lambda_{\text{IMG}}\boldsymbol{I})\right) \tag{81}$$

We can learn the prior precision $\lambda_{\text{IMG}}$ using gradient-based optimisation.

**Distribution over image and vision features**  For completeness, we will briefly derive the distribution over image and vision features. In particular, for the image encoder let $\boldsymbol{P} \sim \mathcal{MN}(\boldsymbol{P}_{\text{MAP}}, \boldsymbol{B}_{\text{IMG}}^{-1}, \boldsymbol{A}_{\text{IMG}}^{-1})$, then:

$$\boldsymbol{g} = \boldsymbol{P}\phi(\boldsymbol{x}^{\text{IMG}}) \tag{82}$$

with $\boldsymbol{P}\phi(\boldsymbol{x}^{\text{IMG}}) \sim$

$$\mathcal{MN}(\boldsymbol{P}_{\text{MAP}}\phi(\boldsymbol{x}^{\text{IMG}}), \boldsymbol{B}_{\text{IMG}}^{-1}, \phi(\boldsymbol{x}^{\text{IMG}})^{\top}\boldsymbol{A}_{\text{IMG}}^{-1}\phi(\boldsymbol{x}^{\text{IMG}}))$$

$$\boldsymbol{g} \sim \mathcal{N}(\boldsymbol{P}_{\text{MAP}}\phi(\boldsymbol{x}^{\text{IMG}}), \left(\phi(\boldsymbol{x}^{\text{IMG}})^{\top}\boldsymbol{A}_{\text{IMG}}^{-1}\phi(\boldsymbol{x}^{\text{IMG}})\right)\boldsymbol{B}_{\text{IMG}}^{-1}). \tag{83}$$

## C.3  DISTRIBUTION OVER COSINE SIMILARITIES

For the derivation of the distribution over cosine similarities, first recall the definition of the cosine similarity between two vectors, $\boldsymbol{g}$ and $\boldsymbol{h}$, which is given as $\text{S}_{\cos}(\boldsymbol{g}, \boldsymbol{h}) = \frac{\boldsymbol{g}^{\top}\boldsymbol{h}}{\|\boldsymbol{g}\|\|\boldsymbol{h}\|}$. Now, let $\boldsymbol{g}$ and $\boldsymbol{h}$ denote random vectors for the image and text embeddings, respectively. Further, let us assume that their distribution follows a Gaussian distribution with mean $\boldsymbol{\mu_g} = (\mu_{\boldsymbol{g},1}, \ldots, \mu_{\boldsymbol{g},d})$ and $\boldsymbol{\mu_h} = (\mu_{\boldsymbol{h},1}, \ldots, \mu_{\boldsymbol{h},d})$ and diagonal covariance structure, *i.e.*, $\boldsymbol{\Sigma_g} = \text{diag}(\sigma_{\boldsymbol{g},1}^2, \ldots, \sigma_{\boldsymbol{g},d}^2)$ and $\boldsymbol{\Sigma_h} = \text{diag}(\sigma_{\boldsymbol{h},1}^2, \ldots, \sigma_{\boldsymbol{h},d}^2)$.

Then the expected value of the cosine similarity is:

$$\mathbb{E}[\text{S}_{\cos}(\boldsymbol{g}, \boldsymbol{h})] = \frac{\mathbb{E}[\boldsymbol{g}^{\top}\boldsymbol{h}]}{\mathbb{E}[\|\boldsymbol{g}\|]\mathbb{E}[\|\boldsymbol{h}\|]} \tag{84}$$

$$= \frac{\sum_i^d \mu_{\boldsymbol{g},i}\mu_{\boldsymbol{h},i}}{\mathbb{E}[\|\boldsymbol{g}\|]\mathbb{E}[\|\boldsymbol{h}\|]}. \tag{85}$$

Note that computing $\mathbb{E}[\|\boldsymbol{x}\|]$ is intractable, and we, therefore, bound the expected value by application of the triangle inequality, *i.e.*,

$$\mathbb{E}[\|\boldsymbol{x}\|] \leq \sqrt{\sum_i \mu_{\boldsymbol{x},i}^2 + \sigma_{\boldsymbol{x},i}^2}, \tag{86}$$

where we use the fact that $\mathbb{E}[x^2] = \mu_x^2 + \sigma_x^2$. Consequently, we obtain an approximation to the expected value of the cosine similarity given by:

$$\mathbb{E}[\text{S}_{\cos}(\boldsymbol{g}, \boldsymbol{h})] \approx \frac{\sum_i^d \mu_{\boldsymbol{g},i}\mu_{\boldsymbol{h},i}}{\sqrt{\sum_i \mu_{\boldsymbol{g},i}^2 + \sigma_{\boldsymbol{g},i}^2}\sqrt{\sum_i \mu_{\boldsymbol{h},i}^2 + \sigma_{\boldsymbol{h},i}^2}}. \tag{87}$$

Next, we will derive the second moment (variance) of the cosine similarity of two random vectors. First, note that the variance can be written as the difference between two expectations, *i.e.*,

$$\mathbb{V}\text{ar}[\text{S}_{\cos}(\boldsymbol{g}, \boldsymbol{h})] = \mathbb{E}[\text{S}_{\cos}(\boldsymbol{g}, \boldsymbol{h})^2] - \mathbb{E}[\text{S}_{\cos}(\boldsymbol{g}, \boldsymbol{h})]^2, \tag{88}$$

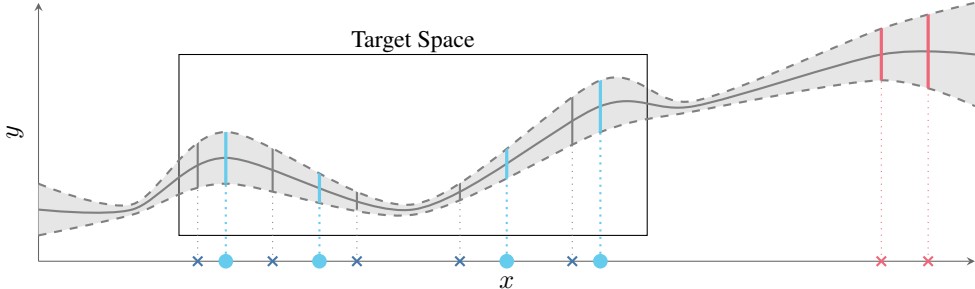

Figure 7: Illustration of targeted support set selection. We aim to select an informative support set that reduces the uncertainty over the predictions on the query set ●. Only focusing on the epistemic uncertainties would not lead to a good selection, as we would select uninformative support set candidates × with high epistemic uncertainty. Hence, we target the selection process.

where the second expectation corresponds to:

$$\mathbb{E}[S_{\cos}(\boldsymbol{g},\boldsymbol{h})]^2 \approx \frac{(\sum_i^d \mu_{\boldsymbol{g},i}\mu_{\boldsymbol{h},i})^2}{\sum_i \mu_{\boldsymbol{g},i}^2 + \sigma_{\boldsymbol{g},i}^2 \sum_i \mu_{\boldsymbol{h},i}^2 + \sigma_{\boldsymbol{h},i}^2}. \tag{89}$$

Next we can obtain $\mathbb{E}[S_{\cos}(\boldsymbol{g},\boldsymbol{h})^2]$ for which we will use the fact that $\mathbb{E}[x^2] = \mu_x^2 + \sigma_x^2$ again, *i.e.*,

$$\mathbb{E}[S_{\cos}(\boldsymbol{g},\boldsymbol{h})^2] = \frac{\mathbb{E}[(\boldsymbol{g}^\top\boldsymbol{h})^2]}{\sum_i \mu_{\boldsymbol{g},i}^2 + \sigma_{\boldsymbol{g},i}^2 \sum_i \mu_{\boldsymbol{h},i}^2 + \sigma_{\boldsymbol{h},i}^2} \tag{90}$$

where

$$\mathbb{E}[(\boldsymbol{g}^\top\boldsymbol{h})^2] = \sum_i \sum_j \mu_{\boldsymbol{g},i}\mu_{\boldsymbol{h},i}\mu_{\boldsymbol{g},j}\mu_{\boldsymbol{h},j} \tag{91}$$

$$+ \sum_i \sigma_{\boldsymbol{g},i}^2\mu_{\boldsymbol{h},i}^2 + \mu_{\boldsymbol{g},i}^2\sigma_{\boldsymbol{h},i}^2 + \sigma_{\boldsymbol{g},i}^2\sigma_{\boldsymbol{h},i}^2. \tag{92}$$

Henceforth, we obtain the variance:

$$\mathbb{V}\mathrm{ar}[S_{\cos}(\boldsymbol{g},\boldsymbol{h})] = \frac{\sum_i \sigma_{\boldsymbol{g},i}^2(\sigma_{\boldsymbol{h},i}^2 + \mu_{\boldsymbol{h},i}^2) + \sigma_{\boldsymbol{h},i}^2\mu_{\boldsymbol{g},i}^2}{\sum_i \mu_{\boldsymbol{g},i}^2 + \sigma_{\boldsymbol{g},i}^2 \sum_i \mu_{\boldsymbol{h},i}^2 + \sigma_{\boldsymbol{h},i}^2}. \tag{93}$$

## D  ACTIVE LEARNING DETAILS

We provide additional details on our active learning setup. Active learning provides a natural setting to evaluate the quality of uncertainty estimates, as it relies on selecting informative samples based on predictive uncertainty. We assess BayesVLM in this setting using acquisition functions from Bayesian active learning, combined with adaptive target region selection. Concretely, given a query set $\mathcal{X}_{\text{test}} = \{x_i^\star\}_{i=1}^{n_{\text{test}}}$ of unseen samples with unknown class labels, our goal is to select a support set $\{(x_j, y_j)\}_{j=1}^m$ of labeled examples such that predictive uncertainty on $\mathcal{X}_{\text{test}}$ is reduced. To this end, we first target the selection process toward the predictive distribution of the query set, and then select support candidates based on their estimated influence on predictive or model uncertainty.

We detail our method in three parts: App. D.1 describes how we reduce the candidate pool by selecting samples that align with the target distribution; App. D.2 outlines the acquisition functions used for (targeted) active fine-tuning; and App. D.3 explains how we update the Laplace approximation in an online fashion during the EPIG acquisition process.

### D.1  TARGETED SELECTION

To target the active learning process towards relevant areas in the data space, we perform a $k$-nearest neighbours ($k$-NN) search around the test data. The main idea behind our adaptive targeted region selection is illustrated in Fig. 7.

Stage 1: 1-NN Selection

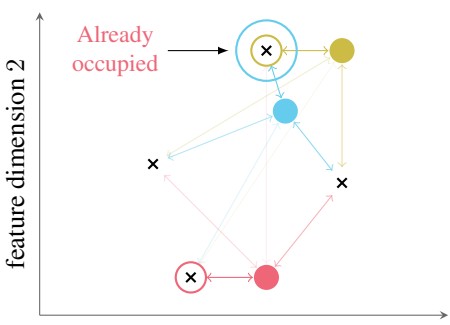

Stage 2: 2-NN Selection

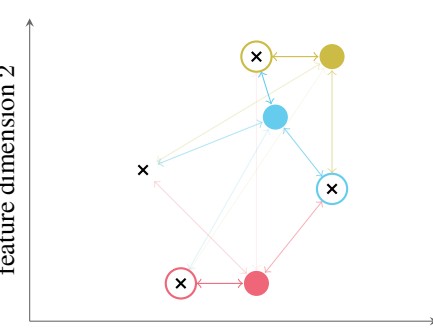

Figure 8: Illustration of the nearest neighbour-based support set selection for adaptive targeted selection. The circles ● show test data points with uncertainty scores depicted through their colours: high, medium, low. For each test datum we find the $k = 1$ nearest neighbour from the support set candidates ✕. If the $k = 1$ nearest neighbour is already selected, we increase $k$ for those with occupied neighbours and choose the second nearest neighbour, *i.e.*, $k = 2$. This recursion continues until every test datum has a selected support set candidate. The selected candidates are shown in coloured circles. Note that in the case of the blue test datum, the closest support set candidate has already been chosen by the yellow, and hence the second closest candidate is selected in the second stage.

Specifically, we greedily acquire an intermediate candidate set $\mathcal{T}^{\star} \subseteq \mathcal{D}_{\text{train}}$ using $k$-NN selection based on the test set $\mathcal{D}_{\text{test}}$. For this, we need to compute a metric comparing the random feature projections. We assessed two different ways, first by computing the 2-Wasserstein distance between the distributions of the embeddings and the second by computing the expected cosine similarity based on App. C.3. Recall that for multivariate Gaussian distributions, the 2-Wasserstein distance exists in closed-form and is given as $W_2^2(\mathcal{N}(\boldsymbol{\mu}_1, \boldsymbol{\Sigma}_1), \mathcal{N}(\boldsymbol{\mu}_2, \boldsymbol{\Sigma}_2)) =$

$$\|\boldsymbol{\mu}_1 - \boldsymbol{\mu}_2\|_2^2 + \text{tr}\left(\boldsymbol{\Sigma}_1 + \boldsymbol{\Sigma}_2 - 2(\boldsymbol{\Sigma}_1^{1/2}\boldsymbol{\Sigma}_2\boldsymbol{\Sigma}_1^{1/2})^{1/2}\right), \tag{94}$$

where $\|\cdot\|_2$ denotes the Euclidean norm, $\text{tr}(\cdot)$ is the trace operator, and $\boldsymbol{\Sigma}^{1/2}$ is the matrix square root of $\boldsymbol{\Sigma}$. As computing the Wasserstein distance exactly is computationally and memory intensive due to the matrix square root, we approximate it by assuming both distributions to be isotropic. Hence, simplifying to $W_2^2(\mathcal{N}(\boldsymbol{\mu}_1, \boldsymbol{\Sigma}_1), \mathcal{N}(\boldsymbol{\mu}_2, \boldsymbol{\Sigma}_2)) =$

$$\sum_{i=1}^{d} (\mu_{1,i} - \mu_{2,i})^2 + \sigma_{1,i}^2 + \sigma_{2,i}^2 - 2\sigma_{1,i}\sigma_{2,i}, \tag{95}$$

where $\boldsymbol{\Sigma}_1 = \text{diag}(\sigma_{1,1}^2, \ldots, \sigma_{1,d}^2)$ and $\boldsymbol{\Sigma}_2$ is given respectively.

Based on a selected metric, we select the training samples closest to the test set in the joint embedding space, resulting in:

$$\mathcal{T} = \bigcup_{\boldsymbol{g}^{\star} \in \mathcal{T}^{\star}} N_k(\boldsymbol{g}^{\star}, \mathcal{D}_{\text{train}}), \tag{96}$$

with $N_k(\boldsymbol{g}^{\star}, \mathcal{D}_{\text{train}})$ denoting the set of $k$-nearest neighbours of $\boldsymbol{g}^{\star}$ in the training set $\mathcal{D}_{\text{train}}$. To ensure that we select $k$ distinct data points for each test sample, we perform an iterative search in which we discard already selected training samples and iteratively increase the search radius until $k$ distinct samples are found for each test datum. This process is illustrated in Fig. 8.

### D.2 ACQUISITION FUNCTIONS

Given a labelled pool $\mathcal{D}_{\text{train}}$ and an unlabelled target set $\mathcal{X}_{\text{test}} = \{\boldsymbol{x} \mid (\boldsymbol{x}, y) \in \mathcal{D}_{\text{test}}\}$, the goal is to select $m$ maximally informative samples from $\mathcal{D}_{\text{train}}$ to reduce predictive uncertainty on $\mathcal{X}_{\text{test}}$. In this section, we provide a detailed explanation of the acquisition functions used for this purpose.

**Naïve random**  For the *naïve random* acquisition function, we randomly sample $m$ data points from the train set $\mathcal{D}_{\text{train}}$ to form the support set $\mathcal{S}_{\text{ID}}$.

**Targeted random**  For the *targeted random* acquisition function, we randomly sample $m$ data points from the unlabelled test set $\mathcal{X}_{\text{test}}$ to form an intermediate support set $\mathcal{T}^*$. According to App. D.1, we then select the nearest neighbours to $\mathcal{T}^*$ from the training set $\mathcal{D}_{\text{train}}$ based on the cosine similarity of the normalized image embeddings to form the support set $\mathcal{T}_{\text{t-ID}}$.

**Targeted maximum entropy**  For the *entropy* acquisition function, we compute the predictive entropy $\mathcal{H}(y_i^* \mid \boldsymbol{x}_i^*)$ for each data point $\boldsymbol{x}_i^* \in \mathcal{X}_{\text{test}}$ and select the $m$ data points with the highest entropy. We use the predictive entropy on the MAP estimate of the model parameters to estimate the predictive entropy of the model:

$$
\mathcal{H}\left(y \mid \boldsymbol{x}, \boldsymbol{\theta}_{\text{MAP}}\right)
$$
$$
= -\sum_{c=1}^{C} p(y = c | \boldsymbol{x}, \boldsymbol{\theta}_{\text{MAP}}) \log p(y = c | \boldsymbol{x}, \boldsymbol{\theta}_{\text{MAP}}) \tag{97}
$$

According to App. D.1, we then select the most similar data points from $\mathcal{X}_{\text{train}}$ to form the support set $\mathcal{T}_{\text{t-entropy}}$.

**BALD**  We compute the BALD score (Houlsby et al., 2011) for each data point in $\mathcal{X}_{\text{train}}$ and select the $m$ data points with the highest score. The score is approximated using nested Monte Carlo sampling, as in (Houlsby et al., 2011).

$$
\text{BALD}(\boldsymbol{x}) \tag{98}
$$
$$
= \mathbb{E}_{p(y|\boldsymbol{x})} \left[ \mathcal{H}\left(p(\boldsymbol{\theta})\right) - \mathcal{H}\left(p(\boldsymbol{\theta} \mid \boldsymbol{x}, y)\right) \right] \tag{99}
$$
$$
= \mathbb{E}_{p(\boldsymbol{\theta}|\mathcal{D})} \left[ \mathcal{H}\left(p(y \mid \boldsymbol{x}, \boldsymbol{\theta})\right) - \mathcal{H}\left(p(y \mid \boldsymbol{x}, \mathcal{D})\right) \right] \tag{100}
$$

**Targeted BALD**  We compute the BALD score (Eq. (100)) for each data point $\boldsymbol{x}_i^* \in \mathcal{X}_{\text{test}}$ and select the $m$ data points with the highest score. According to App. D.1, we then select the most similar data points from $\mathcal{X}_{\text{train}}$ to form the support set $\mathcal{T}_{\text{t-BALD}}$.

**EPIG**  The Expected Predictive Information Gain (EPIG) score (Bickford Smith et al., 2023) calculates the expected mutual information between the model parameters and the predictive distribution resulting from the acquisition of a training data point. This method is specifically designed to target relevant information, eliminating the need for a $k$-nearest neighbour search typically used in other acquisition functions. The EPIG score is given by

$$
\text{EPIG}(\boldsymbol{x})
$$
$$
= \mathbb{E}_{p_*(\boldsymbol{x}^*)p(y|\boldsymbol{x})} \left[ \mathcal{H}\left(p(y^* \mid \boldsymbol{x}^*)\right) - \mathcal{H}\left(p(y^* \mid \boldsymbol{x}^*, \boldsymbol{x}, y)\right) \right] \tag{101}
$$
$$
= \mathbb{E}_{p_*(\boldsymbol{x}^*)} \left[ \mathrm{D}_{\text{KL}}\left(p(y, y^* \mid \boldsymbol{x}, \boldsymbol{x}^*) \,\|\, p(y \mid \boldsymbol{x})p(y^* \mid \boldsymbol{x}^*)\right) \right] \tag{102}
$$
$$
= \mathbb{E}_{p_*(\boldsymbol{x}^*)} \left[ \sum_{y \in \mathcal{Y}} \sum_{y^* \in \mathcal{Y}} p(y, y^* \mid \boldsymbol{x}, \boldsymbol{x}^*) \log \frac{p(y, y^* \mid \boldsymbol{x}, \boldsymbol{x}^*)}{p(y \mid \boldsymbol{x})p(y^* \mid \boldsymbol{x}^*)} \right] \tag{103}
$$

where $p_*(\boldsymbol{x}^*)$ denotes the target input distribution. The EPIG score is approximated using Monte Carlo sampling, as detailed in(Bickford Smith et al., 2023). For the EPIG selection, we perform online updates to the model weights using the online Laplace as described in App. D.3.

### D.3    ONLINE LAPLACE APPROXIMATION

We use an online Laplace approximation to efficiently update the posterior distribution over the image projection matrix $\boldsymbol{P}$ during active learning. Instead of recomputing the posterior from scratch after each support set update, we incrementally refine both the MAP estimate and the Kronecker-factored Hessian approximation using the newly selected datapoint. Concretely, we perform a gradient step to update $\boldsymbol{P}_{\text{MAP}}$, and adjust the Kronecker factors $\boldsymbol{A}_{\text{IMG}}$ and $\boldsymbol{B}_{\text{IMG}}$ based on the contribution of the new sample. This yields a computationally efficient approximation to the posterior over $\boldsymbol{P}$ conditioned on the growing support set. Additionally, the prior precision can optionally be re-estimated after each update step, as commonly done in online Laplace methods (Immer et al., 2021; Lin et al., 2023). In the following, we outline the structure of the Laplace approximation and describe how it is updated online during EPIG-based support set construction.

Recall that we obtain from our post-hoc Laplace approximation the Kronecker factorized Hessian approximation $\boldsymbol{H}_{\text{IMG}} \approx (\sqrt{\tau}\boldsymbol{A}_{\text{IMG}} + \sqrt{\lambda}\boldsymbol{I}) \otimes (\sqrt{\tau}\boldsymbol{B}_{\text{IMG}} + \sqrt{\lambda}\boldsymbol{I})$ with

$$\boldsymbol{A}_{\text{IMG}} = \frac{1}{\sqrt{n}}\sum_{i=1}^{n}\phi(\boldsymbol{x}_i^{\text{IMG}})\phi(\boldsymbol{x}_i^{\text{IMG}})^{\top} \quad \text{and} \tag{104}$$

$$\boldsymbol{B}_{\text{IMG}} = \frac{1}{\sqrt{n}}\sum_{i=1}^{n}\boldsymbol{J}_{\text{IMG}}(\boldsymbol{x}_i^{\text{IMG}})^{\top}\boldsymbol{\Lambda}_{\text{IMG}}\,\boldsymbol{J}_{\text{IMG}}(\boldsymbol{x}_i^{\text{IMG}}), \tag{105}$$

approximating a posterior distribution over the projection weights:

$$\boldsymbol{P} \sim \mathcal{MN}(\boldsymbol{P}_{\text{MAP}}, \widetilde{\boldsymbol{B}}_{\text{IMG}}^{-1}, \widetilde{\boldsymbol{A}}_{\text{IMG}}^{-1}) \tag{106}$$

$$\boldsymbol{Q} \sim \mathcal{MN}(\boldsymbol{Q}_{\text{MAP}}, \widetilde{\boldsymbol{B}}_{\text{TXT}}^{-1}, \widetilde{\boldsymbol{A}}_{\text{TXT}}^{-1}) \tag{107}$$

with $\widetilde{\boldsymbol{A}}$, $\widetilde{\boldsymbol{B}}$ denoting the Kronecker factors after applying $\tau$ and $\lambda$.

Further, utilizing App. C.3 in combination with the generalized probit approximation (as described, for instance, in (Daxberger et al., 2021)), we obtain an analytical form for the predictive posterior distribution $p(y \mid \boldsymbol{x}, \mathcal{D})$ of our few-shot classifier.

Our goal with EPIG is to iteratively construct a support set $\mathcal{T}_t$, where $t$ denotes the current number of selected training data points. We construct $\mathcal{T}_t$ by greedily selecting the training datum that maximises the expected information gain on the predictive distribution in the target domain:

$$\boldsymbol{x}_{t+1} = \underset{\boldsymbol{x}\in\mathcal{D}_{\text{train}}}{\arg\max}\,\text{EPIG}(\boldsymbol{x} \mid \mathcal{T}_t) \tag{108}$$

$$= \underset{\boldsymbol{x}\in\mathcal{D}_{\text{train}}}{\arg\max}\,\mathbb{E}_{p_*(\boldsymbol{x}^*)p(y|\boldsymbol{x})}\big[\mathcal{H}\left(p(y^* \mid \boldsymbol{x}^*, \mathcal{T}_t)\right) - \mathcal{H}\left(p\left(y^* \mid \boldsymbol{x}^*, \mathcal{T}_t \cup \{(\boldsymbol{x}, y)\}\right)\right)\big] \tag{109}$$

and obtain the corresponding label $y_{t+1}$, forming the support set $\mathcal{T}_{t+1} = \mathcal{T}_t \cup \{(\boldsymbol{x}_{t+1}, y_{t+1})\}$.

The integration of $\{(\boldsymbol{x}_{t+1}, y_{t+1})\}$ into the few-shot training set changes the posterior distribution over the image projection. To obtain the updated posterior distribution

$$\boldsymbol{P} \mid \mathcal{T}_{t+1} \sim \mathcal{MN}(\boldsymbol{P}_{\text{MAP}}, \widetilde{\boldsymbol{B}}_{\text{IMG},t+1}^{-1}, \widetilde{\boldsymbol{A}}_{\text{IMG},t+1}^{-1}), \tag{110}$$

we utilise the following online updates to the projection weights and the Laplace approximation:

$$\boldsymbol{P}_{\text{MAP},t+1} = \boldsymbol{P}_{\text{MAP},t} - \gamma\nabla_{\boldsymbol{P}}\mathcal{L}(\boldsymbol{x}_{t+1}^{\text{IMG}}, \boldsymbol{X}^{\text{TXT}}) \tag{111}$$

$$\boldsymbol{A}_{\text{IMG},t+1} = \frac{\sqrt{n+t}\,\boldsymbol{A}_{\text{IMG},t} + \beta\boldsymbol{A}_{\boldsymbol{x}_{t+1}}}{\sqrt{n+t+1}} \tag{112}$$

$$\boldsymbol{B}_{\text{IMG},t+1} = \frac{\sqrt{n+t}\,\boldsymbol{B}_{\text{IMG},t} + \beta\boldsymbol{B}_{\boldsymbol{x}_{t+1}}}{\sqrt{n+t+1}}. \tag{113}$$

where $\gamma \geq 0$ and $\beta \geq 0$ are hyperparameters and

$$\boldsymbol{A}_{\boldsymbol{x}_{t+1}} = \phi(\boldsymbol{x}_{t+1}^{\text{IMG}})\phi(\boldsymbol{x}_{t+1}^{\text{IMG}})^{\top} \tag{114}$$

$$\boldsymbol{B}_{\boldsymbol{x}_{t+1}} = \boldsymbol{J}_{\text{IMG}}(\boldsymbol{x}_{t+1}^{\text{IMG}})^{\top}\boldsymbol{\Lambda}_{\text{IMG}}\boldsymbol{J}_{\text{IMG}}(\boldsymbol{x}_{t+1}^{\text{IMG}}). \tag{115}$$

## E EXPERIMENTAL DETAILS

This section details the experimental setup used in our study. In App. E.1, we describe the pre-trained vision-language models and checkpoints used. App. E.2 explains how we computed the Hessian matrices required for the Laplace approximation. In App. E.3, we outline how we selected the Laplace parameters, such as the pseudo-data count $\tau$ and prior precision $\lambda$. App. E.4 describes our active learning setup, including dataset preparation, selection strategies, and training hyperparameters (see Table 6).

We run experiments on a compute cluster with NVIDIA P100 16GB, V100 32 GB, and A100 80GB GPUs. We used V100 or A100 GPUs for the Huge model variants and the ImageNet experiments.

### E.1 PRE-TRAINED VISION-LANGUAGE MODELS

In this work, we used the OpenCLIP (Ilharco et al., 2021) implementations of CLIP (Radford et al., 2021), which was published under the MIT license. We present additional experimental results on the HuggingFace implementation of SigLIP (Zhai et al., 2023), which was originally published under the Apache2 license.

For PCME++, we used the CLIP ViT-B/16 checkpoint provided by the authors with uncertainty adapter trained on CC-3M (Sharma et al., 2018), CC-12M (Changpinyo et al., 2021), and Redcaps (Desai et al., 2021). For ProLIP, we used the ViT-B/16 model checkpoint released by the authors at https://github.com/naver-ai/prolip, which shares the same backbone architecture.

### E.2 ESTIMATION OF THE HESSIAN MATRICES

We estimated the Hessians separately for the CLIP image and text encoders using the pre-training dataset LAION-400M (Schuhmann et al., 2022) published under MIT license. For this estimation, we randomly sampled a subset of 327.680 data points. The pre-training dataset was filtered to exclude NSFW content. For the Laplace approximation, we used the GGN approximation of the Hessian matrices as described in App. C.2 and estimated the covariance matrices $A$ and $B$ for the image and text encoders.

### E.3 ESTIMATION OF THE HESSIAN PARAMETERS

To ensure that both zero-shot and active learning experiments rely on a well-calibrated posterior covariance, we estimated the pseudo-data count (Ritter et al., 2018), $\tau$, by performing a grid search over values in $\tau \in [1, 5, 10, 15, \dots, 200]$. In the active learning experiments, the step size was reduced to 2. The optimal value of $\tau$ was selected by minimizing the negative log predictive density (NLPD) on a random subset of ImageNet consisting of 100 classes and 1097 test data points in total as a proxy. App. E.3 presents two plots illustrating the NLPD as a function of the pseudo-data count for SigLIP-Base and CLIP-Base, respectively. Once this optimal $\tau$ was identified, we further optimised the prior precision, $\lambda$, using the marginal likelihood on the LAION-400M (Schuhmann et al., 2022) dataset.

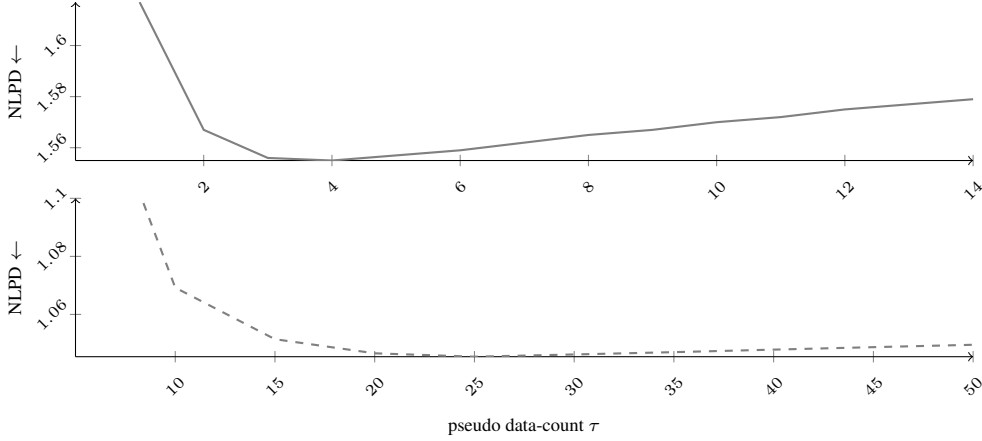

Figure 9: Grid search over the pseudo-data count parameter $\tau$ for CLIP-Base (——) and SigLIP-Base (- - -). The optimal NLPD for CLIP-Base is identified at $\tau = 10$, while the optimal NLPD for SigLIP-Base is identified at $\tau = 100$.

### E.4 ACTIVE LEARNING EXPERIMENTS

We conducted active learning experiments on the OfficeHome data set (Venkateswara et al., 2017), which consists of the domains: art, clipart, product, and real-world, as well as on the ImageNet dataset with the domains ImageNet-R (Hendrycks et al., 2021) and ImageNet-Sketch (Wang et al., 2019). For

these experiments, all training sets from the respective domains were combined into a single, large training set, and the projection layer of either CLIP or SigLIP was fine-tuned for a specific domain. Data selection was performed based on the acquisition functions described in App. D.2. Performance was evaluated at the checkpoint corresponding to the lowest NLPD on a domain-specific validation set. We also performed a grid search for the online learning parameters of the EPIG acquisition rule, selecting the EPIG learning rate $\gamma$ from the range [1e-5, 1e-4, 1e-3, 1e-2] and the EPIG Hessian update scale $\beta$ from [1, 10, 100, 1000], based on the NLPD on the domain-specific validation set. Details on the training hyperparameter settings are given in Table 6.

Table 6: Active fine-tuning hyperparameters.

| config | value |
|---|---|
| optimiser | AdamW |
| learning rate | $1e{-}5$ |
| weight decay | $5e{-}2$ |
| optimiser momentum | $\beta_1, \beta_2 = 0.9, 0.999$ |
| batch size | 32 |
| epochs | 100 |

## F ADDITIONAL RESULTS

In this Appendix, we provide additional results to support the findings in the main paper. Specifically, we detail (i) the approximation quality of the Gaussian approximation to the distribution over cosine similarities in App. F.1, (ii) the active learning experiments in App. F.2, (iii) the ablation of the $k$-NN distance metric in App. F.3, (iv) the influence of the number of data points used for Hessian estimation in App. F.4, (v) the runtime overhead and inference costs of BayesVLM compared to the baselines in App. F.5, and (vi) additional zero-shot learning results in App. F.7 to demonstrate the generality of our approach.

### F.1 APPROXIMATION QUALITY

To assess the approximation quality of the Gaussian approximation to the distribution over cosine similarities, we generated 500 samples for the image and text feature distributions for a given input. For the resulting samples, we then computed the respective cosine similarity for each pair and performed kernel density estimation with a Gaussian kernel and length scale of 0.3 on the similarity scores. We added increasing shifts to the distribution mean to evaluate the change in the approximation quality under varying cosine similarity values. The results are depicted in Fig. 10. We can observe that our approximation through ProbCosine results qualitatively in a low approximation error.

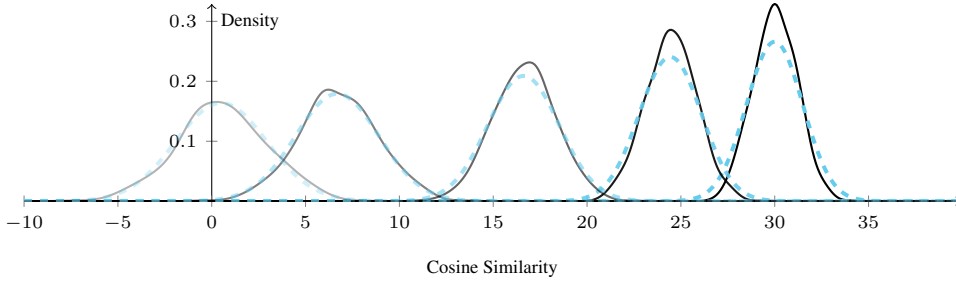

Figure 10: Approximation quality of the Gaussian approximation (ProbCosine) ($- - -$) to the distribution over cosine similarities compared to KDE over samples (——) for image-text pairs with increasing Euclidean distance between their feature projection means ($\mu_g, \mu_h$).

## F.2 ACTIVE LEARNING EXPERIMENTS

We report the active learning results for CLIP-Huge in Fig. 13 and for SigLIP-Base in Fig. 14. For CLIP-Huge, we observe that active learning based on our post-hoc uncertainties consistently improves upon random and entropy-based selection. For SigLIP-Base, we observe significant improvements in terms of accuracy and NLPD on the ImageNet version with active learning based on our post-hoc uncertainties.

## F.3 ABLATION OF THE $k$-NN DISTANCE METRIC

We performed an ablation study on the $k$-NN distance metric for our proposed targeted selection in Sec. 3.3 while fixing the *online LA learning rate* $\gamma = 1e-4$ and the *online LA pseudo-data count* $\beta = 10$. We evaluate performance using two distance metrics: Wasserstein and cosine similarity. Results are reported for EPIG and BALD, with Wasserstein (*solid lines*) and cosine (*dashed lines*) metrics. While Wasserstein-based k-NN selection demonstrates improved performance for datasets such as OfficeHome-Art and ImageNet-R, no clear trend is observed across the other datasets.

## F.4 NUMBER OF DATA POINTS FOR HESSIAN ESTIMATION

We assessed the influence of the number of data points used to estimate the Hessian by estimating the trace of the Hessian using a varying number of data points for 10 random subsets of the Laion400m data set. Fluctuations of the estimated trace can be understood as an indicator that the Hessian estimates are unreliable. Fig. 11 shows that the trace of both the Hessian over the image projection and the text projection quickly converges to a stable value with low fluctuations, as seen by the low variance and stable mean. Moreover, the results indicate that 10 mini-batches suffice to obtain a reliable estimate of the Hessians.

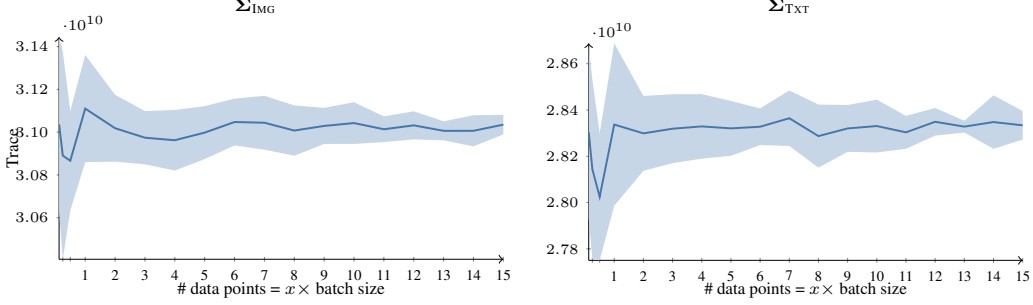

Figure 11: Hessian (Trace) vs. number of samples. Error bars indicate $\pm 1$ standard deviation over 10 random subsets of Laion400m.

## F.5 RUNTIME OVERHEAD

We compared our approach against the vanilla CLIP. We report runtimes on CIFAR-100 with CLIP-Huge/-Base on a Tesla V100 with 3 warmup steps and averaged over 1000 runs (batch size 1) in Table 7. Indicating minimal computational overhead. In addition, we present inference costs in GFLOPs, comparing the original VLM (deterministic) against TTA and BayesVLM in Table 8. We find that BayesVLM results in comparable inference costs, while TTA has an 80-fold increase in the inference cost.

Table 7: Average runtime measured in seconds on CIFAR-100.

| Model | Vanilla | BayesVLM | rel. increase |
|---|---|---|---|
| CLIP-Huge | 43.8178 | 43.9712 | 0.35% |
| CLIP-Bas | 9.4498 | 9.8929 | 4.69% |

Table 8: Inference computational cost per image on CIFAR-100 (in GFLOPs $10^9$).

| Model | Vanilla | TTA (FARINA ET AL., 2024) | BAYESVLM |
|---|---|---|---|
| CLIP-Base | 8.83 | 687.78 | 8.84 |
| CLIP-Lage | 162.06 | 12638.00 | 162.07 |
| CLIP-Huge | 334.71 | 26098.06 | 334.72 |
| SigLIP-Base | 47.00 | 3652.05 | 47.03 |

## F.6 INTERPRETING PROBABILISTIC COSINE SIMILARITIES

We qualitatively assessed the distribution obtained by ProbCosine on a randomly selected test example from the OfficeHome clipart domain, evaluating the mean and variance of the cosine similarity under increasing corruption in both image and text domains. Text corruption was introduced by randomly replacing characters with 'x', and image corruption by randomly adding grey squares. Fig. 12 shows the mean and variance of cosine similarities as corruption increases. We observe that the expected cosine similarity generally decreases and variance increases with more corruption, indicating that our approximation effectively captures model uncertainties under distribution shift. Note that we observe a slight increase in the cosine similarity after one character has been replaced, indicating that performing predictions solely on the expected cosine similarity can be problematic. In this case, the variance over cosine similarities can capture the change in the input, highlighting the importance of capturing and propagating the model uncertainties.

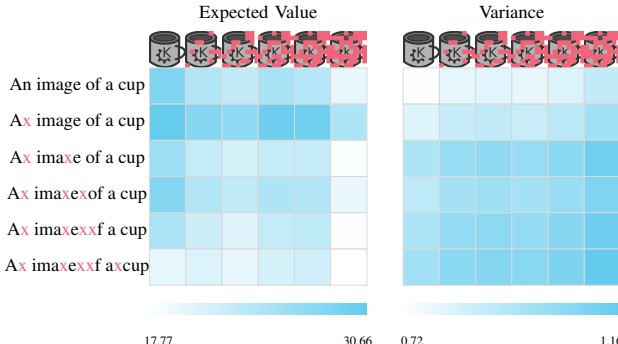

Figure 12: Illustration of ProbCosine under increasing corruption. The mean similarity decreases and variance increases with higher levels of corruption, demonstrating effective uncertainty estimation under distribution shift.

## F.7 ADDITIONAL ZERO-SHOT RESULTS

Here, we report additional zero-shot results for CLIP-huge and SigLIP-base in Table 9 and Table 10, respectively. We also report the results for applying our *ProbCosine* to the ProLIP (Chun et al., 2025) model in Table 11.

**CLIP-huge results** In Table 9, BayesVLM consistently matches or outperforms baseline methods in terms of accuracy (ACC), negative log predictive density (NLPD), and expected calibration error (ECE). We note that TTA achieves worse ACC than CLIP on CIFAR-10 and -100, which we believe is due to test-time augmentation in (Farina et al., 2024) not being optimised for small images (32x32), where the classes could become unrecognisable with additional cropping in the augmentation.

**SigLIP-base results** In Table 10, for SigLIP, BayesVLM still yields improvements on several benchmarks (despite using proxy data for Hessian estimation, as the original training set is not publicly available), demonstrating robustness to such settings. Despite mismatched training and Hessian estimation datasets, BayesVLM remains competitive, especially on CIFAR-10, UCF101, and SUN397, effectively improving calibration without sacrificing predictive performance. We note that TTA achieves better ACC on some benchmarks, which is sensible since the model will get better at predicting the correct class average with more chances (augmentations).

**ProbCosine combined with ProLIP results** In Table 11, we observe that applying ProbCosine to ProLIP improves zero-shot performance across classification benchmarks and metrics.

Table 9: **Zero-shot Results:** Quantitative evaluation of uncertainty estimation across multiple data sets in the zero-shot setting for the OpenCLIP ViT-L-14 model. Our proposed BayesVLM consistently outperforms baseline methods across accuracy (ACC, in %), negative log predictive density (NLPD), and expected calibration error (ECE, in %) metrics.

| Metrics | Methods | FLOWERS-102 | FOOD-101 | CIFAR-10 | CIFAR-100 | IMAGENET-R | UCF101 | SUN397 |
|---|---|---|---|---|---|---|---|---|
| ACC ↑ | CLIP (Radford et al., 2021) | $72.04_{\pm 0.5723}$ | $86.60_{\pm 0.2144}$ | $95.57_{\pm 0.2058}$ | $76.74_{\pm 0.4225}$ | $85.51_{\pm 0.4065}$ | $69.60_{\pm 0.7479}$ | $71.48_{\pm 0.3205}$ |
| | CLIP (temp. scaling) | $72.04_{\pm 0.5723}$ | $86.60_{\pm 0.2144}$ | $95.57_{\pm 0.2058}$ | $76.74_{\pm 0.4225}$ | $85.51_{\pm 0.4065}$ | $69.60_{\pm 0.7479}$ | $71.48_{\pm 0.3205}$ |
| | TTA (Farina et al., 2024) | $71.85_{\pm 0.5735}$ | $87.11_{\pm 0.2109}$ | $92.83_{\pm 0.2580}$ | $71.56_{\pm 0.4511}$ | $87.64_{\pm 0.3800}$ | $70.10_{\pm 0.7443}$ | $71.99_{\pm 0.3187}$ |
| | BayesVLM | $72.42_{\pm 0.4469}$ | $87.20_{\pm 0.3341}$ | $95.49_{\pm 0.2075}$ | $76.77_{\pm 0.4223}$ | $85.63_{\pm 0.3508}$ | $70.26_{\pm 0.4571}$ | $71.12_{\pm 0.4532}$ |
| NLPD ↓ | CLIP (Radford et al., 2021) | $1.75_{\pm 0.0479}$ | $0.48_{\pm 0.0083}$ | $0.15_{\pm 0.0072}$ | $0.90_{\pm 0.0178}$ | $0.58_{\pm 0.0174}$ | $1.36_{\pm 0.0391}$ | $1.02_{\pm 0.0128}$ |
| | CLIP (temp. scaling) | $1.51_{\pm 0.0378}$ | $0.47_{\pm 0.0065}$ | $0.15_{\pm 0.0056}$ | $0.87_{\pm 0.0143}$ | $0.59_{\pm 0.0144}$ | $1.20_{\pm 0.0298}$ | $0.96_{\pm 0.0098}$ |
| | TTA (Farina et al., 2024) | $1.74_{\pm 0.0472}$ | $0.49_{\pm 0.0086}$ | $0.24_{\pm 0.0080}$ | $1.15_{\pm 0.0188}$ | $0.49_{\pm 0.0160}$ | $1.34_{\pm 0.0390}$ | $1.06_{\pm 0.0133}$ |
| | BayesVLM | $1.62_{\pm 0.0335}$ | $0.45_{\pm 0.0110}$ | $0.15_{\pm 0.0061}$ | $0.87_{\pm 0.0155}$ | $0.57_{\pm 0.0132}$ | $1.22_{\pm 0.0204}$ | $0.98_{\pm 0.0154}$ |
| ECE ↓ | CLIP (Radford et al., 2021) | 9.47 | 3.07 | 0.97 | 5.73 | 2.13 | 10.72 | 8.60 |
| | CLIP (temp. scaling) | 4.90 | 3.00 | 1.35 | 2.55 | 5.21 | 3.40 | 1.48 |
| | TTA (Farina et al., 2024) | 11.96 | 3.77 | 1.21 | 3.92 | 1.92 | 11.87 | 10.53 |
| | BayesVLM | 4.66 | 1.00 | 0.62 | 1.91 | 2.15 | 5.37 | 3.89 |

Table 10: **Zero-shot results:** Quantitative evaluation of uncertainty estimation across multiple data sets in the zero-shot setting for the SigLIP-Base model (Zhai et al., 2023). Our proposed BayesVLM often performs competitively to baseline methods across accuracy (ACC, in %), negative log predictive density (NLPD), and expected calibration error (ECE, in %) metrics.

| Metrics | Methods | FLOWERS-102 | FOOD-101 | CIFAR-10 | CIFAR-100 | IMAGENET-R | UCF101 | SUN397 |
|---|---|---|---|---|---|---|---|---|
| ACC ↑ | SigLIP (Zhai et al., 2023) | $82.31_{\pm 0.4867}$ | $88.81_{\pm 0.1984}$ | $93.20_{\pm 0.2517}$ | $71.27_{\pm 0.4525}$ | $89.71_{\pm 0.3509}$ | $59.61_{\pm 0.7978}$ | $67.55_{\pm 0.3323}$ |
| | SigLIP (temp. scaling) | $82.31_{\pm 0.4867}$ | $88.81_{\pm 0.1984}$ | $93.20_{\pm 0.2517}$ | $71.27_{\pm 0.4525}$ | $89.71_{\pm 0.3509}$ | $59.61_{\pm 0.7978}$ | $67.55_{\pm 0.3323}$ |
| | TTA (Farina et al., 2024) | $82.66_{\pm 0.4828}$ | $89.24_{\pm 0.1950}$ | $87.96_{\pm 0.3254}$ | $60.95_{\pm 0.4879}$ | $90.91_{\pm 0.3320}$ | $60.14_{\pm 0.7960}$ | $66.82_{\pm 0.3342}$ |
| | BayesVLM | $82.44_{\pm 0.3805}$ | $88.84_{\pm 0.3148}$ | $93.16_{\pm 0.2524}$ | $71.22_{\pm 0.4527}$ | $89.72_{\pm 0.3037}$ | $59.69_{\pm 0.4905}$ | $67.44_{\pm 0.4686}$ |
| NLPD ↓ | SigLIP (Zhai et al., 2023) | $0.88_{\pm 0.0285}$ | $0.38_{\pm 0.0061}$ | $0.21_{\pm 0.0063}$ | $1.08_{\pm 0.0168}$ | $0.41_{\pm 0.0139}$ | $1.90_{\pm 0.0438}$ | $1.12_{\pm 0.0117}$ |
| | SigLIP (temp. scaling) | $0.84_{\pm 0.0246}$ | $0.40_{\pm 0.0054}$ | $0.22_{\pm 0.0057}$ | $1.09_{\pm 0.0152}$ | $0.43_{\pm 0.0125}$ | $1.77_{\pm 0.0376}$ | $1.12_{\pm 0.0102}$ |
| | TTA (Farina et al., 2024) | $0.85_{\pm 0.0276}$ | $0.37_{\pm 0.0064}$ | $0.36_{\pm 0.0073}$ | $1.62_{\pm 0.0209}$ | $0.37_{\pm 0.0134}$ | $1.88_{\pm 0.0430}$ | $1.18_{\pm 0.0124}$ |
| | BayesVLM | $0.86_{\pm 0.0210}$ | $0.39_{\pm 0.0091}$ | $0.21_{\pm 0.0061}$ | $1.08_{\pm 0.0163}$ | $0.41_{\pm 0.0114}$ | $1.82_{\pm 0.0248}$ | $1.12_{\pm 0.0154}$ |
| ECE ↓ | SigLIP (Zhai et al., 2023) | 4.31 | 1.66 | 0.92 | 1.97 | 1.36 | 12.72 | 3.82 |
| | SigLIP (temp. scaling) | 6.28 | 5.54 | 2.94 | 4.47 | 4.85 | 6.69 | 2.83 |
| | TTA (Farina et al., 2024) | 4.56 | 0.72 | 3.14 | 2.09 | 1.59 | 13.47 | 6.81 |
| | BayesVLM | 4.87 | 3.49 | 1.38 | 1.52 | 2.79 | 9.60 | 1.14 |

Table 11: **Can *ProbCosine* improve the zero-shot performance of pre-trained probabilistic models? Yes.** Applying *ProbCosine* to the ProLIP (Chun et al., 2025) model improves zero-shot performance across classification benchmarks and metrics.

| Metrics | Methods | FLOWERS-102 | FOOD-101 | CIFAR-10 | CIFAR-100 | IMAGENET-R | UCF101 | SUN397 |
|---|---|---|---|---|---|---|---|---|
| ACC ↑ | Mean (Chun et al., 2025) | $77.83_{\pm 0.0053}$ | $90.38_{\pm 0.0019}$ | $96.52_{\pm 0.0018}$ | $82.41_{\pm 0.0038}$ | $84.76_{\pm 0.0042}$ | $69.94_{\pm 0.0033}$ | $65.93_{\pm 0.0077}$ |
| | ProbCosine (Eq. (10)) | $77.74_{\pm 0.0053}$ | $90.35_{\pm 0.0019}$ | $96.52_{\pm 0.0018}$ | $82.48_{\pm 0.0038}$ | $84.91_{\pm 0.0041}$ | $69.99_{\pm 0.0033}$ | $65.82_{\pm 0.0077}$ |
| NLPD ↓ | Mean (Chun et al., 2025) | $1.36_{\pm 0.0411}$ | $0.33_{\pm 0.0060}$ | $0.11_{\pm 0.0052}$ | $0.64_{\pm 0.0148}$ | $0.59_{\pm 0.0164}$ | $1.05_{\pm 0.0120}$ | $1.28_{\pm 0.0316}$ |
| | ProbCosine (Eq. (10)) | $1.28_{\pm 0.0376}$ | $0.34_{\pm 0.0055}$ | $0.11_{\pm 0.0047}$ | $0.63_{\pm 0.0133}$ | $0.60_{\pm 0.0151}$ | $1.02_{\pm 0.0108}$ | $1.24_{\pm 0.0286}$ |
| ECE ↓ | Mean (Chun et al., 2025) | 5.31 | 0.79 | 0.60 | 3.38 | 1.08 | 5.99 | 7.99 |
| | ProbCosine (Eq. (10)) | 3.60 | 2.57 | 0.76 | 1.40 | 3.53 | 2.23 | 4.73 |

### F.8 ROBUSTNESS WRT TO PSEUDO-DATA COUNT

To examine the influence of the pseudo–data count $\tau$ on BayesVLM, we evaluated BayesVLM with the CLIP-Base configuration on four datasets (FLOWERS-102, FOOD-101, CIFAR-10, CIFAR-100) while varying $\tau \in \{1, 3, 5, 7, 9\}$. For each setting, we report classification accuracy (ACC ↑), negative log predictive density (NLPD ↓), and expected calibration error (ECE ↓) in Table 12.

The results show that performance is stable across the tested range of pseudo–data counts. Accuracy and calibration metrics vary only slightly with $\tau$, indicating that the method is robust to this hyperparameter. In the main manuscript, we used $\tau = 4$ as the default setting. We find that slight improvements can be observed when moving the pseudo–data count away from $\tau = 4$, but the value found on the proxy dataset ($\tau = 4$) provides reasonable performance in general.

### F.9 ROBUSTNESS WRT THE NUMBER OF NEGATIVE SAMPLES

To assess the impact of negative samples on the likelihood approximation, we vary the batch size $K \in \{32768, 8192, 2048\}$ and estimate the posterior from 1–5 random batches, reporting mean and standard deviation over five trials. Because the posterior depends on the number of negatives only through the Hessian $\boldsymbol{B}$-factor (*cf.* Eq. (5)), we present the normalised trace $\left(\frac{\text{tr}(\boldsymbol{B}_{i \times K})}{\text{tr}(\boldsymbol{B}_{5 \times K})}\right)$. Ideally, this

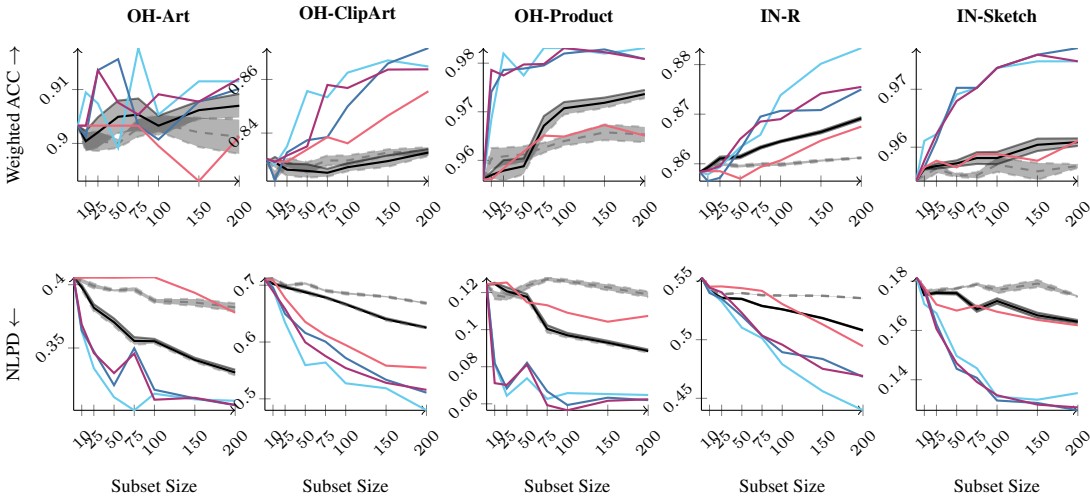

Figure 13: **Active learning results (CLIP Huge):** We present results for EPIG (——), BALD (——), Entropy (targeted) (——), Entropy (——), Random selection (targeted) (——), Random selection (- - -) on the OfficeHome dataset (OH) and ImageNet variants (IN). We observe that active learning based on our post-hoc uncertainties consistently improves upon random and entropy-based selection.

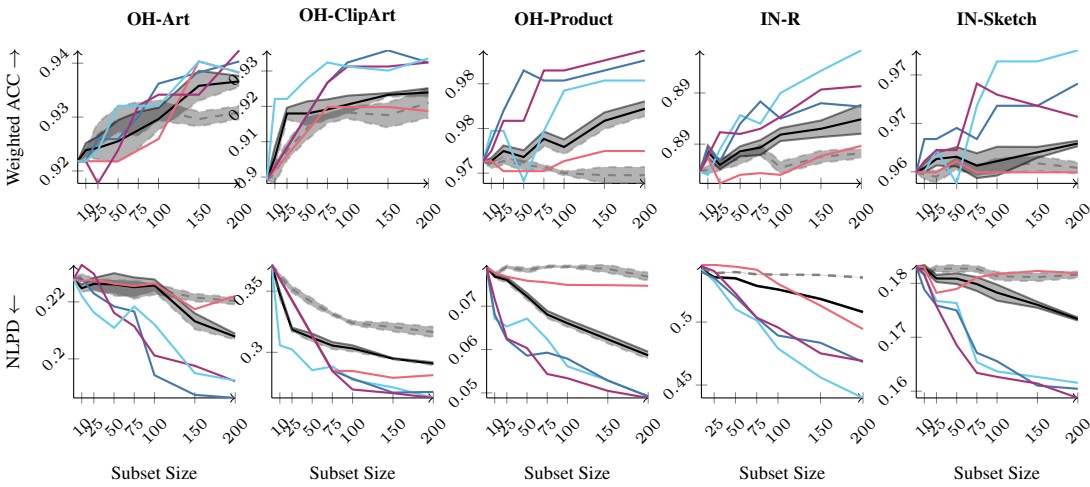

Figure 14: **Active learning results (SigLIP Base):** We present results for EPIG (——), BALD (——), Entropy (targeted) (——), Entropy (——), Random selection (targeted) (——), Random selection (- - -) on the OfficeHome dataset (OH) and ImageNet variants (IN).

ratio equals one with zero variance. Results for the image and text surrogate models are shown in Fig. 16a and Fig. 16b, respectively.

The batch size 32768 yields mean values near one for all numbers of random batches and maintains low standard deviations, indicating reliable estimates for both modalities.

## F.10 ADDITIONAL EXPERIMENTS ON CC12M

We find that BayesVLM provided robust uncertainty estimates for CLIP even when estimated on the proxy dataset, cf., Table 13.

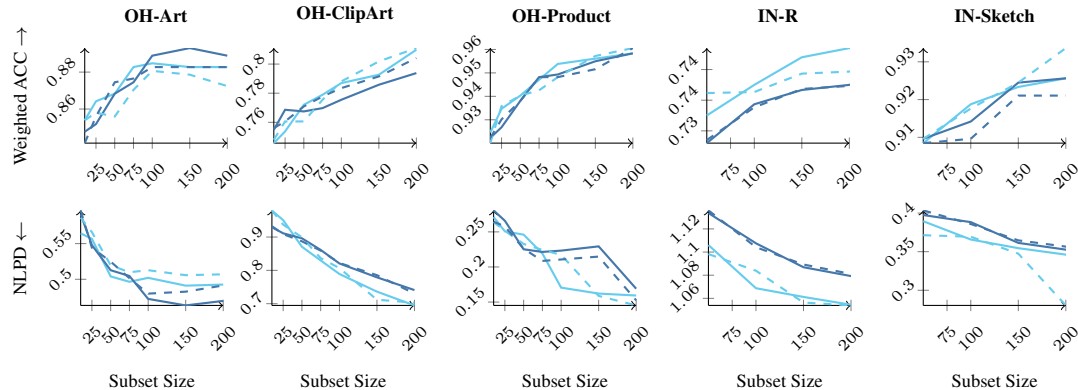

Figure 15: Ablation study on the $k$-NN distance metric, fixing the *online LA learning rate* $\gamma = 1e-4$ and the *online LA pseudo-data count* $\beta = 10$. Results are shown for EPIG (Wasserstein) (——), EPIG (cosine) (- - -), BALD (Wasserstein) (——), and BALD (cosine) (- - -). As shown in Fig. 15, Wasserstein-based $k$-NN selection demonstrates improved performance for datasets such as *OH-Art* and *IN-R*, while no clear trend is observed across the other datasets.

Table 12: Zero-shot CLIP-Base performance versus pseudo-count on four datasets.

| Pseudo Count | FLOWERS-102 | | | FOOD-101 | | | CIFAR-10 | | | CIFAR-100 | | |
|---|---|---|---|---|---|---|---|---|---|---|---|---|
| | ACC↑ | NLPD↓ | ECE↓ | ACC↑ | NLPD↓ | ECE↓ | ACC↑ | NLPD↓ | ECE↓ | ACC↑ | NLPD↓ | ECE↓ |
| 1 | 69.04±0.59 | 1.75±0.04 | 3.98 | 80.62±0.25 | 0.67±0.01 | 0.85 | 93.58±0.25 | 0.20±0.01 | 0.70 | 73.82±0.44 | 0.95±0.02 | 2.58 |
| 3 | 69.43±0.59 | 1.81±0.05 | 3.78 | 80.45±0.25 | 0.68±0.01 | 1.67 | 93.60±0.24 | 0.20±0.01 | 1.10 | 73.80±0.44 | 0.95±0.02 | 4.31 |
| 5 | 69.54±0.59 | 1.83±0.05 | 3.97 | 80.37±0.25 | 0.68±0.01 | 2.31 | 93.60±0.24 | 0.20±0.01 | 1.20 | 73.82±0.44 | 0.96±0.02 | 4.82 |
| 7 | 69.36±0.59 | 1.84±0.05 | 4.50 | 80.34±0.25 | 0.68±0.01 | 2.63 | 93.60±0.24 | 0.21±0.01 | 1.25 | 73.81±0.44 | 0.96±0.02 | 5.10 |
| 9 | 69.43±0.59 | 1.85±0.05 | 4.60 | 80.33±0.25 | 0.68±0.01 | 2.81 | 93.61±0.24 | 0.21±0.01 | 1.27 | 73.79±0.44 | 0.96±0.02 | 5.29 |

## F.11 ROBUSTNESS W.R.T. DISTRIBUTION SHIFT OF THE PROXY DATASET

Most OpenCLIP models are trained on the LAION-400M dataset, which allows us to directly estimate the Hessians on this distribution. However, some large CLIP variants are trained on closed-source data, requiring the use of a proxy dataset. As shown in Table 13, BayesVLM remains effective in a simulated closed-source setting, where we estimate the Hessians using the CC12M dataset.

To further analyse robustness, we conduct additional experiments in which we progressively distort the images of the LAION-400M dataset using different augmentations before estimating the Laplace Hessians: GRAYSCALE (interpolating between the RGB variant and the Grayscaled variant with intensity coefficients in $\{0, 0.2, 0.4, 0.6, 0.8, 1.0\}$), JPEG COMPRESSION (using increasingly lower JPEG compression quality values in $\{100, 50, 25, 10, 5, 1\}$), and GAUSSIAN BLUR (with increasing radius in $\{0, 10, 20, 30, 40, 50\}$). We provide examples in Figure 17). This setup simulates a controlled proxy-distribution shift. Importantly, all zero-shot evaluations are performed on the original (non-augmented) benchmarks.

The zero-shot results on FOOD-101, CIFAR-10, and IMAGENET-R in Tables 14 to 16 show that augmentations that alter image style while preserving semantic content lead to a gradual increase in ECE that closely tracks the augmentation intensity. In contrast, GAUSSIAN BLUR (which directly impairs object recognizability, especially at intensities 0.6 to 1.0) causes substantially stronger degradation in both ECE and NLPD. Overall, these findings indicate that BayesVLM is robust to mild stylistic shifts in the proxy data, but its performance deteriorates once the augmentations begin to compromise semantic information relevant to the model.

## F.12 ADDITIONAL ZERO-SHOT RESULTS ON ADVERSARIAL IMAGENET VARIANTS

To further assess the robustness of BayesVLM under substantial distribution shift, we evaluate the CLIP-Base model on two challenging adversarial ImageNet variants: IMAGENET-A and IMAGENET-SKETCH. These datasets introduce a severe covariate shift. IMAGENET-A contains naturally adversarial, small, or off-centre objects, whereas IMAGENET-SKETCH imposes strong stylistic changes that alter low-level statistics.

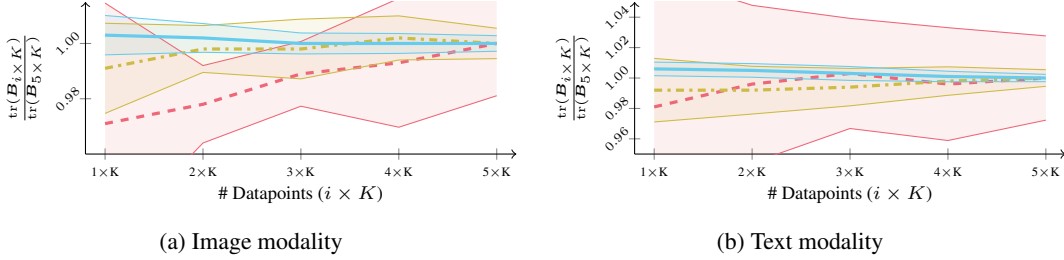

(a) Image modality              (b) Text modality

Figure 16: Normalized trace of the image Hessian $B$-factor for varying base batch sizes $K$ (2048 (- - -), 8192 (-·-), 32768 (——)) and 1–5 random batches. Error bars show $\pm 1$ std over five trials.

Table 13: **Does BayesVLM work in closed-source data settings? Yes.** With OpenCLIP ViT-B-32 trained on LAION-400M and BayesVLM estimated on the proxy dataset CC12M, we find that results are robust and show only slight degradation; statistically significant differences are **bold** ($p = 0.05$).

| Metrics | Dataset | FLOWERS-102 | FOOD-101 | CIFAR-10 | CIFAR-100 | IMAGENET-R | UCF101 | SUN397 |
|---|---|---|---|---|---|---|---|---|
| ACC ↑ | LAION-400M | **68.87**±0.4630 | 80.43±0.3968 | 93.62±0.2444 | 73.63±0.4406 | 74.45±0.4361 | 61.43±0.4868 | 66.96±0.4703 |
| | CC12M | 68.12±0.4660 | 80.35±0.3974 | 93.57±0.2453 | 73.78±0.4398 | 74.32±0.4369 | 61.46±0.4867 | 66.81±0.4709 |
| NLPD ↓ | LAION-400M | **1.73**±0.0320 | 0.68±0.0126 | 0.20±0.0067 | 0.95±0.0152 | **1.03**±0.0177 | 1.44±0.0183 | **1.12**±0.0155 |
| | CC12M | 1.77±0.0330 | 0.68±0.0129 | 0.20±0.0067 | 0.95±0.0152 | 1.03±0.0180 | 1.44±0.0185 | 1.13±0.0162 |
| ECE ↓ | LAION-400M | 4.22 | 1.69 | 0.72 | 1.92 | 1.78 | **3.77** | **2.06** |
| | CC12M | **3.84** | **0.99** | **0.70** | **1.43** | **1.39** | 3.83 | 3.89 |

Table 17 reports zero-shot accuracy, negative log predictive density (NLPD), and expected calibration error (ECE) for CLIP, temperature scaling, test-time augmentation (TTA), and BayesVLM. Across both datasets, BayesVLM does not show signs of epistemic underestimation. On IMAGENET-A, BayesVLM matches TTA in terms of calibration (ECE 0.23 vs. 0.23) while improving over CLIP. On IMAGENET-SKETCH, BayesVLM achieves the best calibration among all methods (ECE 0.08), outperforming temperature scaling and TTA. Although the NLPD on IMAGENET-A is slightly higher than with temperature scaling, BayesVLM still provides an improvement over CLIP.

Notably, TTA yields large accuracy gains, especially on IMAGENET-A, which likely arise from improved viewpoint coverage rather than improved uncertainty modelling: using 64 crops increases the likelihood of capturing an informative region of the image. In contrast, BayesVLM preserves single-pass inference efficiency while improving predictive calibration. Since BayesVLM is complementary to TTA, both approaches can be combined to obtain improvements in accuracy and uncertainty quality simultaneously.

### F.13 EMPIRICAL ASSESSMENT OF THE INDEPENDENCE ASSUMPTION

Our approach relies on an independence assumption between the parameters of the image and text projection layers when constructing the surrogate probabilistic models. This assumption follows from adopting modality-wise surrogate models satisfying the common i.i.d. assumption and is crucial for maintaining computational tractability. As detailed in App. C.1, relaxing this assumption would require estimating and storing the full joint Hessian of the projection parameters, which is computationally infeasible in practice.

To empirically assess the strength of cross-modal dependence in the loss curvature, we estimate the relative magnitude of the off-diagonal Hessian block $\boldsymbol{H}_{PQ}$ compared to the modality-specific blocks $\boldsymbol{H}_{PP}$ and $\boldsymbol{H}_{QQ}$. Specifically, we compute the ratio of Frobenius norms

$$R = \frac{\|\boldsymbol{H}_{PQ}\|_F}{\frac{1}{2}\left(\|\boldsymbol{H}_{PP}\|_F + \|\boldsymbol{H}_{QQ}\|_F\right)}, \tag{116}$$

where $R \approx 1$ would indicate strong cross-modal dependence, and smaller values correspond to weaker coupling in the curvature.

Computing the full Hessian is intractable due to its size ($\mathbb{R}^{512^2 \times 512^2}$, approximately 1.6 TiB in Float32). We therefore approximate the curvature using a Monte Carlo estimate of randomly sampled gradient components per block. The Hessian of the contrastive loss with respect to the projection

Table 14: Zero-shot CLIP-Base performance on FOOD-101 versus augmentation intensity for three augmentation types.

| | GRAYSCALE | | | | JPEG COMPRESSION | | | | GAUSSIAN BLUR | | |
|---|---|---|---|---|---|---|---|---|---|---|---|
| Intensity | ACC↑ | NLPD↓ | ECE↓ | Quality | ACC↑ | NLPD↓ | ECE↓ | Radius | ACC↑ | NLPD↓ | ECE↓ |
| 0.0 | 80.44±0.397 | 0.68±0.013 | 1.69 | 100 | 80.44±0.397 | 0.68±0.013 | 1.69 | 0 | 80.44±0.397 | 0.68±0.013 | 1.69 |
| 0.2 | 80.45±0.397 | 0.68±0.013 | 1.71 | 50 | 80.48±0.396 | 0.68±0.012 | 2.21 | 10 | 80.61±0.395 | 0.68±0.012 | 2.38 |
| 0.4 | 80.43±0.397 | 0.68±0.013 | 1.73 | 25 | 80.54±0.396 | 0.68±0.012 | 2.26 | 20 | 80.58±0.396 | 0.68±0.012 | 3.25 |
| 0.6 | 80.48±0.396 | 0.68±0.013 | 1.85 | 10 | 80.59±0.396 | 0.68±0.012 | 2.23 | 30 | 80.48±0.396 | 0.69±0.011 | 5.69 |
| 0.8 | 80.54±0.396 | 0.68±0.012 | 2.06 | 5 | 80.58±0.396 | 0.68±0.012 | 2.74 | 40 | 80.43±0.397 | 0.72±0.011 | 8.93 |
| 1.0 | 80.56±0.396 | 0.68±0.012 | 2.62 | 1 | 80.51±0.396 | 0.68±0.012 | 4.38 | 50 | 80.40±0.397 | 0.75±0.011 | 11.52 |

Table 15: Zero-shot CLIP-Base performance on CIFAR-10 versus augmentation intensity for three augmentation types.

| | GRAYSCALE | | | | JPEG COMPRESSION | | | | GAUSSIAN BLUR | | |
|---|---|---|---|---|---|---|---|---|---|---|---|
| Intensity | ACC↑ | NLPD↓ | ECE↓ | Quality | ACC↑ | NLPD↓ | ECE↓ | Radius | ACC↑ | NLPD↓ | ECE↓ |
| 0.0 | 93.61±0.245 | 0.203±0.007 | 0.72 | 100 | 93.61±0.245 | 0.203±0.007 | 0.72 | 0 | 93.61±0.245 | 0.203±0.007 | 0.72 |
| 0.2 | 93.61±0.245 | 0.203±0.007 | 0.72 | 50 | 93.60±0.245 | 0.203±0.007 | 1.02 | 10 | 93.57±0.245 | 0.203±0.007 | 0.72 |
| 0.4 | 93.61±0.245 | 0.203±0.007 | 0.71 | 25 | 93.59±0.245 | 0.203±0.007 | 0.96 | 20 | 93.57±0.245 | 0.203±0.007 | 0.65 |
| 0.6 | 93.61±0.245 | 0.203±0.007 | 0.68 | 10 | 93.59±0.245 | 0.203±0.007 | 0.92 | 30 | 93.58±0.245 | 0.204±0.007 | 0.84 |
| 0.8 | 93.58±0.245 | 0.203±0.007 | 0.69 | 5 | 93.57±0.245 | 0.203±0.007 | 0.89 | 40 | 93.58±0.245 | 0.207±0.007 | 1.88 |
| 1.0 | 93.58±0.245 | 0.203±0.007 | 0.80 | 1 | 93.60±0.245 | 0.205±0.007 | 1.35 | 50 | 93.58±0.245 | 0.212±0.007 | 2.73 |

parameters is approximated via the empirical Fisher, $\boldsymbol{H} \approx \mathbb{E}[gg^\top]$, where $g$ denotes the per-batch gradient. While the empirical Fisher is less accurate than the GGN approximation used in the main method, it provides a computationally feasible proxy for estimating relative curvature magnitudes.

For each Hessian block, we estimate the Frobenius norm by averaging the squared magnitudes of the sampled entries. The ratio $R$ is computed across three random seeds and multiple sample sizes. The results are reported in Table 18.

The results indicate only moderate cross-modal dependence in the Hessian (loss curvature). Hence, while secondary interactions between modalities exist, the dominant curvature structure is captured by the modality-wise blocks. This provides empirical support for the independence approximation adopted in BayesVLM. Importantly, the approximation does not eliminate the multimodal coupling learned during pre-training. The curvature is evaluated at the MAP solution obtained via multimodal contrastive learning, and the resulting posterior still reflects the cross-modal alignment encoded in the model. In practice, our method replaces the exact posterior with a tractable approximation that preserves the model's principal geometric structure while discarding interaction terms of moderate magnitude. Finally, the entanglement of modalities in the resulting uncertainties can be qualitatively observed in App. F.6, where corruption of one modality affects the predictive uncertainty of the joint model, indicating that cross-modal dependencies remain reflected in the predictive distribution.

Table 16: Zero-shot CLIP-Base performance on IMAGENET-R versus augmentation intensity for three augmentation types.

| | GRAYSCALE | | | | JPEG COMPRESSION | | | | GAUSSIAN BLUR | | |
|---|---|---|---|---|---|---|---|---|---|---|---|
| Intensity | ACC↑ | NLPD↓ | ECE↓ | Quality | ACC↑ | NLPD↓ | ECE↓ | Radius | ACC↑ | NLPD↓ | ECE↓ |
| 0.0 | 74.49±0.436 | 1.031±0.018 | 1.71 | 100 | 74.48±0.436 | 1.031±0.018 | 1.71 | 0 | 74.49±0.436 | 1.031±0.018 | 1.71 |
| 0.2 | 74.51±0.436 | 1.031±0.018 | 1.70 | 50 | 74.53±0.436 | 1.031±0.017 | 2.05 | 10 | 74.67±0.435 | 1.029±0.017 | 2.04 |
| 0.4 | 74.51±0.436 | 1.031±0.018 | 1.61 | 25 | 74.56±0.436 | 1.031±0.017 | 1.93 | 20 | 74.69±0.435 | 1.031±0.017 | 3.01 |
| 0.6 | 74.48±0.436 | 1.030±0.018 | 1.72 | 10 | 74.63±0.436 | 1.030±0.018 | 1.94 | 30 | 74.76±0.434 | 1.046±0.016 | 5.70 |
| 0.8 | 74.47±0.436 | 1.030±0.018 | 1.89 | 5 | 74.47±0.436 | 1.030±0.018 | 1.89 | 40 | 74.71±0.435 | 1.077±0.016 | 9.28 |
| 1.0 | 74.56±0.436 | 1.029±0.018 | 1.95 | 1 | 74.72±0.435 | 1.032±0.017 | 3.23 | 50 | 74.75±0.434 | 1.109±0.015 | 12.20 |

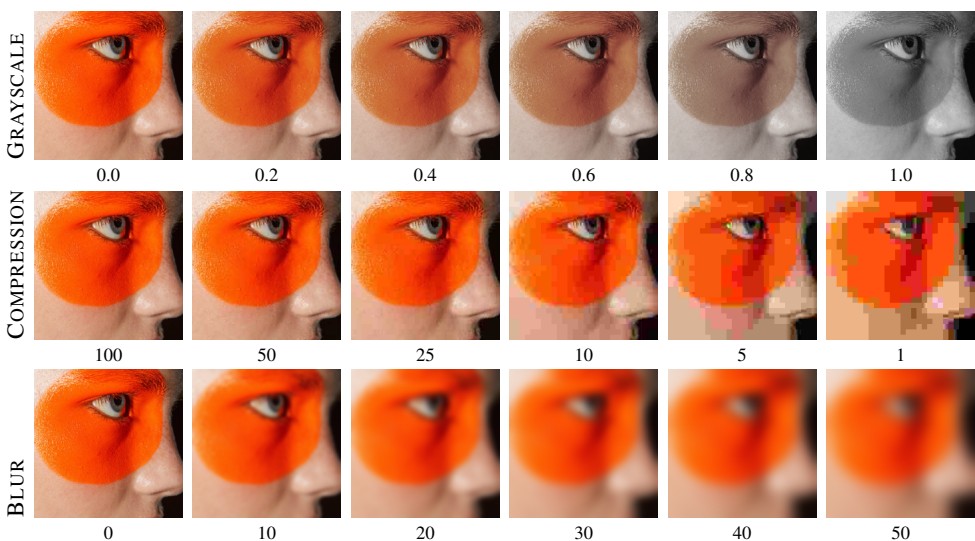

Figure 17: Visualisation of proxy-data augmentations across degradation levels. Rows correspond to augmentation types (GRAYSCALE, JPEG COMPRESSION, GAUSSIAN BLUR). Columns show increasing degradation severity, parameterised by intensity, JPEG quality, or blur radius, respectively.

Table 17: Zero-shot performance on IMAGENET-A and IMAGENET-SKETCH.

| | IMAGENET-A | | | IMAGENET-SKETCH | | |
|---|---|---|---|---|---|---|
| Method | Acc (%) ↑ | NLPD ↓ | ECE ↓ | Acc (%) ↑ | NLPD ↓ | ECE ↓ |
| CLIP (Radford et al., 2021) | 25.87±0.4379 | 3.77±0.0312 | 0.33 | 50.98±0.4999 | 2.37±0.0303 | 0.16 |
| CLIP (temp. scaling) | 25.87±0.4379 | 3.22±0.0226 | 0.18 | 50.98±0.4999 | 2.29±0.0286 | 0.13 |
| TTA (Farina et al., 2024) | 38.67±0.4870 | 2.77±0.0279 | 0.23 | 52.40±0.4994 | 2.36±0.0300 | 0.15 |
| BayesVLM | 26.67±0.4422 | 3.36±0.0257 | 0.23 | 50.59±0.5000 | 2.24±0.0259 | 0.08 |

Table 18: Estimated curvature ratio $R$ for different numbers of sampled gradient components.

| Number of random samples | $R$ |
|---|---|
| 50k | $0.363 \pm 0.002$ |
| 100k | $0.374 \pm 0.035$ |

