# OpenReview forum: "Post-hoc Probabilistic Vision-Language Models"
_ICLR.cc/2026/Conference — ICLR 2026 Poster_

### Official Review · Reviewer_88Gp · 2025-10-18

**Soundness:** 4
**Presentation:** 4
**Contribution:** 3
**Rating:** 6
**Confidence:** 4

**Summary:**

This paper proposes BayesVLM, a post-hoc Bayesian uncertainty estimation method for large-scale vision-language models (VLMs) such as CLIP and SigLIP. The key insight is to apply a Laplace approximation over the last linear projection layers of pre-trained VLMs to obtain Gaussian posteriors over the parameters. This gives a local second-order uncertainty around the learned weights, thereby inducing probabilistic embeddings for both modalities. This also allows the model to analytically compute a distribution over cosine similarities, leading to uncertainty estimates without any retraining, architectural modifications, or additional data. The authors further propose ProbCosine, an analytical approximation for the mean and variance of the cosine similarity under this Bayesian formulation, avoiding costly Monte Carlo sampling. Empirically, the paper demonstrates that BayesVLM yields better-calibrated uncertainty estimates in zero-shot classification and improves sample efficiency in active learning, with minimal computational overhead.

**Strengths:**

- **Novelty and conceptual clarty:** The idea of using a post-hoc Laplace approximation on pre-trained VLMs to obtain Bayesian uncertainty estimates is novel and conceptually elegant. The formulation is well-grounded in Bayesian deep learning literature and bridges it with the practical need for scalable uncertainty estimation in large multimodal models.
- **Practicality and scalability:** The method is traning-free and model-agnostic, requiring only access to the final projection weights and not the original training pipeline. The approach introduces negligible computational overhead (<5% runtime, <0.11% GFLOPs), making it attractive for large-scale or resource-constrained deployments.
- **Analytical Contribution (ProbCosine)** The derivation of a closed-form Gaussian approximation for cosine similarity uncertainty is elegant, interpretable, and computationally efficient. This is a non-trivial mathematical contribution that generalizes beyond this specific application.
- **Strong Empirical Evaluation:** Comprehensive experiments across multiple datasets (Flowers-102, CIFAR, Food-101, ImageNet-R, UCF101, SUN397) and tasks (zero-shot classification, active learning). The paper also shows comparison with strong baselines: temperature scaling, test-time augmentation, and probabilistic embedding models. Results show consistent improvement in calibration metrics (ECE) and competitive or improved accuracy.
- **Robustness and Generality:** The method performs well even when the Hessian is estimated on proxy datasets, showing robustness to missing original training data. The paper provides ablations on hyperparameters (τ, λ) and data subset sizes for Hessian estimation.

Overall, the main paper is well-written and complemented by detailed appendices with derivations, algorithms, and hyperparameters.
The authors mention plans to release code, aiding reproducibility.

**Weaknesses:**

- **Assumption of Independence (Modality Factorization)** The method assumes independence between image and text modalities (P ⊥⊥ Q) to enable tractable posterior estimation. Although the authors justify this as a local approximation around the MAP estimate, it weakens the theoretical rigor since VLMs are inherently cross-modal. Also, the impact of this assumption on downstream uncertainty fidelity is not fully explored. Can the authors comment on this?
- **Limited Scope of Bayesian treatment:** The Laplace approximation is applied only to the final projection layers, while the encoders remain deterministic. This limits the expressiveness of the uncertainty model, potentially underestimating uncertainty arising from deep feature extraction. Although, I understand  that an approximation to the entire encoder layers would be a huge sacrifice for efficiency/practicality, could the authors comment on any other plausible reasons why/why not to consider this?
- **Evaluation of Uncertainty Quality:** While calibration metrics (ECE, NLPD) are reported, it would strengthen the paper to show qualitative examples or diagnostics of uncertainty estimates (e.g., in failure or domain shift cases).
- **Active Learning Evaluation Setup:** The active learning results are promising but limited to relatively small-scale benchmarks (OfficeHome, ImageNet variants). It is unclear how the approach would scale to real-world large-scale active learning pipelines or multimodal retrieval settings.
- **Comparative baselines:** The paper primarily compares with test-time augmentation and temperature scaling, but omits comparison with recent Bayesian adapters or probabilistic prompt tuning methods (e.g., BayesAdapter [1]). A direct comparison could more clearly position BayesVLM in the landscape of uncertainty-aware VLMs.
- **Interpretability of Uncertainty:** Although the authors claim interpretability under corruptions, the paper could better visualize what kind of uncertainties are captured (epistemic vs. aleatoric). Without this, it’s difficult to assess if BayesVLM truly models epistemic uncertainty or merely produces smoother predictions.

**References**

[1] Morales-Alvarez, Pablo et al. “BayesAdapter: enhanced uncertainty estimation in CLIP few-shot adaptation.” (2024)

**Questions:**

- **Direct baselines.** Can the authors comment on adding ProLIP (frozen or few-shot adapted) [1] and CLAP4CLIP (single-task setting) [2] as baselines on selected zero-shot datasets to validate BayesVLM’s calibration per FLOP gains?
- **Data requirements:** When estimating the Hessian with proxy datasets, how do you ensure alignment between proxy and original data distributions? Could this introduce bias in uncertainty estimates?
- **On scalability:** What is the memory or compute bottleneck when applying BayesVLM to very large projection layers (e.g. CLIP large)? Could low-rank approximations be used instead?

**References:**

[1] Chun, Sanghyuk et al. “Probabilistic Language-Image Pre-Training.” ICLR 2025.

[2] Jha, Saurav et al. “CLAP4CLIP: Continual Learning with Probabilistic Finetuning for Vision-Language Models.” NeurIPS 2024.

---

> ### Author Response · Authors · 2025-11-20
>
> We thank the reviewer for the extensive review and the time and effort put into reviewing our submission. We will respond to the comments and questions raised in the review individually below.
>
> **Comment 1:** _The method assumes independence between image and text modalities [...] to enable tractable posterior estimation. [...] the impact of this assumption on downstream uncertainty fidelity is not fully explored. Can the authors comment on this?_
>
> **Response:** As pointed out by the reviewer, the independence assumption, which follows from building surrogate models that satisfy the common iid assumption, is crucial to obtaining a computationally tractable approach in our work. We detailed in Appendix C.1 (What happens without the iid assumption) an explanation of the effect if this assumption is not made. Specifically, in Eqs. 30 -- 33 we show that even only estimating the Hessian matrix used for the Laplace approximation is intractable in this setting (cannot be computed in a reasonable amount of time).
>
> However, we would like to point out that we have found that BayesVLM improves model calibration on zero-shot learning tasks (cf. Section 4.1) and that the uncertainties obtained by BayesVLM help reduce overconfident predictions before adaptation and select support set candidates more effectively in active learning settings (cf. Section 4.2). Note that we also qualitatively observed that BayesVLM effectively reduces overconfident predictions and substantially reduces the error in the predictions after active learning (cf. Figure 2). Thus indicating the effectiveness of our Bayesian formulation.
>
> As suggested by Reviewer 9dNF, we conducted an additional empirical experiment to assess the strength of the cross-modal dependence in the posterior distribution and Hessian matrix. *We provide the full experimental setup and results in our response to Reviewer 9dNF (Question 1)*.
>
> The results indicate moderate cross-modal dependence in the Hessian (loss curvature). Hence, we can expect that only a moderate amount of secondary interactions between both modalities are disregarded in our approximation, while the primary interactions are captured. Capturing those additional secondary interactions in a computationally feasible manner is an interesting but challenging future direction.
>
> We want to emphasise that our approximation does not eliminate the multimodal coupling learned during pre-training. This is because the curvature is evaluated at the MAP solution obtained via multimodal contrastive learning. Therefore, the local posterior used by BayesVLM still reflects the cross-modal alignment encoded in the encoder. In practice, our approximation replaces an intractable exact posterior with a tractable one that retains the key geometric structure of the model while discarding interaction terms that, according to our experiment, have only a small or moderate effect.
>
> Therefore, even though we have one independent surrogate model for each modality, the local posterior (uncertainty) estimates indirectly depend on all modalities. Thus, providing a practical and reasonable solution to the problem.

---

> > ### Author Response · Authors · 2025-11-20
> >
> > **Comment 2:** _Although, I understand that an approximation to the entire encoder layers would be a huge sacrifice for efficiency/practicality, could the authors comment on any other plausible reasons why/why not to consider this?_
> >
> > **Response:** We agree that full Bayesianization of encoders could, in principle, capture more epistemic uncertainties. And doing so would be an interesting direction in the future. However, this requires storing and inverting high-dimensional covariance structures, which remains infeasible even with structured approximations like KFAC at the moment. Moreover, making the whole network Bayesian has been shown to be theoretically problematic [Sharma et al., 2023] and last-layer approximations [Kristiadi et al., 2020] or using only a subset of weights [Daxberger et al., 2021] often provide an efficient and effective solution.
> >
> > In VLMs, the projection layers form the bottleneck where uncertainty can be injected with tractable closed-form computation. Moreover, it provides a natural interpretation as uncertainty over the alignments of the features of each modality in the joint concept space. Thus, providing an appropriate entry point when Bayesianising a VLM.
> >
> > When storing the KFAC approximated Hessian matrices for a CLIP-Large model for every layer, we would require a memory overhead of $\approx 4.22 \text{GiB}$ for the image encoder and $\approx 1.2 \text{GiB}$ for the text encoder, assuming Float32. Note this overhead would be in addition to the memory requirements of storing the model parameters ($\approx 1.6 \text{GiB}$ and $\approx 0.3 \text{GiB}$, respectively) and the memory overhead related to computing the forward pass. While a full Bayesianisation of the CLIP model is possible, it would result in both computational overhead and non-negligible memory overhead. As our work aims to provide an efficient and practical approach with little to no overhead, we focused on Bayesianising only the projection layers. However, note that if one aims to treat the full model in a Bayesian way, the aforementioned computational and memory overhead may be mitigated by combining low-rank approximations of the KFAC factors with a linearised Laplace approximation. However, doing so is non-trivial and we consider this to be future work.
> >
> > [Sharma et al., 2023] Sharma et al. "Do Bayesian neural networks need to be fully stochastic?". In AISTATS 2023.
> >
> > [Kristiadi et al., 2020] Kristiadi et al. "Being Bayesian, Even Just a Bit, Fixes Overconfidence in ReLU Networks". In NeurIPS 2020.
> >
> > [Daxberger et al., 2021] Daxberger et al. "Bayesian deep learning via subnetwork inference". In ICML 2021.
> >
> > ---
> >
> > **Comment 3:** _[...] it would strengthen the paper to show qualitative examples or diagnostics of uncertainty estimates (e.g., in failure or domain shift cases)._
> >
> > **Response:** We qualitatively assessed the uncertainty estimates obtained by BayesVLM under different levels of corruption in either the text or the image domain. The qualitative results are given in Appendix F.6 and Figure 11 and, unfortunately, had to be omitted in the main text due to space constraints. We used the additional page of the camera-ready version to move the qualitative assessment into Section 4.3 in the main text, as the results provide insightful observations.
> >
> > In addition, we want to highlight our qualitative assessment of the predictive uncertainties before and after active learning (see Figure 2). In particular, we observe that BayesVLM results in less overconfident predictions before active learning (compare quadrand b and a) and substantially reduces errors in the predictions after active learning, compared to the fine-tuned CLIP model. These results indicate the benefits of using a Bayesian formulation when actively reducing uncertainties in VLMs.
> >
> > We also evaluate BayesVLM on ImageNet-A and ImageNet-Sketch, two natural distribution-shift benchmarks. As requested by reviewer 9dNF, we added these experiments to assess calibration robustness under significant distribution shift. Across both datasets, BayesVLM maintains or improves calibration relative to CLIP (and is competitive with post-hoc baselines), indicating that the method does not systematically underestimate uncertainty under distribution shift (see Appendix F.12).

---

> > > ### Author Response · Authors · 2025-11-20
> > >
> > > **Comment 4:** _It is unclear how the approach would scale to real-world large-scale active learning pipelines or multimodal retrieval settings._
> > >
> > > **Response:** We appreciate the reviewer’s comment. Our work focuses on efficient post-hoc uncertainty quantification in VLMs, with active learning serving as one application where epistemic uncertainty is important. Because BayesVLM adds minimal overhead for uncertainty estimation, it can in principle be combined with any Bayesian active learning method, as illustrated through our use of BALD and EPIG.
> > >
> > > We agree that scaling active learning to big data settings is an interesting research direction, but is is outside the primary scope of our work, as the main contribution of our paper is on post-hoc uncertainty estimation and active learning is an application in which accurate epistemic uncertainties are key. We believe that our method can be an enabler for efficient and scalable active learning and research in this promising future direction.
> > >
> > > In practice the support set can be very large, making a linear traversal computationally infeasible to compute the EPIG/BALD score for every datapoint. A standard solution is to reduce the candidate set via targeted downsampling, for example, using approximate nearest-neighbor retrieval (e.g., FAISS [Douze et al., 2024]) to identify a small, relevant subset of points on which to compute acquisition scores.
> > >
> > > [Douze et al., 2024] Douze et al., "The Faiss library." arXiv preprint arXiv:2401.08281, 2024.
> > >
> > > ---
> > >
> > > **Comment 5:** _[...] comparison with recent Bayesian adapters or probabilistic prompt tuning methods (e.g., BayesAdapter [1])._
> > >
> > > **Response:** Even though interesting, BayesAdapter and similar adapter-based methods cannot be applied in zero-shot settings as they require training (or fine-tuning) on the target domain. For example, BayesAdapter formulates a classification-based variational objective that requires prior knowledge of the exact set of classes in the downstream task. In zero-shot settings, the classes are not known a priori. In fact, our work is, to the best of our knowledge, the only existing technique for VLMs that is both principled and applicable to zero-shot settings.
> > >
> > > ---
> > >
> > > **Comment 6:** _[...] what kind of uncertainties are captured (epistemic vs. aleatoric)._
> > >
> > > **Response:** Our approach captures the total uncertainty (epistemic and aleatoric) and the epistemic part can be obtained, for example, by using an information-theorethical approach like the expected information gain [Kendall & Gal, 2017], $EIG = H(\mathbb{E}_{\theta}[ p(y \mid x_{\text{txt}}, \theta)]) - \mathbb{E}_{\theta}[ H(p(y \mid x_{\text{txt}}, \theta))]$, where the epistemic part (EIG) is obtained by subtracting the average uncertainty of each individual model instantiation (containing the aleatoric uncertainy) from the overall predictive uncertainty. However, we want to point out that this information-theoretical formulation is only one possibility and formalising how epistemic uncertainties should be defined is an active field of research [Bickford Smith et al., 2025].
> > >
> > > In our active learning experiment, we assessed the performance of BayesVLM for two popular Bayesian active learning score functions, BALD and EPIG. Both score functions utilise a notion of epistemic uncertainties. In fact, BALD is equivalent to the information-theoretical approach by [Kendall & Gal, 2017] illustrated above.
> > >
> > > In addition, in our qualitative study in Section F.6 and Figure 11 in the Appendix, we can observe that the uncertainty estimated by BayesVLM can capture distribution shift (increasing corruption level) in the inputs. This increase in uncertainty is due to epistemic uncertainties related to the distribution shift.
> > >
> > > [Kendall & Gal, 2017] Kendall & Gal, "What uncertainties do we need in Bayesian deep learning for computer vision?". In NIPS 2017.
> > >
> > > [Bickford Smith et al., 2025] Bickford Smith et al. "Rethinking Aleatoric and Epistemic Uncertainty". In ICML 2025.

---

> > > > ### Author Response · Authors · 2025-11-20
> > > >
> > > > **Question 1:** _Can the authors comment on adding ProLIP (frozen or few-shot adapted) and CLAP4CLIP (single-task setting) as baselines on selected zero-shot datasets to validate BayesVLM’s calibration per FLOP gains?_
> > > >
> > > > **Response:** We appreciate the reviewer's suggestion. In our paper, we presented additional results on PCME++ in the main text (Table 2) and ProLIP in the appendix (Table 11). Unfortunately, it was not possible to present both PCME++ and ProLIP results in the main text. We revised the presentation in the main text to increase the visibility of those results.
> > > >
> > > > Regarding CLAP4CLIP: even in the single-task setting, it is not directly comparable to BayesVLM. CLAP4CLIP requires explicit knowledge of the downstream task’s class set and finetuning on labeled examples from each class in order to construct its task-specific probabilistic adapters. Consequently, before finetuning, its predictions reduce to those of the underlying CLIP model, and it does not provide meaningful zero-shot uncertainty estimates.
> > > >
> > > > In contrast, BayesVLM is task-agnostic: it does not require knowledge of downstream classes and also no labeled samples from the downstream task. Its posterior uncertainty is estimated using a subset of the pretraining data (or an appropriate proxy dataset), enabling true zero-shot Bayesian inference.
> > > >
> > > > ---
> > > >
> > > > **Question 2:** _When estimating the Hessian with proxy datasets, how do you ensure alignment between proxy and original data distributions? Could this introduce bias in uncertainty estimates?_
> > > >
> > > > **Response:** We do not enforce alignment between proxy and original data distributions. And indeed, estimating the Hessian on a proxy dataset with a strong domain shift will likely introduce errors into the uncertainty estimates.
> > > >
> > > > To systematically quantify this effect, we conducted an additional analysis (Appendix F.11), following the suggestion of Reviewer 9dNF. In this experiment, we progressively distorted images of the LAION-400M dataset using different augmentations before estimating the Hessians matrix of the Laplace approximation. We then evaluated the resulting model with respect to its zero-shot performance on FOOD-101. The results show that BayesVLM remains robust under mild stylistic shifts, while stronger deviations (particularly those that compromise semantic content) lead to increased calibration error and higher NLPD. This confirms that proxy mismatch can introduce bias, but also demonstrates our method’s tolerance to moderate distribution differences.
> > > >
> > > > The table below shows zero-shot results when the proxy data is perturbed with increasing augmentation strength: B/W (progressive desaturation to full grayscale) and Blur (increasing Gaussian blur). See Appendix F.11 for augmentation visualisations and additional results.
> > > >
> > > > | Intensity | B/W ACC ↑ | B/W NLPD ↓ | B/W ECE ↓ | Blur ACC ↑ | Blur NLPD ↓ | Blur ECE ↓ |
> > > > |-----------|------------------|------------------|------------------|-----------------------|------------------------|-----------------------|
> > > > | 0.0 | 80.44±0.40 | 0.68±0.01 | 1.69 | 80.44±0.40 | 0.68±0.01 | 1.69 |
> > > > | 0.2 | 80.45±0.40 | 0.68±0.01 | 1.71 | 80.61±0.40 | 0.68±0.01 | 2.38 |
> > > > | 0.4 | 80.43±0.40 | 0.68±0.01 | 1.73 | 80.58±0.40 | 0.68±0.01 | 3.25 |
> > > > | 0.6 | 80.48±0.40 | 0.68±0.01 | 1.85 | 80.48±0.40 | 0.69±0.01 | 5.69 |
> > > > | 0.8 | 80.54±0.40 | 0.68±0.01 | 2.06 | 80.43±0.40 | 0.72±0.01 | 8.93 |
> > > > | 1.0 | 80.56±0.40 | 0.68±0.01 | 2.62 | 80.40±0.40 | 0.75±0.01 | 11.52 |

---

> > > > > ### Author Response · Authors · 2025-11-20
> > > > >
> > > > > **Question 3:** _What is the memory or compute bottleneck when applying BayesVLM to very large projection layers (e.g. CLIP large)? Could low-rank approximations be used instead?_
> > > > >
> > > > > **Response:** The main memory bottleneck arises during estimation of the Kronecker-factored Hessian approximation, specifically in computing and storing the KFAC factors $\boldsymbol{A}$ and $\boldsymbol{B}$, especially when the pre- or post-projection embeddings are very large. In our setting, the KFAC factors correspond to empirical second-moment matrices of the layer inputs and output gradients, with shapes $p \times p$ and $d \times d$, respectively. Although KFAC does not require forming the full Hessian, it does require maintaining these Kronecker factors, and their size scales quadratically with the embedding dimensions.
> > > > >
> > > > > For models such as CLIP-Large (where $p = d = 768$), computing $\boldsymbol{A}$ involves summing outer products of the pre-projection embeddings $\phi(x_i)\phi(x_i)^\top$. Computing $\boldsymbol{B}$ appears costly on first thought because the InfoNCE loss Hessian scales quadratically with batch size (typically $\approx 32k$). But by exploiting its low rank structure, we find an efficient expression that is linear in batch size and allows for efficient chunked computation, after which only a $d\times d$ factor has to be stored. We provide technical details in Section C.2.2 in the Appendix.
> > > > >
> > > > > Since these KFAC factors $\boldsymbol{A}$ and $\boldsymbol{B}$ are _only computed once_ as a preprocessing step, the runtime cost during inference is negligible. For example, for CLIP-Huge, we found that BayesVLM results in a relative increase of the runtime (in seconds) of only 0.35%. Further, in Table 8 in Appendix F.5, we show runtime comparisons to test-time augmentation (TTA), which results in an 80-fold runtime increase while BayesVLM has a runtime almost identical to CLIP.
> > > > >
> > > > > However, the quadratic memory scaling in $p$ and $d$ could become a bottleneck for very large projection layers. In such cases, a low-rank approximation of $A$ and/or $B$ can be beneficial to reduce memory, reducing storage requirements from $\mathcal{O}(p^2 + d^2)$ to $\mathcal{O}(pk + dk)$ where $k$ is the chosen rank that trades off memory usage against posterior fidelity.

---

> > > > > > ### Author Response · Authors · 2025-11-26
> > > > > >
> > > > > > We hope that our rebuttal addressed the reviewers’ questions about our work. Should you have any further questions, we will be more than happy to answer them during the ongoing discussion phase.

---

### Official Review · Reviewer_9dNF · 2025-10-29

**Soundness:** 3
**Presentation:** 3
**Contribution:** 3
**Rating:** 6
**Confidence:** 4

**Summary:**

The paper proposes BayesVLM, a post-hoc, training-free method to endow pre-trained VLMs (CLIP, SigLIP) with epistemic+aleatoric uncertainty. The approach fits a Laplace approximation over the final projection layers of the image and text encoders, assumes modality-wise independence to get tractable posteriors, and then analytically propagates uncertainty to cosine similarities via a Gaussian (“ProbCosine”) moment approximation. The method improves calibration (ECE) in zero-shot classification.

**Strengths:**

1. The proposed method is training-free and works with off-the-shelf CLIP-like models.

2. The proposed method provides overall better calibration performance on the evaluated benchmarks with small computation overhead.

3. Proxy-data robustness: Hessians from CC12M still work decently for CLIP trained on LAION-400M.

**Weaknesses:**

1. Treating image and textual modalities as independent is the core approximation. While the authors justify it via local post-hoc around MAP, it remains a potential mismatch for strongly coupled modalities; discussion is present but could use a stronger empirical illustration.

2. The proposed method puts all epistemic uncertainty in the final projections. This may under-estimate uncertainty on heavy distribution shifts.

3. For closed-source models, the approach needs proxy data; results are promising but slight degradations exist. A sensitivity analysis to proxy domain mismatch (e.g., caption style, image domain) would help.

**Questions:**

1. Please provide some empirical results on the independence of two modalities.

2. For ImageNet setup, please include more ImageNet variants such as ImageNet-A.

---

> ### Author Response · Authors · 2025-11-20
>
> First, we would like to thank the reviewer for their time and effort put into reviewing our submission. We will respond to the comments and questions raised in the review individually below.
>
> **Comment 1:** _For closed-source models, the approach needs proxy data; results are promising but slight degradations exist. A sensitivity analysis to proxy domain mismatch (e.g., caption style, image domain) would help._
>
> **Response:** We want to thank the reviewer for this suggestion. To assess the sensitivity of BayesVLM to proxy-domain mismatch, we conducted a controlled analysis in which we progressively distorted images of the LAION-400M dataset using different augmentations before estimating the Hessians matrix of the Laplace approximation. We then evaluated the resulting model with respect to its zero-shot performance on FOOD-101. This allows us to emulate different forms of distribution shift while keeping the evaluation set fixed. As detailed in Appendix F.11, BayesVLM remains stable under mild stylistic shifts (e.g., grayscale or mild decreases of JPEG compression quality) and degrades slightly under stronger stylistic shifts. Still, performance drops more noticeably once augmentations begin to remove or obscure semantic information (e.g., strong blur). This confirms that proxy mismatch can introduce bias, but also demonstrates our method’s tolerance to moderate distribution differences.
>
> The table below shows zero-shot results on FOOD-101 when the proxy data is perturbed with increasing augmentation strength: B/W (progressive desaturation to full grayscale) and Blur (increasing Gaussian blur). See Appendix F.11 for augmentation visualisations and additional results on CIFAR-10 and ImageNet-R.
>
>
> | Intensity | B/W ACC ↑ | B/W NLPD ↓ | B/W ECE ↓ | Blur ACC ↑ | Blur NLPD ↓ | Blur ECE ↓ |
> |-----------|------------------|------------------|------------------|-----------------------|------------------------|-----------------------|
> | 0.0 | 80.44±0.40 | 0.68±0.01 | 1.69 | 80.44±0.40 | 0.68±0.01 | 1.69 |
> | 0.2 | 80.45±0.40 | 0.68±0.01 | 1.71 | 80.61±0.40 | 0.68±0.01 | 2.38 |
> | 0.4 | 80.43±0.40 | 0.68±0.01 | 1.73 | 80.58±0.40 | 0.68±0.01 | 3.25 |
> | 0.6 | 80.48±0.40 | 0.68±0.01 | 1.85 | 80.48±0.40 | 0.69±0.01 | 5.69 |
> | 0.8 | 80.54±0.40 | 0.68±0.01 | 2.06 | 80.43±0.40 | 0.72±0.01 | 8.93 |
> | 1.0 | 80.56±0.40 | 0.68±0.01 | 2.62 | 80.40±0.40 | 0.75±0.01 | 11.52 |

---

> > ### Author Response · Authors · 2025-11-20
> >
> > **Question 1:** _Please provide some empirical results on the independence of two modalities._
> >
> > **Response:** The independence assumption, which follows from building surrogate models that satisfy the common iid assumption, is crucial to obtaining a computationally tractable approach in our work. As detailed in Appendix C.1 (What happens without the iid assumption) and Eqs. 30 -- 33, even only estimating the Hessian matrix used for the Laplace approximation is computationally infeasable in a reasonable amount of time. Therefore, it is challenging to provide an empirical comparison to the model in which an independence assumption is not made.
> >
> > However, to provide quantitative evidence regarding the strength of cross-modal dependence, we performed an experiment estimating the relative curvature of the Hessian blocks $H_{PP}$, $H_{QQ}$, and $H_{PQ}$ for CLIP-B/32 (OpenCLIP) on LAION-400M. Based on those, we compared the curvature information in terms of the ratio of Frobenius norms of the respective Hessian blocks: $R = \frac{\lVert H_{PQ}\rVert_F}{\tfrac{1}{2}(\lVert H_{PP}\rVert_F + \lVert H_{QQ}\rVert_F)}$
> > A small ratio $R$ would indicate weak cross-modal dependence in the loss curvature and consequently also in the posterior over P and Q.
> >
> > Computing the full Hessian is computationally challenging due to its size ($\mathbb{R}^{512^2 \times (512 \cdot 796)}$ which is $\approx 1.6 \text{TiB}$ for Float32). Therefore, we adopt a Monte Carlo approximation based on randomly sampled gradient components per block, and estimated over five batches of size $\sim 32k$ and three random seeds. Further, we use the empirical-Fisher approximation for computational reasons, under which the Hessian of the contrastive loss with respect to the projection parameters can be written as $H \approx \mathbb{E}[g g^\top]$, where $g$ denotes the per-batch gradient. Note that the empirical-Fisher approximation is widely used in Laplace approximations as it is computationally more amenable but also less accurate than the GGN approximation used in our paper.
> >
> > For each Hessian block, we then estimated the Frobenius norm $\lVert H \rVert_F$ by averaging the squared magnitudes of the sampled entries. Consequently, we compute $R = \frac{\lVert H_{PQ}\rVert_F}{\tfrac12(\lVert H_{PP}\rVert_F + \lVert H_{QQ}\rVert_F)}$ across three seeds, where $R \approx 1$ would indicate strong dependence and a small ratio ($R$) correspond to weak dependence. The results for different sample sizes are shown in the table below.
> >
> > | Number of random samples | R |
> > |---|---|
> > | 50k | 0.363 ± 0.002 |
> > | 100k | 0.374 ± 0.035 |
> >
> > These results indicate only moderate cross-modal dependence in the Hessian (loss curvature). Hence, we can expect that a moderate amount of secondary interactions between both modalities are disregarded in our approximation, while the primary interactions are captured. Thus supporting our approximation. Capturing those additional secondary interactions in a computationally feasible manner is an interesting but challenging future direction.
> >
> > We want to emphasise that our approximation does not eliminate the multimodal coupling learned during pre-training. This is because the curvature is evaluated at the MAP solution obtained via multimodal contrastive learning. Therefore, the local posterior used by BayesVLM still reflects the cross-modal alignment encoded in the encoder. In practice, our approximation replaces an intractable exact posterior with a tractable one that retains the key geometric structure of the model while discarding interaction terms that, according to our experiment, have only a small or moderate effect.
> >
> > Lastly, note that this entanglement of modalities in the uncertainties obtained by BayesVLM can be observed through the qualitative experiment conducted in Section F.6 and Figure 11 in the Appendix. In this experiment, we gradually corrupt either modality and find that BayesVLM captures the dependencies in the change in the input in terms of the prediction and the predictive variance.
> >
> > If the reviewer has any additional suggestions for an empirical evaluation of the effect of the independence assumption, we would be happy to discuss them.

---

> > > ### Author Response · Authors · 2025-11-20
> > >
> > > **Question 2:** _For ImageNet setup, please include more ImageNet variants such as ImageNet-A._
> > >
> > > **Response:** We thank the reviewer for pointing out our additional experiments on datasets with a large distribution shift. To analyse the robustness of our approach in those settings, we have conducted additional zero-shot experiments on the adversarial ImageNet variants ImageNet-A and ImageNet-Sketch with the CLIP-Base model.
> > >
> > > The results below show no indication of epistemic underestimation: BayesVLM consistently improves calibration over CLIP and is competitive with or better than other post-hoc baselines. On ImageNet-A, it reduces ECE from 0.33 to 0.23 (on par with TTA), and on ImageNet-Sketch, it achieves the best calibration (0.08 vs. 0.13–0.16). While NLPD on ImageNet-A shows a small degradation relative to temperature scaling, BayesVLM still improves over CLIP. Even under a strong distribution shift, BayesVLM provides well-calibrated uncertainty estimates without signs of underestimating epistemic uncertainty. Note that TTA significantly improves accuracy and NLPD, especially for ImageNet-A. However, the increase in accuracy does not translate into a corresponding improvement in ECE. One possible explanation is that ImageNet-A often contains small or off-centre objects. With 64 cropped views, TTA is more likely to produce crops where the object is more easily identifiable, yielding higher confidence and more correct predictions.
> > >
> > > Crucially, because TTA uses 64 forward passes, its accuracy improvements largely stem from viewpoint search rather than improved uncertainty modelling. BayesVLM, in contrast, preserves single-pass efficiency and improves the quality of the uncertainty estimates and calibration. Note that BayesVLM can be combined with TTA, which could lead to both accuracy improvements and improvements in uncertainty estimates and calibration.
> > >
> > > We added these results to Appendix F.12.
> > >
> > > **ImageNet-A**
> > > | Method | Accuracy (%) | NLPD| ECE |
> > > |--------|------------------------|------------------------|------|
> > > | CLIP Zero-Shot | 25.87 ± 0.4379 | 3.77 ± 0.0312 | 0.33 |
> > > | Temp Scaling | 25.87 ± 0.4379 | 3.22 ± 0.0226 | 0.18 |
> > > | TTA | 38.67 ± 0.4870 | 2.77 ± 0.0279 | 0.23 |
> > > | BayesVLM | 26.67 ± 0.4422 | 3.36 ± 0.0257 | 0.23 |
> > >
> > > **ImageNet-Sketch**
> > > | Method | Accuracy (%) | NLPD| ECE |
> > > |--------|------------------------|------------------------|------|
> > > | CLIP Zero-Shot | 50.98% ± 0.4999 | 2.37 ± 0.0303 | 0.16 |
> > > | Temp Scaling | 50.98% ± 0.4999 | 2.29 ± 0.0286 | 0.13 |
> > > | TTA | 52.40% ± 0.4994 | 2.36 ± 0.0300 | 0.15 |
> > > | BayesVLM | 50.59% ± 0.5000 | 2.24 ± 0.0259 | 0.08 |

---

> > > > ### Author Response · Authors · 2025-11-26
> > > >
> > > > We hope that our rebuttal addressed the reviewers’ questions about our work. Should you have any further questions, we will be more than happy to answer them during the ongoing discussion phase.

---

### Official Review · Reviewer_VDEV · 2025-11-01

**Soundness:** 3
**Presentation:** 3
**Contribution:** 2
**Rating:** 6
**Confidence:** 3

**Summary:**

This paper proposes post-hoc uncertainty estimation in VLMs (CLIP) that does not require additional training. This method performs Laplace approximation in the last layer of encoder to induce probabilistic feature embeddings, which can quantify uncertainty. Then it transform the cosine similarity into a probablistic distribution, and use uncertainty score to support set selection in active learning. In the experiments, they show uncertainty estimates derived from this approximation improve the calibration of these models on several zero-shot classification benchmarks and are effective in active learning.

**Strengths:**

1. The mathmatical process is solid.
2. This is a post-hoc method, which does not require any retraining, fine-tuning, or modifications to the VLM architecture. It is only a Laplace approximation to quantify uncertainty. It is easy to apply.

**Weaknesses:**

1. The motivation to measure uncertainty of VLM (clip) is not attractive. The significance of the topic need be emphasized. Maybe it is for effectively selecting training data in active learning. If BayesVLM has more application fields, it will be better.
2. The method relies on two assumptions that may not always hold: (a) the image and text embeddings can be modeled with Gaussian distributions, and (b) the two modalities are independent. These are simplified situations.

**Questions:**

See weakness.

---

> ### Author Response · Authors · 2025-11-20
>
> We thank the reviewer for the time and effort put into reviewing our submission. We will respond to the questions raised in the review individually below.
>
> **Question 1:** _The motivation to measure uncertainty of VLM (clip) is not attractive. The significance of the topic need be emphasized. [...] If BayesVLM has more application fields, it will be better._
>
> **Response:** We agree that highlighting broader applications strengthens the paper. There are many scenarios where the Bayesian formulation used in BayesVLM and the resulting uncertainties can be beneficial. We list some directions below:
>
> - _Out-of-domain (OOD) and failure detection:_ An interesting and relevant application is to detect out-of-domain (OOD) or failure mode settings. Having access to uncertainties over the VLM embeddings directly enables both scenarios. For example, credible intervals obtained from our Bayesian treatment would be a promising direction to detect such settings. Our qualitative experiment in Appendix F.6 and Figure 11 indicate that the uncertainty estimates obtained by BayesVLM are a promising direction for such scenarios. In addition, at the request of reviewer 9dNF, we conducted further zero-shot experiments on OOD datasets ImageNet-A and ImageNet-Sketch, two natural distribution-shift benchmarks, and found that BayesVLM improves calibration relative to CLIP and is competitive with post-hoc baselines (see Appendix F.12).
> - _Uncertainty-aware retrieval:_ Another interesting and promising application of our work is in the context of retrieval, in which the uncertainties over the embedding projections can, for example, be incorporated into the retrieval process. This way, retrieval methods that utilise a VLM embedding space can more reliably detect cases where the data used in the retrieval task falls outside the VLM's training domain due to a distribution shift. Recent works have investigated non-Bayesian strategies for these settings with promising results [Chun et al., 2025].
> - _Probabilistic open-vocabulary classification:_ Lastly, we would also highlight the applicability of BayesVLM in open-vocabulary classification. While those are within the scope of classification tasks, it is worth mentioning that BayesVLM is one of the few Bayesian approaches that naturally support open-vocabulary classification and allow us to quantify uncertainties in a principled manner without knowing the number of classes a priori, while other methods typically require re-training or fine-tuning with a pre-defined set of classes.
>
> [Chun et al., 2025] Chun et al. "Probabilistic Language-Image Pre-Training". In ICLR 2025.

---

> > ### Author Response · Authors · 2025-11-20
> >
> > **Question 2:** _The method relies on two assumptions that may not always hold: (a) the image and text embeddings can be modeled with Gaussian distributions, and (b) the two modalities are independent._
> >
> > **Response (Gaussian embeddings):** We would like to clarify that our method _does not_ assume that image and text embeddings are Gaussian distributed. The Gaussianity solely results from the posterior approximation used in our work and is not a modelling assumption made in BayesVLM.
> >
> > Specifically, we use a Laplace approximation of the posterior distribution, which uses a local Gaussian approximation of the posterior. Using a Gaussian approximation of the posterior distribution is a common step in the Bayesian deep learning literature (e.g., variational inference with Gaussian approximation [Blei et al., 2017 and Shen et al., 2024] and Laplace approximations [Daxberger et al., 2021]) and can be relaxed if necessary. However, computations would be significantly more challenging and would require sampling-based approximations, which come with their own downsides and increase computational overhead (e.g., [Rezende et al., 2015 and Sladek et al., 2025]).
> >
> > Note that image and text features are obtained by applying a linear transformation parameterized by the projection-layer weights. Therefore, when using a Laplace approximation, the Gaussianity of the uncertainties over feature embeddings is a direct consquence as Gaussian random variables (the weights of the projection layers) remain Gaussian distributed under linear transformations.
> >
> >
> > **Response (independence assumption):** The independence assumption between modalities is a direct consequence of the iid assumption made in the paper (see Section 3.2, "Likelihood approximation" and Section C.2.1 in the appendix). Note that an iid assumption is a standard assumption in Bayesian settings. Doing so, allows us to build surrogate models for efficient Bayesian treatment and uncertainty quantification.
> >
> > As suggested by Reviewer 9dNF, we conducted an additional empirical experiment to assess the strength of the cross-modal dependence in the posterior distribution and Hessian matrix. *We provide the full experimental setup and results in our response to Reviewer 9dNF (Question 1)*.
> >
> > The results indicate a relatively moderate cross-modal dependence in the Hessian (loss curvature). Hence, we can expect that only a moderate amount of secondary interactions between both modalities are disregarded in our approximation, while the primary interactions are captured. Capturing those additional secondary interactions in a computationally feasible manner is an interesting but challenging future direction.
> >
> > We want to emphasise that our approximation does not eliminate the multimodal coupling learned during pre-training. This is because the curvature is evaluated at the MAP solution obtained via multimodal contrastive learning. Therefore, the local posterior used by BayesVLM still reflects the cross-modal alignment encoded in the encoders. In practice, our approximation replaces an intractable exact posterior with a tractable one that retains the key geometric structure of the model while discarding interaction terms that, according to our experiment, have only a small or moderate effect.
> >
> > Therefore, even though we have one independent surrogate model for each modality, the local posterior (uncertainty) estimates indirectly depend on all modalities. Thus, providing a practical and reasonable solution to the problem. A more detailed discussion of the effect of the independence assumption and the practical challenge of estimating the posterior without it is given in Appendix C.1 (What happens without the iid assumption). Additionally, we provided a derivation for the case where an independence assumption is not made (Eqs. 30 -- 33), showing that even estimating the local posterior with a Laplace approximation is computationally infeasable.
> >
> > [Blei et al., 2017] Blei et al. "Variational inference: A review for statisticians". Journal of the American statistical Association, 2017.
> >
> > [Shen et al., 2024] Shen et al. "Variational learning is effective for large deep networks". In ICML 2024.
> >
> > [Daxberger et al., 2021] Daxberger et al. "Laplace redux-effortless Bayesian deep learning". In NeurIPS 2021.
> >
> > [Rezende et al., 2015] Rezende. "Variational inference with normalizing flows". In ICML 2015.
> >
> > [Sladek et al., 2025] Sladek et al. "Approximate Bayesian inference via bitstring representations". In UAI 2025.

---

> > > ### Author Response · Authors · 2025-11-26
> > >
> > > We hope that our rebuttal addressed the reviewers’ questions about our work. Should you have any further questions, we will be more than happy to answer them during the ongoing discussion phase.

---

### Author Response · Authors · 2025-11-20

We thank the reviewers for their efforts and interest in our work, in which we presented _BayesVLM_, an efficient and effective post-hoc uncertainty quantification method for pre-trained VLMs. In our work, we leverage a Laplace approximation to the Bayesian posterior of the projection layers, thereby eliminating the need for additional training, architectural changes, or modifications to the training objective. For efficient posterior inference, we utilise independent probabilistic models for each modality and derive an analytical expression for the distribution over cosine similarities. Lastly, we showed the effectiveness of _BayesVLM_ on zero-shot and active learning tasks.

**Strengths:** We are happy to read that our paper has been described as "**mathematically solid**" [VDEV], that it provides "**better calibration performance [...] with small computation overhead**" [9dNF], and is "**novel and conceptually elegant**" [88Gp]. We also appreciate the recognition that our analytical expression of the distribution over cosine similarities (ProbCosine) is an "**elegant, interpretable, and computationally efficient**" [88Gp] approach that "is a **non-trivial mathematical contribution that generalizes beyond this specific application**" [88Gp].

**Common questions:** We identified the following common questions and concerns about our paper. Below, we give a high-level response to those questions and the changes/additional experiments performed. Detailed responses are found in the individual responses to each reviewer.

- **Validity of the independence assumption [VDEV, 9dNF, 88Gp]:** A recurring question of the reviewers was about the independence assumption made in our work. This assumption follows from building surrogate models that satisfy the common iid assumption and is crucial for efficient computation of the approximate posterior. While a full assessment of the assumption is computationally challenging, we conducted a proxy experiment to provide insights into its validity. Empirically, we found only moderate cross-modal dependence in the Hessian, indicating that, while some secondary interactions between the modalities are disregarded, the primary interactions are not cross-modal and are therefore captured by BayesVLM. Thus supporting our approach. Further details are given in the comments.
- **Sensitivity/bias wrt the proxy dataset [9dNF, 88Gp]:**  Some reviewers raised concerns about potential bias from/sensitivity to the proxy dataset chosen in closed-source settings. In response to these concerns, we performed an ablation in which we progressively distorted images of the LAION-400M dataset to simulate an increase in distribution shift/bias in the proxy dataset. Our results indicate that BayesVLM remains robust under mild shifts, while stronger deviations (particularly those that compromise semantic content) lead to increased calibration error and higher NLPD. Further details are given in the comments.

**Changes to the manuscript:** We have updated the manuscript with additional experiments in Appendix F.11 (Robustness wrt distribution shift of the Proxy Dataset) and F.12 (Additional Zero-shot Results on Adversarial ImageNet Variants) and colour-coded the changes in red. Additionally, following reviewer feedback, we moved the qualitative uncertainty analysis from Appendix F.6 into Section 4.3 in the main text.

We reply to each reviewer separately below in the comment section. Please let us know if you have any further questions.

---

### Meta-Review · Area_Chair_UBgY · 2026-01-07

**Summary:**

The reviews are all positive with 6/6/6, with consistent acknowledgement of (i) a mathematically solid formulation and (ii) a practical, training-free post-hoc uncertainty method for VLMs with small overhead. The concerns across reviewers concentrate on scope/framing (the importance of uncertainty for VLMs matters and broader applicability, VDEV), core approximations/assumptions, e.g., modality independence, restricting Bayesian treatment to projection layers, proxy-data Hessian reliability, robustness under large distribution shift,  and positioning vs adapter/probabilistic baselines, and missing discussion or comparison with related works.

The rebuttal adds substantial new experiments and analyses, including cross-modal dependence estimation, proxy-data mismatch sensitivity studies, evaluations on ImageNet-A and ImageNet-Sketch, and a clearer justification for restricting the Bayesian treatment to the projection layers. These additions largely address reviewer concerns regarding assumptions, robustness, and scope. Most reviewer questions related to specific technical components are directly and satisfactorily addressed in the detailed rebuttal.
That said, the most significant remaining concerns relate to the core modality independence assumption and the positioning and comparison with existing related work. These issues stem from fundamental design choices and, while they do not invalidate the approach, they require clearer exposition and more careful framing. In particular, these aspects should be explicitly discussed in the main paper to improve clarity, properly contextualize the contribution, and avoid potential overclaiming.

AC recommends acceptance, considering its overall value particularly in practice, conditional on careful revision for the final version. The AC strongly encourages the authors to thoroughly incorporate the rebuttal content into the main paper and to explicitly address the reviewers’ questions and concerns in the final revision.

**Reviewer Concerns:**

The main concern raised by Reviewer VDEV relates to the broader applicability and motivation of uncertainty estimation for vision–language models. The rebuttal provides expanded discussion and clearer explanations, which largely address this concern.

Reviewer 9dNF raised questions regarding the empirical evidence supporting the independence assumption (also noted by VDEV and 88Gp), sensitivity to proxy data, and robustness under large distribution shifts. The authors provided additional experiments and analyses to address these issues. Overall, most concerns appear to be addressed, although some minor to moderate reservations may remain. For instance, while the rebuttal clearly explains the rationale for restricting the Bayesian treatment to the projection layers, this design choice may still leave some residual concern. Similarly, the added ImageNet-A and ImageNet-Sketch experiments strengthen the case for robustness, but may not fully settle all questions regarding behavior under extreme distribution shifts.

Reviewer 88Gp raised concerns about the independence assumption (overlapping with 9dNF), the limited scope of the Bayesian treatment, comparisons with probabilistic embedding baselines (notably ProLIP and CLAP4CLIP), and scalability in active learning settings. The rebuttal responds to these points and partially addresses them. In particular, only limited empirical evidence is provided for the independence assumption, and while additional comparisons and explanations are included, the discussion of direct baselines could be further strengthened. The relationship with ProLIP and CLAP4CLIP should be more carefully and transparently positioned in the main paper. Presenting only PCME++ in the main text is somewhat limiting and tricky; results for ProLIP should also be explicitly discussed. Although CLAP4CLIP is designed for downstream adaptation, its methodology is broadly relevant and should be acknowledged accordingly. Overall, the authors should refine the positioning of the proposed method in the final version and avoid potential overclaiming.

**Reviewer Scores:**

VDEV. Likely stays 6. The rebuttal directly answers the two key weaknesses (motivation/applicability and assumption clarification). I don’t see a strong push to 7 unless the main paper’s framing becomes substantially more compelling, which is hard during rebuttal.


9dNF: Likely stays 6, maybe with increased confidence. Their main questions  (independence evidence, ImageNet variants, proxy sensitivity) were all addressed with additional experiments/analysis.

88Gp: Likely stays 6. They were already positive; most concerns were addressed. Remaining issues are mainly “scope/limitations” rather than correctness, so a move to 7 is unlikely but conceivable if the revised paper makes the contributions and positioning unmistakably crisp. And the responses to the question related to the direct related works may not fully satisfy the reviewer.

---

### Decision · Program_Chairs · 2026-01-26

Accept (Poster)